# A Biconvex Formulation for Transport of Mixture Models

## Abstract

Optimal transport (OT) provides a principled framework for mapping between probability distributions. Despite extensive progress in the field, OT remains computationally demanding, and the resulting transport plans are often difficult to interpret. Here, we propose Optimal Mixture Transport (OMT), an efficient algorithm that leverages mixture modeling and entropic regularization to yield interpretable transport plans. We show that transport between mixtures, in particular mixtures of Gaussians which are universal approximators in $L^2$, can be formulated as a biconvex optimization problem with a unique minimizer. This formulation not only reduces computational cost, but also provides component-level correspondences, offering insights into complex distributions. We demonstrate the practicality and effectiveness of OMT across a diverse collection of synthetic benchmarks and real-world datasets, including large-scale single-cell RNA sequencing measurements.

## 1 Introduction

Optimal Transport (OT) offers a powerful mathematical framework for comparing probability distributions and finding optimal mappings between them (Santambrogio, 2015). Its versatility has led to advances in diverse fields, including domain adaptation (Grave et al., 2019; Struckmeier et al., 2023; Chuang et al., 2023; Fernandes Montesuma et al., 2025), data integration and alignment (Demetci et al., 2022), and predicting cell fates (Tong et al., 2020; Bunne et al., 2023; 2024). At its core, OT seeks to find the most cost-effective way to transform one probability distribution into another, subject to constraints on the total mass being transported (Peyré et al., 2019; Villani et al., 2008).

A major challenge in OT has been its high computational cost. Cuturi (2013) introduced an entropy regularization term to the OT objective to obtain a strictly convex problem (EOT) and an elegant solution known as the Sinkhorn algorithm. However, even with EOT, sample-to-sample transportation remains limited by the curse of dimensionality and can be slow on large datasets (Genevay et al., 2018). To mitigate this, mini-batch strategies (MB-OT) have been developed to approximate the transport plan by operating on subsets of the data (Genevay et al., 2018; Fatras et al., 2021b;a). While computationally cheaper, these methods often yield suboptimal transport plans, as cost estimation from subsets can be inaccurate and satisfying the mass preservation constraint of balanced OT becomes difficult.

To improve transport accuracy over batches, one prominent class of methods approximates the OT path by non-parametric interpolation within the Wasserstein space. These techniques range from deterministic approaches, such as Progressive Optimal Transport (ProGOT) (Kassraie et al., 2024), to stochastic methods based on gradient flows and Schrödinger bridges (Albergo & Vanden-Eijnden, 2023; Albergo et al., 2024). Stochastic methods, often employing neural networks such as those based on gradient flows (Daniels et al., 2021) or Schrödinger bridges (Gushchin et al., 2023b;a), also typically necessitate inner iterations to achieve accurate transport maps. Such methods construct a sequence of intermediate distributions to bridge the source and target, often requiring numerous intermediate steps, significant memory overhead, and many inner iterations to converge to an accurate solution. Furthermore, simpler displacement strategies, like the McCann interpolation (McCann, 1997) used in ProGOT, do not always produce interpretable intermediate distributions. While regularization techniques, can improve the robustness of the transport map approximation (Buzun et al., 2024b), their performance is sensitive to the regularization parameters.

In contrast to non-parametric and relaxation-based approaches, an effective strategy for large-scale problems is to adopt a parametric model of the data, thereby simplifying the task. Following this direction, we propose Optimal Mixture Transport (OMT), an efficient and scalable framework for computing EOT between mixture models. While EOT is generally intractable for most parametric families, a closed-form solution exists for transport between two Gaussian distributions in both balanced and unbalanced settings (Janati et al., 2020). Building on this result, we tailor the framework to the Gaussian family, focusing on Gaussian Mixture Models (GMMs). This specialization is powerful as GMMs are universal function approximators capable of representing any sufficiently smooth density with arbitrary precision (Goodfellow et al., 2016). Our formulation recasts the transport problem as a uniquely solvable biconvex optimization, yielding a computationally efficient and theoretically grounded alternative for large-scale transport tasks. Furthermore, we observe that one limitation of many learned coupling transport maps or dual functions is their pronounced directional bias (source $\rightarrow$ target), which leads to performance degradation when inverted. In contrast, we show that OMT maps remain robust regardless of transport direction.

Our main contributions are summarized as follows:

- We propose a parametric EOT framework, called OMT, which operates by transporting sub-populations rather than individual samples.

- We show that the OMT formulation is strictly biconvex and, when solved as part of a global optimization algorithm, this subproblem converges to a unique solution in a single step.

- Building on closed-form results for entropic Gaussian transport, we propose a formulation of OMT within Gaussian distributions, as a flexible and expressive parametric family.

- Through experiments on synthetic and real-world datasets, we demonstrate that OMT consistently matches or surpasses the performance of state-of-the-art OT solvers, while requiring substantially less computation and memory.

## 2 BACKGROUND

**Optimal transport:** For $\mathcal{X}, \mathcal{Y} \subset \mathbb{R}^d$, probability measures $\mu_0, \mu_1 \in \mathcal{P}(\mathbb{R}^d)$ and $c : \mathcal{X} \times \mathcal{Y} \to \mathbb{R}$, a cost function associated with transporting a unit of mass from a point in $\mathcal{X}$ to a point in $\mathcal{Y}$, the minimum total cost of transport can be obtained as

$$\inf_{T\sharp\mu_0=\mu_1} \int_{\mathcal{X}} c(\mathbf{x}, T(\mathbf{x})) \, d\mu_0(\mathbf{x}), \tag{1}$$

where $T_\sharp\mu_0 = \mu_1$ denotes the pushforward of $\mu_0$ by $T$, defined as $\mu_1(A) = \mu_0(T^{-1}(A)), \forall A \subset \mathcal{Y}$, ensuring mass conservation. Problem (1) is known as the Monge problem (Peyré et al., 2019) which seeks a map $T : \mathbb{R}^d \to \mathbb{R}^d$ referred to as the transport map between $\mu_0$ and $\mu_1$. The Monge formulation is often problematic because the optimization is over a non-convex set of maps, and a deterministic map $T$ may not exist. To bypass this, Kantorovich presented a relaxed formulation, which seeks a distribution $\pi \in \mathbb{R}^d \times \mathbb{R}^d$ referred to as "coupling" of $\mu_0$ and $\mu_1$, as

$$\inf_{\pi\in\prod(\mu_0,\mu_1)} \int_{\mathcal{X}\times\mathcal{Y}} c(\mathbf{x}, \mathbf{y}) \, d\pi(\mathbf{x}, \mathbf{y}), \tag{2}$$

where $\prod(\mu_0, \mu_1) := \{\pi \in \mathcal{P}(\mathcal{X} \times \mathcal{Y}) \mid P_{\mathcal{X}} : \pi \to \mu_0, P_{\mathcal{Y}} : \pi \to \mu_1\}$.

When $\mathcal{X} = \mathcal{Y}$ and $c(\mathbf{x}, \mathbf{y}) = d(\mathbf{x}, \mathbf{y})^p$, $p \geq 1$, where $d$ is a distance on $\mathcal{X}$, the Kantorovich problem is equivalent to the Wasserstein p-distance between probability measures, $\mathcal{W}_p(\mu_0, \mu_1)$. Specifically, for $c(\mathbf{x}, \mathbf{y}) = \|\mathbf{x} - \mathbf{y}\|_2^2$, the Kantorovich problem yields the squared Wasserstein-2 distance:

$$\mathcal{W}_2^2(\mu_0, \mu_1) = \inf_{\pi\in\prod(\mu_0,\mu_1)} \int_{\mathcal{X}\times\mathcal{Y}} \|\mathbf{x} - \mathbf{y}\|_2^2 \, d\pi(\mathbf{x}, \mathbf{y}). \tag{3}$$

According to the Brenier Theorem (Santambrogio, 2015), in (3), if $\mu_0$ and $\mu_1$ are absolutely continuous with respect to the Lebesgue measure, there exists a unique optimal solution that can be expressed as $\pi^* = (I_d \times T^*)\sharp\mu_0$, where $I_d$ stands for the identity map, and $T^*$ is the unique minimizer of (1). Moreover, the unique optimal transport $T(\mathbf{x}) = \nabla\phi(\mathbf{x})$, where $\phi$ is a convex function.

**Entropic optimal transport:** Entropic optimal transport introduces an entropic regularization term to (3), transforming the problem into a strictly convex optimization that can be efficiently solved using algorithms like the Sinkhorn-Knopp method (Cuturi, 2013; Janati et al., 2020). For a regularization parameter $\varepsilon > 0$, the entropic optimal transport cost is defined as:

$$d_\varepsilon(\mu_0, \mu_1) = \inf_{\pi \in \Pi(\mu_0, \mu_1)} \left\{ \int_{\mathcal{X} \times \mathcal{Y}} \|\mathbf{x} - \mathbf{y}\|_2^2 \, d\pi(\mathbf{x}, \mathbf{y}) - 2\varepsilon H(\pi) \right\}. \tag{4}$$

In addition to convexity, entropy regularization encourages parsimonious, smoother maps, rather than concentrating mass in a one-to-one mapping Jaynes (1957).

Considering $D_{KL}(P\|Q) = \int dP \left( \log \frac{dP}{dQ} - 1 \right) + dQ$, minimizing the objective in (4) is equivalent to minimizing

$$\min_{\pi \in \prod(\mu_0, \mu_1)} \int_{\mathcal{X} \times \mathcal{Y}} \|\mathbf{x} - \mathbf{y}\|_2^2 \, d\pi(\mathbf{x}, \mathbf{y}) + 2\varepsilon D_{KL}(\pi\|\mu_0 \otimes \mu_1).$$

**Transport between Gaussian measures:** When both mass measures are Gaussian distributions, i.e., $\mu_0 = \mathcal{N}(\mathbf{m}_0, \boldsymbol{\Sigma}_0)$ and $\mu_1 = \mathcal{N}(\mathbf{m}_1, \boldsymbol{\Sigma}_1)$, (3) is simplified as

$$\mathcal{W}_2^2(\mathcal{N}(\mathbf{m}_0, \Sigma_0), \mathcal{N}(\mathbf{m}_1, \Sigma_1)) = \|\mathbf{m}_0 - \mathbf{m}_1\|_2^2 + tr\{\Sigma_0 + \Sigma_1 - 2\Gamma\}, \tag{5}$$

where $\Gamma = \left( \Sigma_0^{\frac{1}{2}} \Sigma_1 \Sigma_0^{\frac{1}{2}} \right)^{\frac{1}{2}}$ (Bhatia et al., 2019; Janati et al., 2020). Furthermore, the optimal transport map $T^* : \mathbb{R}^d \to \mathbb{R}^d$ that pushes $\mu_0$ forward to $\mu_1$ admits a closed-form as follows.

$$T^*(\mathbf{x}) = A(\mathbf{x} - \mathbf{m}_0) + \mathbf{m}_1, \tag{6}$$

where $A = \Sigma_0^{-1} \# \Sigma_1 = \Sigma_0^{-\frac{1}{2}} \left( \Sigma_0^{\frac{1}{2}} \Sigma_1 \Sigma_0^{\frac{1}{2}} \right)^{\frac{1}{2}} \Sigma_0^{-\frac{1}{2}} = \Sigma_0^{-\frac{1}{2}} \Gamma \Sigma_0^{-\frac{1}{2}}$, corresponding to the geometric mean of the precision and covariance matrices at source and target points, respectively.

**Gaussian mixture models:** Gaussian mixture models (GMMs) are widely used for density estimation due to several key advantages: (i) As a linear combination of Gaussian distributions, GMMs allow for analytical tractability and have favorable asymptotic properties. (ii) GMMs are universal approximators for continuous density functions: any smooth density can be approximated with arbitrary accuracy by a mixture of Gaussians with enough number of components (Titterington et al., 1985; Scott, 2015; Zeevi & Meir, 1997). (iii) Many real-world datasets are naturally organized into clusters with unimodal distributions, making GMMs particularly effective for modeling such structures. These motivate our exploration of the optimal transport problem for Gaussian mixtures.

Using GMMs for density approximation involves approximating density functions by a convex combination of "basis" densities (Zeevi & Meir, 1997). Consider the set of square-integrable density functions in $\mathbb{R}^d$, denoted as $\mathcal{F} = \{f \mid f \in L^2(\mathbb{R}^d), \ f \geq 0, \ \int_{\mathbb{R}^d} f(\mathbf{x}) d\mathbf{x} = 1\}$. We define the set of GMM densities with $K$ components, $\mathcal{G}_K$, as:

$$\mathcal{G}_K = \left\{ f_K^{\boldsymbol{\theta}} \mid f_K^{\boldsymbol{\theta}}(\cdot) = \sum_{i=1}^K \alpha_i \phi(\cdot, \mu_i, \Sigma_i), \ \alpha_i > 0, \ \sum_{i=1}^K \alpha_i = 1 \right\}, \tag{7}$$

where $\boldsymbol{\theta} = \{\alpha_i, \boldsymbol{\mu}_i, \Sigma_i\}_{i=1}^K$ represents the collection of parameters for the $K$ components. $\mathcal{G}_K \subset \mathcal{F}$ and the universal approximation property implies that for any $f \in \mathcal{F}$, $\lim_{K \to \infty} \inf_\theta \mathcal{D}(f, f_K^{\boldsymbol{\theta}}) = 0$, where $\mathcal{D}$ denotes a distance (Titterington et al., 1985; Zeevi & Meir, 1997).

## 3 RELATED WORK

To address the limitations of Sinkhorn-based methods, researchers turned to deep learning, giving rise to Neural Optimal Transport (Neural OT) (Makkuva et al., 2020; Korotin et al., 2023), which uses neural networks to learn a continuous mapping between distributions, while enforcing theoretical constraints (Genevay et al., 2018; Buzun et al., 2024b). Another direction directly learns the transport map via neural networks, transforming samples from a source to a target distribution. This approach

is widely used in domain adaptation and generative modeling, where models such as normalizing flows learn invertible maps from simple to complex distributions. A prominent example of neural OT connects diffusion models with the theory of Schrödinger Bridges De Bortoli et al. (2021); Shi et al. (2023); Gushchin et al. (2024b;a), a classic stochastic transport problem. This establishes a learning framework for diffusion models equivalent to solving a Schrödinger Bridge problem, which can be viewed as a form of neural EOT (Gushchin et al., 2024b). Broadly, stochastic and neural OT methods typically require extensive training with many samples, making the process slower and computationally expensive.

Alongside neural methods, several non-neural OT solvers address scalability through iterative, mini-batch frameworks or low-rank factorizations. PROGOT (Kassraie et al., 2024) constructs the transport map sequentially. While this approach can be parallelized using libraries like OTT-Jax (Cuturi et al., 2022), it remains memory-intensive. Low-rank OT (LOT) methods (Scetbon et al., 2021; Scetbon & Cuturi, 2022; Halmos et al., 2024; 2025) improve efficiency by factorizing the coupling matrix into low-rank matrices. HiRef (Halmos et al., 2025) uses a hierarchical refinement strategy, applying LOT across multi-scale partitions to achieve state-of-the-art performance among LOT solvers. However, it is restricted to learning bijective Monge maps and cannot model transport couplings that involve mass splitting or merging, a critical feature for many real-world applications. Moreover, HiRef cannot guarantee a unique solution when the cost matrix is not *r-Monge separable* (Halmos et al., 2025).

Parametric OT simplifies the transport problem by assuming data distributions belong to a parametric family, which often yields computationally more efficient solutions. A prominent example involves Gaussian distributions, for which both $\mathcal{W}_2(\mu_0, \mu_1)$ and its entropically regularized version (Eq. 4) admit closed-form solutions Kassraie et al. (2024). Building on this, the parametric formulation has been extended to the more general case of Gaussian Mixture Models (GMMs). This body of work approximates (bounds) the Wasserstein distance between Gaussian components, proposed as the aggregated Wasserstein distance (Chen et al., 2019) or $GW_2$ (Delon & Desolneux, 2020), by considering the transport between their individual components, to reduce complexity by depending on the number of mixture components instead of data points and offer a scalable solution for high-density data. However, most existing studies are limited to simpler applications, such as simple 2D tasks and color transfer. A recent extension leverages $GW_2$ for unsupervised domain adaptation, facilitating label transfer from a source domain to a target domain (Fernandes Montesuma et al., 2025). A key challenge in using $GW_2$ lies in optimizing the component weights, which reduces the task to a discrete OT problem, a computationally challenging paradigm that may lack a unique solution (see Appendix B.4).

Here, we extend parametric OT to the entropic mixture transport setting. We show that this formulation is strictly biconvex and guarantees uniqueness for both the transport plan over mixing weights and the individual component distributions. Moreover, building on Kassraie et al. (2024), we focus on over-parameterized GMM, a regime in which a global convergence of Expectation-Maximization can be established (Xu et al., 2024), supporting practical viability across different tasks.

## 4 TRANSPORT PROBLEM FOR MIXTURE MODELS

Let $\nu \in M_K(\mathbb{R}^d)$ denote a mixture model in $\mathbb{R}^d$ with $K$ components:

$$\nu = \sum_{i=0}^{K} \alpha_i \mu_i, \tag{8}$$

where $\mu_i$ are probability measures and $\sum_i \alpha_i = 1$, $\alpha_i \geq 0, \forall i$.

**Definition 1** (Mixture transport coupling). *Given two measures $\nu_0 \in M_{K_0}(\mathbb{R}^d)$ and $\nu_1 \in M_{K_1}(\mathbb{R}^d)$, we define the mixture transport coupling as follows:*

$$\pi_{\mathcal{M}}^* \quad := \quad \underset{\pi \in \prod(\nu_0, \nu_1) \cap M_K(\mathbb{R}^{2d})}{\arg\min} \int_{\mathcal{X} \times \mathcal{Y}} \|\mathbf{x} - \mathbf{y}\|_2^2 \, d\pi(\mathbf{x}, \mathbf{y}), \tag{9}$$

*where $\prod(\nu_0, \nu_1) := \{\pi \in \mathcal{P}(\mathcal{X} \times \mathcal{Y}) \mid P_{\mathcal{X}} : \pi \to \nu_0(\mathbf{x}), P_{\mathcal{Y}} : \pi \to \nu_1(\mathbf{y})\}$ and $K \leq K_0 K_1$.*

Therefore, the transport policy belongs to the mixture model family and can be expressed as

$$d\pi(\mathbf{x}, \mathbf{y}) = \sum_{i=1}^{K} \omega_i dp_i(\mathbf{x}, \mathbf{y}), \quad \text{where } \forall i, \ p_i \in \mathcal{P}(\mathbb{R}^d).$$

Note that the trivial choice of $dp_1 = \ldots = dp_K$, $\omega_1 = \ldots = \omega_K = 1/K$ makes the solver in (9) equal to $\mathcal{W}_2(\nu_0, \nu_1)$ in (3). We next constrain that marginals of the components of the transport coupling are the same as the components of the source and target functions:

$$\mathcal{D}_{\mathcal{M}}(\nu_0, \nu_1) = \min_{\Omega, P} \sum_{i,j} \omega_{ij} \int_{\mathcal{X} \times \mathcal{Y}} \|\mathbf{x} - \mathbf{y}\|_2^2 \, dp_{ij}(\mathbf{x}, \mathbf{y}),$$

$$\text{s.t.} \qquad \mathbf{1^T}\Omega = \boldsymbol{\alpha}_0, \qquad \Omega^T \mathbf{1} = \boldsymbol{\alpha}_1$$

$$\forall i, \; \sum_j \int_{\mathcal{Y}} dp_{ij}(\mathbf{x}, \mathbf{y}) = \mu_{0_i}, \qquad \forall j, \; \sum_i \int_{\mathcal{X}} dp_{ij}(\mathbf{x}, \mathbf{y}) = \mu_{1_j}, \qquad (10)$$

where $\Omega = [w_{ij}]$ denotes the matrix of mixture weights. With this constraint, $\mathcal{D}_{\mathcal{M}}(\nu_0, \nu_1) \geq \mathcal{W}_2(\nu_0, \nu_1)$ and equality is achieved in the $K \to \infty$ limit when the source and target functions are over-parametrized by a dense mixture model family (e.g., GMM (Goodfellow et al., 2016)). The problem in (10) is similar to minimizing the aggregated Wasserstein distance, which was proposed for comparing hidden Markov models with Gaussian state conditional distributions (Chen et al., 2019).

### 4.1 REGULARIZED MIXTURE TRANSPORT

A common approach to ensure the uniqueness of solutions in optimal transport problems is to introduce an entropy regularization term, which makes the objective function strictly convex and improves the numerical stability of optimization. In the context of mixture transport optimization in (10), we adopt a similar approach by incorporating a weighted average entropy term as a regularizer.

**Definition 2** (Optimal Mixture Transport). *We introduce two forms of regularization into the mixture transport problem* (9)*: (i) a component-wise regularizer, and (ii) a mixing-matrix regularizer, controlled respectively by parameters $\varepsilon_1, \varepsilon_2 > 0$. The resulting problem is formulated as the following optimization:*

$$\mathcal{D}_{OMT} := \min_{\omega_{ij}, dp_{ij}} \sum_{i,j}^K \omega_{ij} \left[ \int_{\mathcal{X} \times \mathcal{Y}} \|\mathbf{x} - \mathbf{y}\|_2^2 \, dp_{ij}(\mathbf{x}, \mathbf{y}) - \varepsilon_1 H(p_{ij}) \right] - \varepsilon_2 H(\Omega) \,.$$

$$\text{for} \;\; \Omega = [\omega_{ij}]_{K_0 \times K_1} \in \mathcal{S}^{K-1}, \; P = [p_{ij}]_{K_0 \times K_1}, \text{where } p_{ij} \in \prod(\mu_{0_i}, \mu_{1_j}) \qquad (11)$$

Minimizing the objective in (11) is equivalent to minimizing $\mathcal{L}_{\varepsilon_1, \varepsilon_2}(\Omega, P)$, defined as follows.

$$\mathcal{L}_{\varepsilon_1, \varepsilon_2}(\Omega, P) = \sum_{i,j}^K \omega_{ij} \left[ \int_{\mathcal{X} \times \mathcal{Y}} \|\mathbf{x} - \mathbf{y}\|_2^2 \, dp_{ij}(\mathbf{x}, \mathbf{y}) + \varepsilon_1 D_{KL}(p_{ij} \| \mu_{0_i} \otimes \mu_{1_j}) \right] + \varepsilon_2 D_{KL}(\Omega \| \boldsymbol{\alpha}_0 \otimes \boldsymbol{\alpha}_1)$$

$$(12)$$

**Remark 1.** *The problem in Eq. 11 is a generalization of entropic optimal transport in the sense that Eq. 11 collapses to entropic optimal transport when $K_0 = K_1 = 1$.*

Therefore, we consider an optimization problem of the form

$$\min_{\Omega, P} \mathcal{L}_{\varepsilon_1, \varepsilon_2}(\Omega, P)$$

$$\text{s.t.} \quad \mathbf{1}\Omega = \boldsymbol{\alpha}_0, \;\; \Omega^T \mathbf{1} = \boldsymbol{\alpha}_1, \;\; \int_{\mathcal{Y}} dp_{ij}(\mathbf{x}, \mathbf{y}) = \mu_{0_i}, \;\; \int_{\mathcal{X}} dp_{ij}(\mathbf{x}, \mathbf{y}) = \mu_{1_j}, \qquad (13)$$

where $\Omega \in \mathcal{S}^{K-1}$ and $P \in \mathcal{P}^K(\mathcal{X} \times \mathcal{Y})$.

Eq. 4.1 no longer defines a convex program. However, as we show now in Lemma 1, the objective is biconvex. Moreover, while biconvex problems don't have unique solutions generally, Eq. 4.1 has a unique minimizer that can be obtained efficiently (Theorem 1 and Corollary 1 below).

**Lemma 1.** *For any $\varepsilon_1, \varepsilon_2 > 0$, $\mathcal{L}_{\varepsilon_1, \varepsilon_2}(\Omega, P)$ is strictly biconvex.* (Proof in Appendix A)

Floudas & Visweswaran (1990) proposed the *Global Optimization Algorithm* (GOP) to solve constrained biconvex problems. It decomposes the optimization into disjoint blocks similar to the *Alternate Convex Search*(ACS) method and exploits the convex substructure of the problem by a

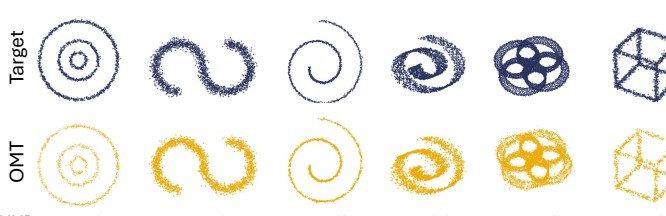

Figure 1: Transporting samples from a normal distribution to various target distributions with entropic OMT. **Top:** target point cloud distributions. **Bottom:** distributions generated by OMT after training with $10,000$ samples. MMD between the target and OMT-generated samples expressed as a percentage.

primal-relaxed dual approach (Gorski et al., 2007). The GOP algorithm is guaranteed to terminate after a finite number of steps for an $\epsilon$-global optimum solution, for any $\epsilon > 0$ (**Theorem 4.11**, **Corollary 4.12**, Ref. (Gorski et al., 2007)). As mentioned above, the uniqueness of this global optimum is not guaranteed in the general case. However, by exploiting the structure of Eq. 4.1, we show that its solution is unique and obtained in a single iteration:

**Theorem 1.** *For the optimization problem defined in* (4.1)*, the GOP algorithm converges to a unique solution in a single iteration.* (Proof in Appendix A)

## 4.2 REGULARIZED MIXTURE TRANSPORT FOR GMMS

If the probability measures $\nu_0 \in G_{K_0}(\mathbb{R}^d)$ and $\nu_1 \in G_{K_1}(\mathbb{R}^d)$ are defined as mixtures of Gaussian distributions, by using the results from the optimal mixture transport framework introduced in Section 4.1, we can compute an optimal transport plan between the two GMMs. Notably, the resulting optimal mixture transport between two GMMs can also be shown to be a GMM itself, thereby preserving the Gaussian structure in the transported distribution.

**Corollary 1.** *Let $\nu_0 \in G_{K_0}(\mathbb{R}^d)$ and $\nu_1 \in G_{K_1}(\mathbb{R}^d)$ be two Gaussian mixture models (GMMs) in $\mathbb{R}^d$ with $K_0$ and $K_1$ components, respectively. Then, the optimal mixture transport map between $\nu_0$ and $\nu_1$ is itself a Gaussian mixture model with $K$ components, where $K \leq K_0 K_1$.*

Stability of the transport plan with respect to perturbations of the source or target distributions is an appealing property (Divol et al., 2025). While such theoretical guarantees do not typically exist for Neural OTs, they are an important feature of multiple non-Neural OT formulations (Divol et al., 2025; Kassraie et al., 2024). We first introduce a notion of stability under source perturbation (Appendix A.1). Then, we demonstrate the stability of OMT maps within compact sets (Theorem 2).

**Theorem 2** (Stability of OMT under perturbation)**.** (Informal version) *Bounded changes to the source distribution within a compact set results in bounded changes to the OMT map.* (Full derivations and proof in Appendix A.1.2)

## 5 EXPERIMENTS

**Synthetic datasets.** We conduct two sets of simulation experiments. The first set focuses on synthetic 2D tasks with multiple target distributions, designed to demonstrate the capability of the proposed optimal mixture transport strategy. As shown in Figure 1, OMT successfully recovers the target shapes across all cases. In these tasks, the source data is sampled from a normal distribution. The second set of experiments evaluates our method on the W2-Benchmark tasks (Korotin et al., 2021), which are widely adopted in recent studies on both neural and non-neural OT. We compare the proposed OMT method against state-of-the-art approaches: ExNOT (Buzun et al., 2024a), an entropic Neural-OT solver, ENOT (Gushchin et al., 2024b), a diffusion-based Neural-OT, PROGOT (Kassraie et al., 2024), amortized W2-OT (Amos, 2022), GMM-OT (Delon & Desolneux, 2020), and the classical entropic OT (EOT) solver. Figure 2 presents the comparative performance of OMT across three evaluation metrics: Sinkhorn divergence ($D_\varepsilon$), mean squared error (MSE), and runtime. In all experiments, OMT was trained with $K_s = 3$, $K_t = 15$, $\varepsilon_{1,2} = 0.01$. For dimensions $d > 64$, we impose a diagonal structure on the covariance matrix instead of using the full covariance. Appendix B reports the transport cost ($T_c$) and total memory usage for each method. As shown in Figures 2 and 7 (Appendix), OMT outperforms EOT, W2-OT, GMM-OT, ENOT and in most cases, PROGOT. It also outperforms ExNOT at higher latent dimensions. Note that methods like PROGOT and EOT are sample-based solvers, whereas OMT, similar to neural OT, solves the continuous transport problem at the distribution level. Despite this difference, OMT still performs reasonably well on sample-to-sample metrics such as MSE. Considering all metrics along with transport costs, OMT achieves

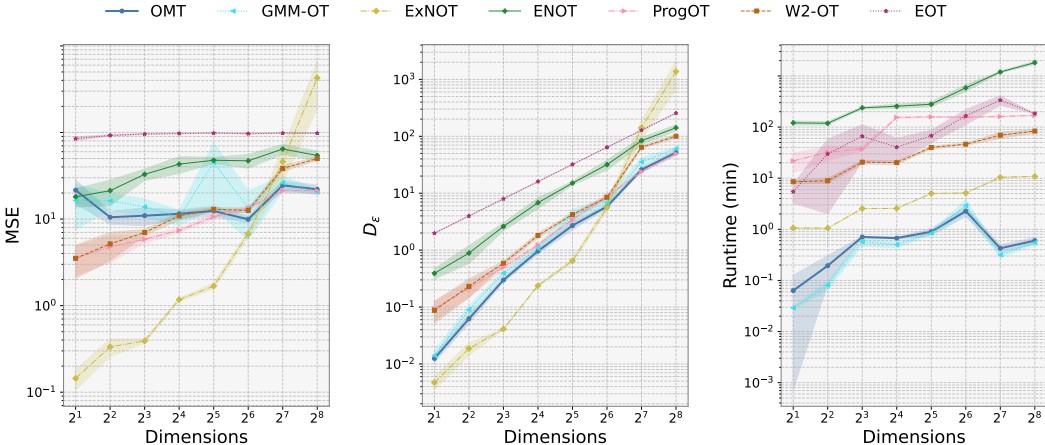

Figure 2: Comparison of OMT against baseline methods on the Wasserstein-2 benchmark tasks in Korotin et al. (2021). The reported plots are averaged over both forward and backward directions. All results are evaluated on the test set with $10,000$ samples and averaged across five random initializations. MSE captures fidelity in sample-to-sample transportation whereas $D_\varepsilon$ is more suitable for transportation between distributions. The runtime is measured on allocated nodes of a cluster, each equipped with one NVIDIA A100 GPU, 4 Intel Xeon Gold 6330N CPU cores, and 128 GB of RAM. The reported time corresponds to the optimization of the transport plan, and the metric calculations are excluded.

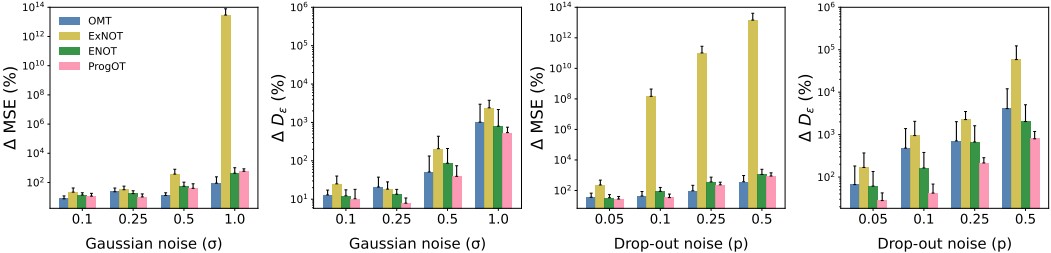

Figure 3: Stability of OT solvers under noise in the W2 benchmark task. The performance of OMT together with baseline methods is evaluated under two types of perturbations applied to the source data. The two left panels illustrate the effect of additive Gaussian noise with increasing standard deviations, while the two right panels show the effect of dropout noise. Reported values indicate performance changes relative to the noise-free case, evaluated on the test set with 10,000 samples and averaged over five random initializations.

strong overall performance while using substantially less resources, as reflected by shorter runtimes and smaller memory usage.

To investigate the stability of OMT under noise, we conduct an ablation study: noise is added to the source data during training, while the original clean data is used for evaluation. We consider two types of perturbations: white noise, controlled by $\sigma$ and dropout noise with probability $p$. Figure 3 demonstrates that OMT produces the most robust OT plans under both noise models, considering the relative change in MSE in response to input perturbations. Overall, OMT consistently delivers strong performance across metrics, often matching or exceeding existing baselines, highlighting both the robustness and competitiveness of our approach. Additional details are provided in Appendix B.2.

Table 1: Average $D_\varepsilon \downarrow$ values for forward and backward OMT maps compared to PROGOT, GMM-OT, EOT on the human scRNA-seq dataset (Srivatsan et al., 2020). The results are reported as mean±std. dev. ($d_{\mathrm{PCA}} = 16$, 5 random seeds). Additional results for other dimensions and computational costs are in Appendix C.

|  | Belinostat | Dacinostat | Givinostat | Quisinostat | Hesperadin |
|---|---|---|---|---|---|
| EOT | $17.4 \pm 0.01$ | $18.4 \pm 0.01$ | $17.5 \pm 0.01$ | $17.6 \pm 0.03$ | $17.5 \pm 0.01$ |
| PROGOT | $8.43 \pm 0.01$ | $8.84 \pm 0.03$ | $8.82 \pm 0.04$ | $9.52 \pm 0.01$ | $8.06 \pm 0.01$ |
| GMM-OT | $7.99 \pm 0.02$ | $8.89 \pm 0.17$ | $8.89 \pm 0.06$ | $8.75 \pm 0.23$ | $8.00 \pm 0.03$ |
| OMT | $\mathbf{7.91 \pm 0.06}$ | $\mathbf{8.12 \pm 0.04}$ | $\mathbf{8.75 \pm 0.14}$ | $\mathbf{8.72 \pm 0.10}$ | $8.00 \pm 0.03$ |

**Single-cell Data.** OT has emerged as a powerful tool in computational biology, with applications such as aligning cell populations across conditions and inferring their trajectories over time (Tong et al., 2020; Bunne et al., 2023; 2024). Here, we focus on two single-cell modalities, (i) single-cell RNA sequencing (scRNA-seq) and (ii) single-cell transcriptomic profiling with multiplexed error-robust fluorescence *in situ* hybridization (MERFISH). scRNA-seq generates high-dimensional molecular profiles by measuring the expression of thousands of genes at single-cell resolution, whereas MERFISH provides spatially resolved transcriptomic profiles, capturing the organization of cells within tissue. We consider three scRNA-seq datasets: one human dataset, sci-Plex (Srivatsan et al., 2020), and two 10x Genomics mouse brain datasets, one collected during development (Gao et al., 2025) and the other during aging (Jin et al., 2025). Additionally, we include the Vizgen MERFISH Mouse Brain Receptor Map dataset. The sci-Plex data serves as a common benchmark for assessing OT performance on real-world biological data (Cuturi et al., 2023; Janati et al., 2020). Consistent with previous work, our analysis focuses on a subset of this dataset comprising three cell lines (A549, K562, and MCF7) exposed to five different cancer treatments for 24 hours. Following the preprocessing steps recommended in Cuturi et al. (2023), the final dataset contains $77,920$ cells and $34,636$ genes. Similarly, for the transport analysis, we perform dimensionality reduction using PCA, retaining the same number of PCs as in Kassraie et al. (2024). Table 1 summarizes the results for OMT against EOT and PROGOT, which is the top-performing baseline. The results indicate that our OMT model outperforms PROGOT across all treatment conditions. For this comparison, PROGOT is configured with the recommended scheduling parameters from its original publication, with $K = 4$.

We note that, here, the data subset for each task is relatively small ($\sim 10^4$ cells). While this scale is computationally feasible for sample-based approaches like PROGOT, it does not represent the large-scale datasets in modern single-cell studies. We next extend our study to a larger single-cell dataset. To benchmark our OMT approach against the HiRef algorithm, we analyze MERFISH data from the Vizgen mouse brain dataset. We select the same two slices studied in Halmos et al. (2025), containing $85,958$ and $84,172$ cells, respectively. The alignment is formulated as a transport problem using only the spatial coordinates of the cells. Using the computed transport map, we impute gene expression in aligned cells based on the source slice data (Figures 4, 11, 12). Table 2 benchmarks OMT against mini-batch (MB; batch size 1024), low-rank OTs, and GMM-OT. We evaluate performance across 5 marker genes based on cosine similarity and transport cost. The former measures the agreement between imputed gene expression at the transported locations and the ground-truth expression of spatially adjacent cells in the target slice. OMT consistently outperforms all baselines in both metrics. See Appendices C.4 and C.3 for additional details and assessments.

For further validation, we extend our analysis beyond small-scale data, we apply OMT to larger scRNA-seq datasets from the mouse brain, encompassing the entire lifespan from development to aging. For brain development, we use data from the visual cortex spanning a wide period from embryonic days to postnatal days (E11.5-P28) (Gao et al., 2025). For aging, we consider data from Jin et al. (2025) collected from 108 mice, span six brain regions at two timepoints: adult (P53–69) and aged (P540–553). Our analysis focuses on the cellular dynamics of the oligodendrocyte lineage, including oligodendrocyte precursor cells (OPCs) and mature oligodendrocytes (Oligos). These glial cells, which are responsible for myelinating axons to facilitate neural communication, exhibit significant heterogeneity in their lifespan and function, making them a suitable candidate for studying time-dependent cellular transitions (Marques et al., 2016; Jin et al., 2025).

Table 2: Comparison of OMT against mini-batch, low-rank OT, and GMM-OT solvers on MERFISH mouse brain data. Cosine similarity ($\uparrow$) and spatial Euclidean costs ($\downarrow$) are reported. Cosine similarity measures the agreement between imputed and original expression profiles in the target slice for the genes used in Halmos et al. (2025) for evaluation, while the transport map is optimized only with respect to spatial coordinates. Both GMM-OT and OMT utilize 1000 components to fit the source and target distributions.

|  | Slc17a7 ($\uparrow$) | Grm4 ($\uparrow$) | Olig1 ($\uparrow$) | Gad1 ($\uparrow$) | Peg10 ($\uparrow$) | Cost ($\downarrow$) |
|---|---|---|---|---|---|---|
| MB (1024) | 0.73 | 0.76 | 0.69 | 0.47 | 0.56 | 384.25 |
| LOT | 0.32 | 0.23 | 0.30 | 0.17 | 0.07 | 3722.32 |
| FRLC | 0.22 | 0.21 | 0.19 | 0.10 | 0.10 | 415.07 |
| HiRef | 0.81 | 0.80 | 0.75 | 0.49 | 0.60 | 330.33 |
| GMM-OT | 0.89 | 0.90 | 0.90 | 0.73 | 0.73 | 107.20 |
| OMT | **0.90** | **0.93** | **0.90** | **0.78** | **0.75** | **101.69** |



Figure 4: Cellular alignment across MER-FISH mouse brain data using OMT on spatial coordinates. From left to right, the expression profiles of the marker gene *Slc17a7* are illustrated in the source, target, and transported spatially distributed cells within the brain tissue.

After preprocessing (Appendix C), the data includes $32,998$ cells and $9,900$ highly variable genes (HVGs) from the developmental data, alongside $253,468$ cells and $9,359$ HVGs from the ageing dataset. We utilized a VAE model to learn a compressed representation of the cells. The OMT model was then trained on these low-dimensional embeddings ($d_z = 10$). OMT is applied across 11 consecutive time pairs between E11.5 and P28 for the developmental data, and between adult and aged time points for ageing data. Figure 5 summarizes the analysis of the mouse datasets. The UMAP plots show that the cell population transported by the model, whether forward or backward in time, closely mirrors the empirical cell distribution at the target timepoints. This demonstrates the model's ability to learn the global distribution across cell subclasses. The right panels of the figure illustrate the clear developmental and aging trajectories revealed by our OMT model. The transport map reveals the known developmental pathway, beginning with neuroepithelial cells (NECs) that mature into radial glia (RG). These cells subsequently differentiate into glioblasts (Gliob), which are the common progenitors for both the astrocyte (Astro) and the OPC-Oligo lineages. This temporal progression is visually represented by a color gradient, transitioning from yellow (E11.5) to dark red (P28). Focusing on the OPC-Oligo lineage, the rightmost column provides a detailed view of this population during development (top) and aging (bottom). It highlights the specific cellular maturation sequence from oligodendrocyte precursor cells (OPCs) to committed oligodendrocyte precursors (COPs), newly formed oligodendrocytes (NFOLs), myelin-forming oligodendrocytes (MFOLs), and finally, mature oligodendrocytes (MOLs). See Appendix C.4 for additional evaluation in gene space.

**Image Datasets.** To further demonstrate the applicability of the the proposed OMT framework beyond tabular data, we apply it to an unpaired image-to-image translation task using two benchmark datasets: MNIST (LeCun, 1998) and CIFAR-10 (Krizhevsky et al., 2009). In MNIST, the task involves translating images of one digit into another (e.g., learning transport maps such as $T : 1 \rightarrow 7$). Similarly, in CIFAR-10, the goal is to translate images from one semantic class (e.g., airplane) into another (e.g., bird). Although OMT can in principle be applied directly to raw image data, the resulting mappings are not semantically meaningful and fail to capture class-level translations.

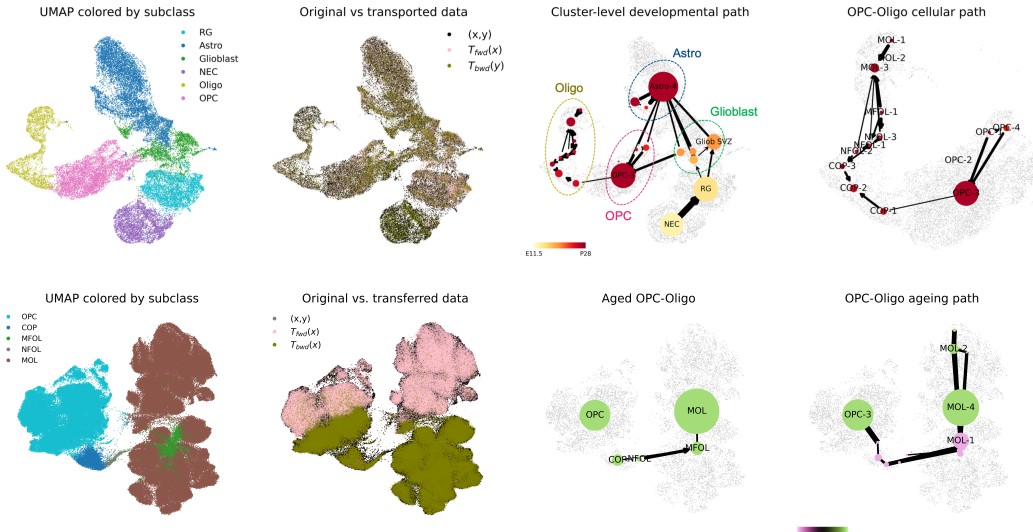

Figure 5: OPC–Oligo trajectories across the mouse lifespan. Top row: developmental dataset from the mouse visual cortex. From left to right: (1) UMAP projection showing distinct neural cell subclasses. (2) The alignment between the original measured data and the transferred data using OMT. (3) The inferred global developmental trajectory at the cluster level, tracing paths from early progenitors like neuroepithelial cells (NEC) and radial glia (RG). (4) The specific cellular pathway detailing the differentiation from OPC to oligodendrocytes. Bottom row: mouse aging dataset. From left to right: (1) UMAP projection of cell subtypes within the oligodendrocyte lineage. (2) The alignment of original and transferred data distributions. (3) The network graph illustrating the stages of the myelination cycle in aged mice. (4) The inferred aging pathway.

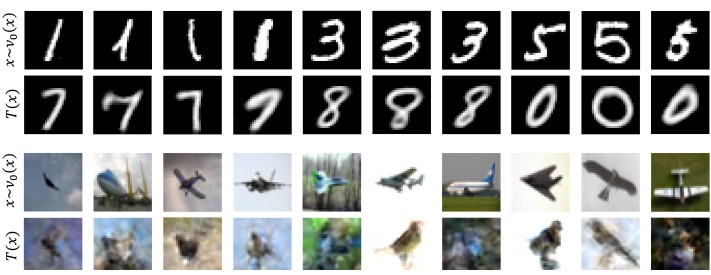

Figure 6: Performance of OMT for unpaired image-to-image translation on the MNIST and CIFAR-10 datasets. For each dataset, the top row shows original samples from the source distribution, $x \sim \nu_0$, and the bottom row shows the corresponding transported images $T_{\mathrm{OMT}}^{\nu_0 \to \nu_1}$.

To enable this, we first train an autoencoder on the entire dataset, covering all classes, to obtain compact and semantically meaningful low-dimensional embeddings. Within this latent space, the OMT is then applied to learn optimal transport maps across different classes. Figures 6 and 17 illustrate representative examples of these class-to-class translations for test images in MNIST and CIFAR-10, respectively. Quantitative evaluation of the generated translations is reported in Table 3 using the widely adopted Fréchet Inception Distance (FID). These experiments highlight that, OMT can be effectively extended to image-based applications as well. For context and to benchmark our performance against established OT based approaches, we also report the FID scores for WGAN (Arjovsky et al., 2017) and WGAN-GP (Gulrajani et al., 2017). The results show that OMT performs in a similar range to WGAN on CIFAR-10, while outperforming both WGAN and WGAN-GP on the MNIST dataset. See Appendix D for further implementation details, including the autoencoder architectures used for dimensionality reduction and the hyperparameters for OMT.

|  | MNIST | CIFAR-10 |
|---|---|---|
| WGAN | $6.7 \pm 0.4$ | 55.2 |
| WGAN-GP | $7.4 \pm 0.3$ | 39.4 |
| OMT | $1.2 \pm 0.1$ | $56.1 \pm 1.5$ |

Table 3: FID ($\downarrow$) values for unpaired image translation on the MNIST (grayscale) and CIFAR-10 (color) datasets. Reported values for WGAN and WGAN-GP are taken from previous studies (Choi et al., 2023; Rout et al., 2022; Qian et al., 2021). Results for OMT are computed over 10 random initializations.

To demonstrate the applicability of OMT to large-scale, high-dimensional data, we perform an alignment task on ImageNet (Deng et al., 2009), following Halmos et al. (2025). We use 2048-d embeddings from a pretrained ResNet-50 (He et al., 2016) and construct source and target subsets via random sampling. OMT outperforms both MB and LOT solvers that can be optimized in this setting, achieving the lowest Euclidean distance between aligned images (Table 4). OMT is trained with 1000 components for each of the source and target. See Appendix D for implementation details.

| Method | MB (1024) | FRLC | HiRef | OMT |
|---|---|---|---|---|
| Cost ($\downarrow$) | 19.58 | 24.12 | 18.97 | **14.30** |

Table 4: Cost values for the ImageNet alignment task (Deng et al., 2009). LOT was not evaluated due to memory limitations (Halmos et al., 2025).

## 6 CONCLUSION

In this work, we introduced OMT, a new family of EOT solvers that enhance performance by moving beyond sample-to-sample transportation toward subpopulation-level transportation, leveraging mixture-model representations and the closed-form structure of Gaussian families. OMT achieves computational efficiency through a strictly biconvex formulation which, when embedded in a global optimization framework, ensures that each subproblem converges in a single step to a unique solution, thereby providing a stable and reliable estimator of the OT plan. Empirically, we showed OMT matches or exceeds the performance of state-of-the-art non-neural OT solvers, while remaining competitive with neural approaches, but with substantially lower computational and memory requirements. One promising direction for future work is to extend OMT to the unbalanced OT setting, particularly for mixtures with unequal component masses, which is highly relevant in applications such as single-cell RNA-seq. The current OMT learns static transport plans between pairs of time points. In future work, it could be extended to learn dynamic transport maps between Gaussians, building on the approach of Bunne et al. (2023). A current limitation, however, lies in applying OMT to high-resolution image generation tasks, where straightforward extensions are not yet practical. Another interesting avenue would be to explore a neural extension of OMT, enabling scalable applications in such high-dimensional domains.

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

APPENDIX

## A  PROOFS

**Lemma 1.** *For any $\varepsilon_1, \varepsilon_2 > 0$, $\mathcal{L}_{\varepsilon_1, \varepsilon_2}(\Omega, P)$ is strictly biconvex.*

*Proof.* A function $f : \mathcal{X} \times \mathcal{Y} \to \mathbb{R}$ is called *biconvex* if, for fixed $x \in \mathcal{X}$, the function $f(x, y)$ is convex in $y$, and for fixed $y \in \mathcal{Y}$, it is convex in $x$. According to **Theorem 3.1** Gorski et al. (2007), $f(x, y)$ is biconvex if and only if for all $(x_1, y_1), (x_1, y_2), (x_2, y_1), (x_2, y_2) \in \mathcal{X} \times \mathcal{Y}$ and all $\lambda, \tau \in [0, 1]$, the following inequality holds:

$$f(x_\lambda, y_\tau) \leq \lambda\tau f(x_1, y_1) + (1 - \lambda)\tau f(x_2, y_1) + \lambda(1 - \tau)f(x_1, y_2) + (1 - \lambda)(1 - \tau)f(x_2, y_2),$$

where $(x_\lambda, y_\tau) := (\lambda x_1 + (1 - \lambda)x_2, \ \tau y_1 + (1 - \tau)y_2)$.

Then, for given $\omega_{ij_\lambda} = \lambda\tilde{\omega}_{ij} + (1 - \lambda)\tilde{\tilde{\omega}}_{ij}$ and $dp_{ij_\tau} = \tau\tilde{dp}_{ij} + (1 - \tau)\tilde{\tilde{dp}}_{ij}$, the following inequality must hold.

$$\mathcal{L}_{\varepsilon_1, \varepsilon_2}(\Omega_\lambda, P_\tau) < \lambda\tau\mathcal{L}_{\varepsilon_1, \varepsilon_2}(\tilde{\Omega}, \tilde{P}) + (1 - \lambda)\tau\mathcal{L}_{\varepsilon_1, \varepsilon_2}(\tilde{\tilde{\Omega}}, \tilde{P}) + \lambda(1 - \tau)\mathcal{L}_{\varepsilon_1, \varepsilon_2}(\tilde{\Omega}, \tilde{\tilde{P}}) + (1 - \lambda)(1 - \tau)\mathcal{L}_{\varepsilon_1, \varepsilon_2}(\tilde{\tilde{\omega}}, \tilde{\tilde{P}}) \tag{14}$$

$$\begin{aligned}
\mathcal{L}_{\varepsilon_1, \varepsilon_2}(\Omega_\lambda, P_\tau) &= \sum_{i,j}^{K} \left(\lambda\tilde{\omega}_{ij} + (1 - \lambda)\tilde{\tilde{\omega}}_{ij}\right) \left[\int_{\mathcal{X} \times \mathcal{Y}} \|\mathbf{x} - \mathbf{y}\|_2^2 \left(\tau\tilde{dp}_{ij}(\mathbf{x}, \mathbf{y}) + (1 - \tau)\tilde{\tilde{dp}}_{ij}(\mathbf{x}, \mathbf{y})\right)\right] + \\
&\quad \varepsilon_2 D_{KL}(\Omega_\lambda \| \boldsymbol{\alpha}_0 \otimes \boldsymbol{\alpha}_1) + \sum_{i,j}^{K} \left(\lambda\tilde{\omega}_{ij} + (1 - \lambda)\tilde{\tilde{\omega}}_{ij}\right) \varepsilon_1 D_{KL}(p_{ij_\tau} \| \mu_{0_i} \otimes \mu_{1_j}) \\
&= \underbrace{\lambda\tau \sum_{i,j} \tilde{\omega}_{ij} \int \|\mathbf{x} - \mathbf{y}\|_2^2 \tilde{dp}_{ij}(\mathbf{x}, \mathbf{y})}_{A} + \underbrace{(1 - \lambda)\tau \sum_{i,j} \tilde{\tilde{\omega}}_{ij} \int \|\mathbf{x} - \mathbf{y}\|_2^2 \tilde{dp}_{ij}(\mathbf{x}, \mathbf{y})}_{B} + \\
&\quad \underbrace{\lambda(1 - \tau) \sum_{i,j} \tilde{\omega}_{ij} \int \|\mathbf{x} - \mathbf{y}\|_2^2 \tilde{\tilde{dp}}_{ij}(\mathbf{x}, \mathbf{y})}_{C} + \underbrace{(1 - \lambda)(1 - \tau) \sum_{i,j} \tilde{\tilde{\omega}}_{ij} \int \|\mathbf{x} - \mathbf{y}\|_2^2 \tilde{\tilde{dp}}_{ij}(\mathbf{x}, \mathbf{y})}_{D} + \\
&\quad \varepsilon_1 \sum_{i,j} \left(\lambda\tilde{\omega}_{ij} + (1 - \lambda)\tilde{\tilde{\omega}}_{ij}\right) \underbrace{D_{KL}(p_{ij_\tau} \| \mu_{0_i} \otimes \mu_{1_j})}_{E} + \varepsilon_2 \underbrace{D_{KL}(\Omega_\lambda \| \boldsymbol{\alpha}_0 \otimes \boldsymbol{\alpha}_1)}_{F}. \tag{15}
\end{aligned}$$

$$\begin{aligned}
\lambda\tau\mathcal{L}_{\varepsilon_1, \varepsilon_2}(\tilde{\Omega}, \tilde{P}) &+ (1 - \lambda)\tau\mathcal{L}_{\varepsilon_1, \varepsilon_2}(\tilde{\tilde{\Omega}}, \tilde{P}) + \lambda(1 - \tau)\mathcal{L}_{\varepsilon_1, \varepsilon_2}(\tilde{\Omega}, \tilde{\tilde{P}}) + (1 - \lambda)(1 - \tau)\mathcal{L}_{\varepsilon_1, \varepsilon_2}(\tilde{\tilde{\omega}}, \tilde{\tilde{P}}) = \\
&\underbrace{\lambda\tau \sum_{i,j} \tilde{\omega}_{ij} \int \|\mathbf{x} - \mathbf{y}\|_2^2 \tilde{dp}_{ij}(\mathbf{x}, \mathbf{y})}_{A} + \varepsilon_1 \sum_{i,j} \lambda\tilde{\omega}_{ij} \left(\tau D_{KL}(\tilde{p}_{ij_\tau} \| \mu_{0_i} \otimes \mu_{1_j})\right) + \\
&\varepsilon_2\tau\lambda D_{KL}(\tilde{\Omega} \| \boldsymbol{\alpha}_0 \otimes \boldsymbol{\alpha}_1) + \underbrace{(1 - \lambda)\tau \sum_{i,j} \tilde{\tilde{\omega}}_{ij} \int \|\mathbf{x} - \mathbf{y}\|_2^2 \tilde{dp}_{ij}(\mathbf{x}, \mathbf{y})}_{B} + \\
&\varepsilon_1 \sum_{i,j} (1 - \lambda)\tilde{\tilde{\omega}}_{ij} \left(\tau D_{KL}(\tilde{p}_{ij_\tau} \| \mu_{0_i} \otimes \mu_{1_j})\right) + \varepsilon_2\tau(1 - \lambda) D_{KL}(\tilde{\tilde{\Omega}} \| \boldsymbol{\alpha}_0 \otimes \boldsymbol{\alpha}_1) + \\
&\underbrace{\lambda(1 - \tau) \sum_{i,j} \tilde{\omega}_{ij} \int \|\mathbf{x} - \mathbf{y}\|_2^2 \tilde{\tilde{dp}}_{ij}(\mathbf{x}, \mathbf{y})}_{C} +
\end{aligned}$$

$$\varepsilon_1 \sum_{i,j} \lambda \tilde{\omega}_{ij} \left((1-\tau) D_{KL}(\tilde{\bar{p}}_{ij_\tau} \| \mu_{0_i} \otimes \mu_{1_j})\right) + \varepsilon_2 (1-\tau) \lambda D_{KL}(\tilde{\Omega} \| \boldsymbol{\alpha}_0 \otimes \boldsymbol{\alpha}_1) +$$

$$\underbrace{(1-\lambda)(1-\tau) \sum_{i,j} \tilde{\bar{\omega}}_{ij} \int \|\mathbf{x} - \mathbf{y}\|_2^2 d\tilde{\bar{p}}_{ij}(\mathbf{x}, \mathbf{y})}_{D} + \varepsilon_1 \sum_{i,j} (1-\lambda) \tilde{\bar{\omega}}_{ij} \left((1-\tau) D_{KL}(\tilde{\bar{p}}_{ij} \| \mu_{0_i} \otimes \mu_{1_j})\right) +$$

$$\varepsilon_2 (1-\tau)(1-\lambda) D_{KL}(\tilde{\bar{\Omega}} \| \boldsymbol{\alpha}_0 \otimes \boldsymbol{\alpha}_1),$$

$$= A + B + C + D + \varepsilon_2 \left(\underbrace{\lambda D_{KL}(\tilde{\Omega} \| \boldsymbol{\alpha}_0 \otimes \boldsymbol{\alpha}_1) + (1-\lambda) D_{KL}(\tilde{\bar{\Omega}} \| \boldsymbol{\alpha}_0 \otimes \boldsymbol{\alpha}_1)}_{G}\right) +$$

$$\varepsilon_1 \sum_{i,j} \left(\lambda \tilde{\omega}_{ij} + (1-\lambda) \tilde{\bar{\omega}}_{ij}\right) \left(\underbrace{\tau D_{KL}(\tilde{p}_{ij} \| \mu_{0_i} \otimes \mu_{1_j}) + (1-\tau) D_{KL}(\tilde{\bar{p}}_{ij} \| \mu_{0_i} \otimes \mu_{1_j})}_{H}\right). \tag{16}$$

For any fixed $q$, $D_{KL}(p\|q)$ is strictly convex in $p$. Consequently, we have $E < H$ and $F < G$, which together imply that inequality (14) holds. $\qquad\square$

**Theorem 1.** *For the optimization problem defined in* (4.1)*, the GOP algorithm converges to a unique solution in a single iteration.*

*Proof.* Consider the biconvex optimization problem defined as follows:

$$\min \left\{\mathcal{L}_{\varepsilon_1 \varepsilon_2}(\Omega, P), \ (\Omega, P) \in \Lambda\right\}.$$

Let's begin by selecting an arbitrary initial point $Z_0 = (\Omega_0, P_0) \in \Lambda$ and set the iteration index $s = 0$. Without loss of generality, we assume that $\omega_{ij}^0 > 0$, for all $i, j$. We then solve the following convex optimization problem with respect to $P$, keeping $\Omega_s$ fixed.

$$\min_P \sum_{i,j} \omega_{ij}^0 \left[\int_{\mathcal{X} \times \mathcal{Y}} \|\mathbf{x} - \mathbf{y}\|_2^2 \, dp_{ij}(\mathbf{x}, \mathbf{y}) + \varepsilon_1 D_{KL}(p_{ij} \| \mu_{0_i} \otimes \mu_{1_j})\right]$$

$$\text{s.t.} \quad \int_{\mathcal{Y}} dP(\mathbf{x}, \mathbf{y}) = \boldsymbol{\mu}_0, \quad \int_{\mathcal{X}} dP(\mathbf{x}, \mathbf{y}) = \boldsymbol{\mu}_1 \tag{17}$$

Since the objective function in (A) is convex in $P$ for any fixed $\Omega$ and Slater's condition is satisfied Floudas & Visweswaran (1993), strong duality holds. Accordingly, problem (A) admits the following strong dual formulation:

$$\max_{\Phi, \Psi} \min_P \sum_{i,j} \omega_{ij}^0 \left[\int_{\mathcal{X} \times \mathcal{Y}} \|\mathbf{x} - \mathbf{y}\|_2^2 \, dp_{ij}(\mathbf{x}, \mathbf{y}) + \varepsilon_1 D_{KL}(p_{ij} \| \mu_{0_i} \otimes \mu_{1_j})\right] - \tag{18}$$

$$\sum_{i,j} \omega_{ij}^0 \left[\int_{\mathcal{X}} \varphi_{ij}(\mathbf{x}) \left(\int_{\mathcal{Y}} dp_{ij}(\mathbf{x}, \mathbf{y}) - d\mu_{0_i}(\mathbf{x})\right) + \int_{\mathcal{Y}} \psi_{ij}(\mathbf{y}) \left(\int_{\mathcal{X}} dp_{ij}(\mathbf{x}, \mathbf{y}) - d\mu_{1_j}(\mathbf{y})\right)\right]. \tag{19}$$

Here, $\phi_{ij}, \psi_{ij} \geq 0$ are the Lagrange multipliers associated with the marginal constraints.

Denoting the inner minimization problem in (18) by $\min_P f(P, \Phi, \Psi)$, we seek the optimal policy that minimizes the loss in (18). To do so, we compute the functional derivative of the loss with respect to $dp_{ij}(\mathbf{x}, \mathbf{y})$.

$$\frac{df}{dp_{ij}(\mathbf{x}, \mathbf{y})} = \omega_{ij}^0 \left(\|\mathbf{x} - \mathbf{y}\|_2^2 + \varepsilon_1 \log \frac{dp_{ij}(\mathbf{x}, \mathbf{y})}{d\mu_{0_i}(\mathbf{x}) d\mu_{1_j}(\mathbf{y})} - \varphi_{ij}(\mathbf{x}) - \psi_{ij}(\mathbf{y})\right). \tag{20}$$

Since the objective is strictly convex in $P$, it admits a unique solution that is independent of $\Omega$.

$$dp_{ij}^*(\mathbf{x}, \mathbf{y}) = \exp\left(\frac{\varphi_{ij}(\mathbf{x}) + \psi_{ij}(\mathbf{y}) - \|\mathbf{x} - \mathbf{y}\|_2^2}{\varepsilon_1}\right) d\mu_{0_i}(\mathbf{x}) d\mu_{1_j}(\mathbf{y}). \tag{21}$$

Substituting $dp_{ij}^*$ in (18), the dual form can be written as:

$$\max_{\Phi, \Psi} \sum_{i,j} \omega_{ij}^0 \left[ \int_{\mathcal{X}} \varphi_{ij}(\mathbf{x}) d\mu_{0_i}(\mathbf{x}) - \int_{\mathcal{Y}} \psi_{ij}(\mathbf{y}) d\mu_{0_j}(\mathbf{y}) - \varepsilon_1 \left( \int_{\mathcal{X} \times \mathcal{Y}} dp_{i,j}^*(\mathbf{x}, \mathbf{y}) - 1 \right) \right]. \quad (22)$$

To determine the optimal Lagrange multipliers that maximize the dual objective, we differentiate the loss function in (22), denoted as $g(\Phi, \Psi, P^*)$ with respect to each multiplier as follows.

$$\frac{dg}{\varphi_{ij}(\mathbf{x})} = \omega_{ij}^0 \left( d\mu_{0_i}(\mathbf{x}) - \int_{\mathcal{Y}} \exp\left(\frac{\varphi_{ij}(\mathbf{x}) + \psi_{ij}(\mathbf{y}) - \|\mathbf{x} - \mathbf{y}\|_2^2}{\varepsilon_1}\right) d\mu_{0_i}(\mathbf{x}) d\mu_{1_j}(\mathbf{y}) \right) \quad (23)$$

$$\frac{dg}{\psi_{ij}(\mathbf{x})} = \omega_{ij}^0 \left( d\mu_{1_j}(\mathbf{x}) - \int_{\mathcal{X}} \exp\left(\frac{\varphi_{ij}(\mathbf{x}) + \psi_{ij}(\mathbf{y}) - \|\mathbf{x} - \mathbf{y}\|_2^2}{\varepsilon_1}\right) d\mu_{0_i}(\mathbf{x}) d\mu_{1_j}(\mathbf{y}) \right) \quad (24)$$

Assuming that, for all $i, j$, the measures $\mu_{0_i}$ and $\mu_{1_j}$ have finite second-order moments, the pair $(\varphi_{ij}, \psi_{ij})$ is optimal if and only if the following conditions are satisfied.

$$\int_{\mathcal{Y}} \exp\left(\frac{\varphi_{ij}(\mathbf{x}) + \psi_{ij}(\mathbf{y}) - \|\mathbf{x} - \mathbf{y}\|_2^2}{\varepsilon_1}\right) d\mu_{1_j}(\mathbf{y}) = 1, \qquad \int_{\mathcal{X}} \exp\left(\frac{\varphi_{ij}(\mathbf{x}) + \psi_{ij}(\mathbf{y}) - \|\mathbf{x} - \mathbf{y}\|_2^2}{\varepsilon_1}\right) d\mu_{0_i}(\mathbf{x}) = 1,$$
$$(25)$$

which is equivalent to the following expressions for the optimal multipliers:

$$\varphi_{ij}(\mathbf{x}) = -\varepsilon_1 \log \int_{\mathcal{Y}} \exp\left(\frac{\psi_{ij}(\mathbf{y}) - \|\mathbf{x} - \mathbf{y}\|_2^2}{\varepsilon_1}\right) d\mu_{1_j}(\mathbf{y}), \quad (26)$$

$$\psi_{ij}(\mathbf{y}) = -\varepsilon_1 \log \int_{\mathcal{X}} \exp\left(\frac{\varphi_{ij}(\mathbf{x}) - \|\mathbf{x} - \mathbf{y}\|_2^2}{\varepsilon_1}\right) d\mu_{0_i}(\mathbf{x}). \quad (27)$$

As observed, the optimal Lagrange multipliers are also independent of $\Omega$. Therefore, the unique optimal solution of (18) remains the same for any fixed $\Omega_s$, implying

$$\forall s, \quad \mathcal{L}_{\varepsilon_1 \varepsilon_2}(\Omega_s, P^*) \leq \mathcal{L}_{\varepsilon_1 \varepsilon_2}(\Omega_s, P),$$

$$\forall s \neq s', \quad \mathcal{L}_{\varepsilon_1 \varepsilon_2}(\Omega_s, P^*) = \mathcal{L}_{\varepsilon_1 \varepsilon_2}(\Omega_{s'}, P^*),$$

$$\forall s \neq s', \quad \mathcal{L}_{\varepsilon_1 \varepsilon_2}(\Omega_s, P^*) = \mathcal{L}_{\varepsilon_1 \varepsilon_2}(\Omega_{s'}, P_{s'}), \text{ iff } P_{s'} = P^* \quad (28)$$

Proceeding to the next step, we set $s = 1$, which gives $P_1 = P^*$. For a given $\varepsilon_2 > 0$, we solve the following strictly convex optimization problem with respect to $\Omega$, keeping $P$ fixed.

$$\min_{\Omega} \sum_{i,j} \omega_{ij} \left[ \int_{\mathcal{X} \times \mathcal{Y}} \|\mathbf{x} - \mathbf{y}\|_2^2 \, dp_{ij}^*(\mathbf{x}, \mathbf{y}) + \varepsilon_1 D_{KL}(p_{ij}^* \| \mu_{0_i} \otimes \mu_{1_j}) \right] + \varepsilon_2 D_{KL}(\Omega \| \boldsymbol{\alpha}_0 \otimes \boldsymbol{\alpha}_1)$$
$$(29)$$

$$\text{s.t.} \quad \mathbf{1}\Omega = \boldsymbol{\alpha}_0, \qquad \Omega^T \mathbf{1} = \boldsymbol{\alpha}_1 \quad (30)$$

This formulation, similar to equation (A), admits the following dual form:

$$\max_{\boldsymbol{\lambda}, \boldsymbol{\tau}} \min_{\Omega} \sum_{i,j} \omega_{ij} \left[ \int_{\mathcal{X} \times \mathcal{Y}} \|\mathbf{x} - \mathbf{y}\|_2^2 \, dp_{ij}^*(\mathbf{x}, \mathbf{y}) + \varepsilon_1 D_{KL}(p_{ij}^* \| \mu_{0_i} \otimes \mu_{1_j}) \right] + \varepsilon_2 D_{KL}(\Omega \| \boldsymbol{\alpha}_0 \otimes \boldsymbol{\alpha}_1) -$$
$$(31)$$

$$\sum_i \lambda_i \left( \sum_j \omega_{ij} - \alpha_{0_i} \right) - \sum_j \tau_j \left( \sum_i \omega_{ij} - \alpha_{1_j} \right). \quad (32)$$

The inner minimization in (31), denoted $f'(\Omega, \boldsymbol{\lambda}, \boldsymbol{\tau})$, admits a unique solution for the optimal weights. These weights can be derived by computing the functional derivative of the objective with respect to $\omega_{ij}$, yielding:

$$\frac{df'}{\omega_{ij}} = \underbrace{\int_{\mathcal{X} \times \mathcal{Y}} \|\mathbf{x} - \mathbf{y}\|_2^2 \, dp_{ij}^*(\mathbf{x}, \mathbf{y}) + \varepsilon_1 D_{KL}(p_{ij}^* \| \mu_{0_i} \otimes \mu_{1_j})}_{\mathcal{L}_{p_{ij}^*}} + \varepsilon_2 \log \frac{\omega_{ij}}{\alpha_{0_i} \alpha_{1_j}} - \lambda_i - \tau_j \quad (33)$$

$$\omega_{ij}^* = \exp\left(\frac{\lambda_i + \tau_j - \mathcal{L}_{p_{ij}^*}}{\varepsilon_2}\right) \alpha_{0_i} \alpha_{1_j} \quad (34)$$

To obtain the optimal Lagrangian multipliers that maximize the loss in (31), denoted $\max_{\boldsymbol{\lambda}, \boldsymbol{\tau}} g'(\boldsymbol{\lambda}, \boldsymbol{\tau}, \Omega^*)$, we compute the partial derivatives of $g'$ with respect to each multiplier.

$$\frac{dg'}{\lambda_i} = \alpha_{0_i} - \sum_j \exp\left(\frac{\lambda_i + \tau_j - \mathcal{L}_{p^*_{ij}}}{\varepsilon_2}\right)\alpha_{0_i}\alpha_{1_j} \tag{35}$$

$$\frac{dg'}{\tau_j} = \alpha_{1_j} - \sum_i \exp\left(\frac{\lambda_i + \tau_j - \mathcal{L}_{p^*_{ij}}}{\varepsilon_2}\right)\alpha_{0_i}\alpha_{1_j} \tag{36}$$

Solving these yields the optimal multipliers as:

$$\lambda_i = -\varepsilon_2 \log \sum_j \exp\left(\frac{\tau_j - \mathcal{L}_{p^*_{ij}}}{\varepsilon_2}\right)\alpha_{1_j}, \qquad \tau_j = -\varepsilon_2 \log \sum_i \exp\left(\frac{\lambda_i - \mathcal{L}_{p^*_{ij}}}{\varepsilon_2}\right)\alpha_{0_i} \tag{37}$$

Since the optimal weight in (34) minimizes $\mathcal{L}_{\varepsilon_1 \varepsilon_2}(\Omega, P_1)$ uniquely for the fixed choice $P_1 = P^*$, it follows that:

$$\mathcal{L}_{\varepsilon_1 \varepsilon_2}(\Omega^*, P_1) \leq \mathcal{L}_{\varepsilon_1 \varepsilon_2}(\Omega_0, P_1) \leq \mathcal{L}_{\varepsilon_1 \varepsilon_2}(\Omega_0, P_0) \tag{38}$$

Now, lets update the optimization by setting $\Omega_1 = \Omega^*$, and advancing to step $s = 2$. According to (28), the next update satisfies:

$$\mathcal{L}_{\varepsilon_1 \varepsilon_2}(\Omega_1, P^*) = \min_P \mathcal{L}_{\varepsilon_1 \varepsilon_2}(\Omega_1, P). \tag{39}$$

Since we find that $P_2 = P_1 = P^*$ and consequently, $\Omega_2 = \Omega_1 = \Omega^*$, the stopping criterion of the overall alternating optimization procedure is met after just a single iteration. $\square$

**Corollary 1.** *Let $\nu_0 \in G_{K_0}(\mathbb{R}^d)$ and $\nu_1 \in G_{K_1}(\mathbb{R}^d)$ be two Gaussian mixture models (GMMs) in $\mathbb{R}^d$ with $K_0$ and $K_1$ components, respectively. Then, the optimal mixture transport map between $\nu_0$ and $\nu_1$ is itself a Gaussian mixture model with $K$ components, where $K \leq K_0 K_1$.*

*Proof.* According to **Theorem** 1, the optimal mixture transport policy between each pair $\mu_0^i(\mathbf{x}), \mu_1^j(\mathbf{y})$, is independent of the weight variable and is given by:

$$dp^*_{ij}(\mathbf{x}, \mathbf{y}) = \exp\left(\frac{\varphi_{ij}(\mathbf{x}) + \psi_{ij}(\mathbf{y}) - \|\mathbf{x} - \mathbf{y}\|_2^2}{\varepsilon_1}\right)d\mu_{0_i}(\mathbf{x})d\mu_{1_j}(\mathbf{y}),$$

where $\phi_{ij}$ and $\psi_{ij}$ are Lagrange multipliers defined by::

$$\varphi_{ij}(\mathbf{x}) = -\varepsilon_1 \log \int_{\mathcal{Y}} \exp\left(\frac{\psi_{ij}(\mathbf{y}) - \|\mathbf{x} - \mathbf{y}\|_2^2}{\varepsilon_1}\right)d\mu_{1_j}(\mathbf{y}),$$

$$\psi_{ij}(\mathbf{y}) = -\varepsilon_1 \log \int_{\mathcal{X}} \exp\left(\frac{\varphi_{ij}(\mathbf{x}) - \|\mathbf{x} - \mathbf{y}\|_2^2}{\varepsilon_1}\right)d\mu_{0_i}(\mathbf{x}).$$

For $\mu_0^i(\mathbf{x}) = \mathcal{N}(\mathbf{x}|\mathbf{m}_{i_\mathbf{x}}, \Sigma_{i_{\mathbf{xx}}})$ and $\mu_1^j(\mathbf{y}) = \mathcal{N}(\mathbf{y}|\mathbf{m}_{j_y}, \Sigma_{j_{\mathbf{yy}}})$, it was shown in Janati et al. (2020) that $\varphi_{ij}$ and $\psi_{ij}$ admit closed-form solutions in the form of quadratic functions as follows ( **Proposition 1** in Janati et al. (2020)).

$$\varphi_{ij}(\mathbf{x}) = -(\mathbf{x} - \mathbf{m}_{i_\mathbf{x}})^T U_{ij}(\mathbf{x} - \mathbf{m}_{i_\mathbf{x}}), \qquad U_{ij} = \Sigma_{j_\mathbf{yy}}\left(\Sigma_{ij}^{\varepsilon_1} + \varepsilon_1 \mathbf{I}_d\right)^{-1} - \mathbf{I}_d$$

$$\psi_{ij}(\mathbf{y}) = -(\mathbf{y} - \mathbf{m}_{j_y})^T V_{ij}(\mathbf{y} - \mathbf{m}_{j_y}), \qquad V_{ij} = \left(\Sigma_{ij}^{\varepsilon_1} + \varepsilon_1 \mathbf{I}_d\right)^{-1}\Sigma_{i_\mathbf{xx}} - \mathbf{I}_d \tag{40}$$

where $\Sigma_{ij}^{\varepsilon_1} = \Sigma_{i_\mathbf{xx}}^{\frac{1}{2}}\Gamma_{ij}^{\varepsilon_1}\Sigma_{i_\mathbf{xx}}^{-\frac{1}{2}} - \frac{\varepsilon_1}{2}I_d$, and $\Gamma_{ij}^{\varepsilon_1} = (\Sigma_{i_\mathbf{xx}}^{\frac{1}{2}}\Sigma_{j_\mathbf{yy}}\Sigma_{i_\mathbf{xx}}^{\frac{1}{2}} + \frac{\varepsilon_1^2}{4}I_d)^{\frac{1}{2}}$.

Accordingly, the closed-form unique solution for $dp^*_{ij}$ can be obtained as:

$$p^*_{ij}(\mathbf{x}, \mathbf{y}) = \mathcal{N}\left(\begin{bmatrix}\mathbf{x}\\\mathbf{y}\end{bmatrix} \middle| \begin{bmatrix}\mathbf{m}_{i_\mathbf{x}}\\\mathbf{m}_{i_\mathbf{y}}\end{bmatrix}, \begin{bmatrix}\Sigma_{i_\mathbf{xx}} & \Sigma_{ij}^{\varepsilon_1}\\\Sigma_{ij}^{\varepsilon_1 T} & \Sigma_{i_\mathbf{yy}}\end{bmatrix}\right) \tag{41}$$

Therefore, the optimal mixture transport policy is itself a GMM, given by:

$$\pi(\mathbf{x}, \mathbf{y}) = \sum_{i,j}^K \omega_{ij}p_{ij}(\mathbf{x}, \mathbf{y}) = \sum_{i,j} \omega_{ij}\mathcal{N}\left(\begin{bmatrix}\mathbf{x}\\\mathbf{y}\end{bmatrix} \middle| \begin{bmatrix}\mathbf{m}_{i_\mathbf{x}}\\\mathbf{m}_{i_\mathbf{y}}\end{bmatrix}, \begin{bmatrix}\Sigma_{i_\mathbf{xx}} & \Sigma_{i_\mathbf{xy}}\\\Sigma_{i_\mathbf{xy}}^T & \Sigma_{i_\mathbf{yy}}\end{bmatrix}\right). \tag{42}$$

$\square$

## A.1 Stability of OMT maps

This section aims to establish stability bounds for transport maps in OMT framework under variations of the source measure. To achieve this, we proceed in two steps. First, in Section A.1.1, we show that when all measures have bounded support, the OMT map satisfies locally Lipschitz stability (see Lemma 2). Next, in Section A.1.2, we introduce another notion of tilt stability (entropic stability) in Definition 5. Corollary 2 establishes that OMT map is tilt-stable. Combining these two stability results allows us to prove Theorem 2.

**Transport map in OMT.** By the definition of the OMT problem in (11), the corresponding transport map satisfies

$$T_{\text{OMT}}^{\nu_0 \to \nu_1}(\mathbf{x}) = \int \mathbf{y} d\pi_{\text{OMT}}(\mathbf{y}|\mathbf{x}),$$

where $d\pi_{\text{OMT}}(\mathbf{x}, \mathbf{y}) \in G_{K_0 K_1}(\mathbb{R}^d)$. The conditional distribution can be expressed as

$$d\pi_{\text{OMT}}(\mathbf{y}|\mathbf{x}) = \frac{d\pi_{\text{OMT}}(\mathbf{x}, \mathbf{y})}{\int_{\mathbf{y}'} d\pi_{\text{OMT}}(\mathbf{x}, \mathbf{y}')}$$

$$= \frac{\sum_{i,j} \omega_{ij} dp_{ij}(\mathbf{x}, \mathbf{y})}{\sum_{r,k} \omega_{rk} \int_{\mathbf{y}'} dp_{rk}(\mathbf{x}, \mathbf{y}')} = \frac{\sum_{i,j} \omega_{ij} dp_i(\mathbf{x}) dp_{ij}(\mathbf{y}|\mathbf{x})}{\sum_{r,k} \omega_{rk} \int_{\mathbf{y}'} dp_{rk}(\mathbf{x}, \mathbf{y}')}$$

$$= \sum_{i,j} \underbrace{\frac{\omega_{ij} \int_{\mathbf{y}} dp_{ij}(\mathbf{x}, \mathbf{y})}{\sum_{r,k} \omega_{rk} \int_{\mathbf{y}'} dp_{rk}(\mathbf{x}, \mathbf{y}')}}_{\tilde{\omega}_{ij}(\mathbf{x})} dp_{ij}(\mathbf{y}|\mathbf{x}) = \sum_{i,j} \tilde{\omega}_{ij}(\mathbf{x}) dp_{ij}(\mathbf{y}|\mathbf{x}), \tag{43}$$

where $\sum_{i,j} \tilde{\omega}_{ij}(\mathbf{x}) = 1$. Accordingly, the conditional distribution is also a GMM. Consequently, the transport map can be decomposed as

$$T_{\text{OMT}}^{\nu_0 \to \nu_1}(\mathbf{x}) = \sum_{i,j} \tilde{\omega}_{ij}(\mathbf{x}) \int \mathbf{y} dp_{ij}(\mathbf{y}|\mathbf{x}) = \sum_{i,j} \tilde{\omega}_{ij}(\mathbf{x}) T^{\mu_{i_0} \to \mu_{j_1}}(\mathbf{x}), \tag{44}$$

where $T^{\mu_{i_0} \to \mu_{j_1}}(\mathbf{x})$ denotes the entropic map transporting the $i$-th Gaussian component of $\nu_0$ to the $j$-th Gaussian component of $\nu_1$. For notational simplicity, in what follows we denote $T_{\text{OMT}}^{\nu_0 \to \nu_1}(\mathbf{x}) := T_{\text{OMT}}(\mathbf{x})$, with

$$T_{\text{OMT}}(\mathbf{x}) := \sum_{i,j} \tilde{\omega}_{ij}(\mathbf{x}) T_{ij}(\mathbf{x}).$$

### A.1.1 Local Lipschitz stability of $T_{\text{OMT}}(\mathbf{x})$

**Definition 3** (Local Lipschitzness). *The local robustness, also known as local Lipschitzness (Mangall et al., 2021) at a point $(x_c)$, requires that*

$$\forall x, x' \in S, \ \|f(x) - f(x')\| \le L\|x - x'\|,$$

*where $\exists c > 0, \ S := \{x \in \mathbb{R}^d \mid \|x - x_c\| \le c\}$.*

**Definition 4** (Doubly truncated GMM). *Let $\nu \in G_K(\mathbb{R}^d)$ be an $K$-component GMM defined as*

$$\nu(\mathbf{x}) = \sum_i^K \alpha_i \mu_i(\mathbf{x})$$

*We define the truncation set $S_c$ as the closed (bounded) ball of radius $c \in \mathbb{R}^+$ centered at the global mean, $\bar{\mathbf{m}}$ as follows.*

$$S_c = \{\mathbf{x} \sim \nu_0(\mathbf{x}) \mid \|\mathbf{x} - \bar{\mathbf{m}}\|_2^2 \leq c^2, \bar{\mathbf{m}} = \sum_i \alpha_i \mathbf{m}_{i_\mathbf{x}}\}. \tag{45}$$

*Accordingly, each truncated component is given by*

$$\mu_i(\mathbf{x}, c) = \begin{cases} \dfrac{\mu_i(\mathbf{x})}{\kappa_i} & \text{if } x \in S_c \\ 0 & \text{otherwise} \end{cases}$$

*where $\kappa_i = \int_{-c}^{c} d\mu_i(\mathbf{x})$, is the normalization constant. Consequently, the truncated mixture measure becomes*

$$\nu(\mathbf{x}, c) = \begin{cases} \sum_i \dfrac{\alpha_i}{\kappa_i} \mu_i(\mathbf{x}) = \sum_i \alpha_i' \mu_i(\mathbf{x}) & \text{if } x \in S_c \\ 0 & \text{otherwise} \end{cases}. \tag{46}$$

**Lemma 2.** *Let $\pi_{\text{OMT}}(\cdot|\mathbf{x})$, be a conditional OMT coupling between two dtGMM, i.e., $\nu_0(\mathbf{x}, c) \in G_{K_0}(\mathbb{R}^d), \nu_1 \in G_{K_1}(\mathbb{R}^d)$. Then, for all $\mathbf{x} \in S_c$ the conditional OMT coupling is locally Lipschitz with Lipschitz constant:*

$$L_{\nu_0} = 2\varepsilon_1^{-1}\|\Sigma_{q_{\mathbf{yy}}} - \Sigma_{kq}^{\varepsilon_1}{}^T \Sigma_{k_{\mathbf{xx}}}^{-1}\Sigma_{kq}^{\varepsilon_1}\| + \left(\|\mathbf{m}_{q_\mathbf{y}}\| + \zeta_k(c)\|\Sigma_{kq}^{\varepsilon_1}{}^T\Sigma_{k_{\mathbf{xx}}}^{-1}\|\right)\left(\zeta_u(c)\|\Sigma_{s_{\mathbf{xx}}}^{-1}\| + \zeta_k(c)\|\Sigma_{k_{\mathbf{xx}}}^{-1}\|\right).$$

*Proof.* First we need to compute the Jacobian of $T_{\text{OMT}}$. The Jacobian of the transport map can be obtained as

$$\nabla T_{\text{OMT}}(\mathbf{x}) = \sum_{i,j} \tilde{\omega}_{ij}(\mathbf{x})\nabla T_{ij}(\mathbf{x}) + T_{ij}(\mathbf{x})\nabla^T\tilde{\omega}_{ij}(\mathbf{x}), \tag{47}$$

where

$$\forall i,j \quad \nabla T_{ij}(\mathbf{x}) = \int \mathbf{y}\nabla_x dp_{ij}(\mathbf{y}|\mathbf{x}), \quad \nabla\tilde{\omega}_{ij}(\mathbf{x}) = \nabla\left(\frac{\omega_{ij}\int_\mathbf{y} dp_{ij}(\mathbf{x},\mathbf{y})}{\sum_{r,k}\omega_{rk}\int_{\mathbf{y}'} dp_{rk}(\mathbf{x},\mathbf{y}')}\right) \tag{48}$$

For notational convenience, define $dp_{ij}(\mathbf{y}|\mathbf{x}) = p_{ij}(\mathbf{y}|\mathbf{x})d\mathbf{y}$, $d\mu_{0_i}(\mathbf{x}) = \mu_{0_i}(\mathbf{x})d\mathbf{x}$, and $d\mu_{1_j}(\mathbf{y}) = \mu_{1_j}(\mathbf{y})d\mathbf{y}$.

Following the derivation in Section 4.2, we obtain

$$p_{ij}(\mathbf{x},\mathbf{y}) = \exp\left(\frac{\varphi_{ij}(\mathbf{x}) + \psi_{ij}(\mathbf{y}) - \|\mathbf{x}-\mathbf{y}\|_2^2}{\varepsilon_1}\right)\mu_{0_i}(\mathbf{x})\mu_{1_j}(\mathbf{y}),$$

$$\varphi_{ij}(\mathbf{x}) = -\varepsilon_1\log\int_\mathcal{Y}\exp\left(\frac{\psi_{ij}(\mathbf{y}) - \|\mathbf{x}-\mathbf{y}\|_2^2}{\varepsilon_1}\right)\mu_{1_j}(\mathbf{y})d\mathbf{y}.$$

This implies

$$p_{ij}(\mathbf{y}|\mathbf{x}) = \frac{p_{ij}(\mathbf{x},\mathbf{y})}{\int_{\mathbf{y}'} p_{ij}(\mathbf{x},\mathbf{y}')d\mathbf{y}'} \tag{49}$$

$$= \frac{\exp\left(\frac{\varphi_{ij}(\mathbf{x})}{\varepsilon_1}\right)\mu_{0_i}(\mathbf{x})\exp\left(\frac{\psi_{ij}(\mathbf{y}) - \|\mathbf{x}-\mathbf{y}\|_2^2}{\varepsilon_1}\right)\mu_{1_j}(\mathbf{y})}{\exp\left(\frac{\varphi_{ij}(\mathbf{x})}{\varepsilon_1}\right)\mu_{0_i}(\mathbf{x})\int_{\mathbf{y}'}\exp\left(\frac{\psi_{ij}(\mathbf{y}') - \|\mathbf{x}-\mathbf{y}'\|_2^2}{\varepsilon_1}\right)\mu_{1_j}(\mathbf{y}')d\mathbf{y}'}$$

$$= \frac{\exp\left(\frac{\psi_{ij}(\mathbf{y}) - \|\mathbf{x}-\mathbf{y}\|_2^2}{\varepsilon_1}\right)\mu_{1_j}(\mathbf{y})}{\int_{\mathbf{y}'}\exp\left(\frac{\psi_{ij}(\mathbf{y}') - \|\mathbf{x}-\mathbf{y}'\|_2^2}{\varepsilon_1}\right)\mu_{1_j}(\mathbf{y}')d\mathbf{y}'} \tag{50}$$

$$\log p_{ij}(\mathbf{y}|\mathbf{x}) = \log\left(\exp\left(\frac{\psi_{ij}(\mathbf{y}) - \|\mathbf{x}-\mathbf{y}\|_2^2}{\varepsilon_1}\right)\mu_{1_j}(\mathbf{y})\right) - \log\left(\int_{\mathbf{y}'}\exp\left(\frac{\psi_{ij}(\mathbf{y}') - \|\mathbf{x}-\mathbf{y}'\|_2^2}{\varepsilon_1}\right)\mu_{1_j}(\mathbf{y}')d\mathbf{y}'\right)$$

$$= \left(\frac{\psi_{ij}(\mathbf{y}) - \|\mathbf{x}-\mathbf{y}\|_2^2}{\varepsilon_1}\right) + \log\mu_{1_j}(\mathbf{y}) - \log\left(\int_{\mathbf{y}'}\exp\left(\frac{\psi_{ij}(\mathbf{y}') - \|\mathbf{x}-\mathbf{y}'\|_2^2}{\varepsilon_1}\right)\mu_{1_j}(\mathbf{y}')d\mathbf{y}'\right)$$

$$= \left(\frac{\psi_{ij}(\mathbf{y}) - \|\mathbf{x}-\mathbf{y}\|_2^2}{\varepsilon_1}\right) + \log\mu_{1_j}(\mathbf{y}) + \varepsilon^{-1}\varphi_{ij}(\mathbf{x}) \tag{51}$$

Differentiating with respect to $\mathbf{x}$,

$$\nabla_x \log p_{ij}(\mathbf{y}|\mathbf{x}) = -2\varepsilon^{-1}(\mathbf{x}-\mathbf{y})^T + \varepsilon^{-1}\nabla\varphi_{ij}(\mathbf{x}). \tag{52}$$

Next, using the definition of $\varphi_{ij}(\mathbf{x})$,

$$\nabla\varphi_{ij}(\mathbf{x}) = 2\frac{\int_{\mathbf{y}}(\mathbf{x}-\mathbf{y})^T \exp\left(\frac{\psi_{ij}(\mathbf{y}) - \|\mathbf{x}-\mathbf{y}\|_2^2}{\varepsilon_1}\right)\mu_{1_j}(\mathbf{y})d\mathbf{y}}{\int_{\mathbf{y}'}\exp\left(\frac{\psi_{ij}(\mathbf{y}') - \|\mathbf{x}-\mathbf{y}'\|_2^2}{\varepsilon_1}\right)\mu_{1_j}(\mathbf{y}')d\mathbf{y}'}$$

$$= 2\mathbf{x}^T - 2\int_{\mathbf{y}}\mathbf{y}^T\frac{\exp\left(\frac{\psi_{ij}(\mathbf{y}) - \|\mathbf{x}-\mathbf{y}\|_2^2}{\varepsilon_1}\right)\mu_{1_j}(\mathbf{y})d\mathbf{y}}{\int_{\mathbf{y}'}\exp\left(\frac{\psi_{ij}(\mathbf{y}') - \|\mathbf{x}-\mathbf{y}'\|_2^2}{\varepsilon_1}\right)\mu_{1_j}(\mathbf{y}')d\mathbf{y}'}$$

$$= 2\mathbf{x}^T - 2\int_{\mathbf{y}}\mathbf{y}^T p_{ij}(\mathbf{y}|\mathbf{x})d\mathbf{y} \tag{53}$$

$$= 2(\mathbf{x} - T_{ij}(\mathbf{x}))^T \tag{54}$$

Substituting Eq. 53 into Eq. 52 yields

$$\nabla_x \log p_{ij}(\mathbf{y}|\mathbf{x}) = -2\varepsilon_1^{-1}(\mathbf{x}-\mathbf{y})^T + 2\varepsilon_1^{-1}(\mathbf{x} - T_{ij}(\mathbf{x}))^T$$

$$= 2\varepsilon_1^{-1}(\mathbf{y} - T_{ij}(\mathbf{x}))^T \tag{55}$$

Applying the logarithmic derivative formula

$$\nabla_x p_{ij}(\mathbf{y}|\mathbf{x}) = \nabla_x \log p_{ij}(\mathbf{y}|\mathbf{x})p_{ij}(\mathbf{y}|\mathbf{x}),$$

$$= 2\varepsilon_1^{-1}(\mathbf{y} - T_{ij}(\mathbf{x}))^T p_{ij}(\mathbf{y}|\mathbf{x}).$$

Hence, the Jacobian of $T_{ij}$ can be defined as

$$\nabla_x T_{ij}(\mathbf{x}) = \int \mathbf{y}\nabla_x p_{ij}(\mathbf{y}|\mathbf{x})d\mathbf{y}$$

$$= 2\varepsilon_1^{-1}\int \mathbf{y}(\mathbf{y} - T_{ij}(\mathbf{x}))^T p_{ij}(\mathbf{y}|\mathbf{x})d\mathbf{y}$$

$$= 2\varepsilon_1^{-1}\left(\int \mathbf{y}\mathbf{y}^T p_{ij}(\mathbf{y}|\mathbf{x})d\mathbf{y} - \int \mathbf{y}p_{ij}(\mathbf{y}|\mathbf{x})T_{ij}^T(\mathbf{x})d\mathbf{y}\right)$$

$$= 2\varepsilon_1^{-1}\left(\int \mathbf{y}\mathbf{y}^T p_{ij}(\mathbf{y}|\mathbf{x})d\mathbf{y} - T_{ij}(\mathbf{x})T_{ij}^T(\mathbf{x})d\mathbf{y}\right)$$

$$= 2\varepsilon_1^{-1}\left(\mathbb{E}_{p_{ij}(\mathbf{y}|\mathbf{x})}\left[\mathbf{y}\mathbf{y}^T\right] - \mathbb{E}_{p_{ij}(\mathbf{y}|\mathbf{x})}^2\left[\mathbf{y}\right]\right) = 2\varepsilon_1^{-1}\text{Cov}_{p_{ij}(\mathbf{y}|\mathbf{x})}(Y|X=\mathbf{x}) \tag{56}$$

If $\mathbf{x}, \mathbf{y} \sim p_{ij}(\mathbf{x}, \mathbf{y}) = \mathcal{N}(\mathbf{m}_{ij}, \Sigma_{ij})$ (see Eq. 41), then

$$T_{ij}(\mathbf{x}) = \mathbf{E}_{p_{ij}(\mathbf{y}|\mathbf{x})}[\mathbf{y}] = \mathbf{m}_{j_{\mathbf{y}}} + \Sigma_{ij}^{\varepsilon_1}{}^T\Sigma_{i_{\mathbf{xx}}}^{-1}(\mathbf{x} - \mathbf{m}_{i_{\mathbf{x}}}) \tag{57}$$

$$\nabla T_{ij}(\mathbf{x}) = 2\varepsilon_1^{-1}\left(\Sigma_{j_{\mathbf{yy}}} - \Sigma_{ij}^{\varepsilon_1}{}^T\Sigma_{i_{\mathbf{xx}}}^{-1}\Sigma_{ij}^{\varepsilon_1}\right) \tag{58}$$

Now, we focus on the derivative of the mixture weights $\tilde{\omega}_{ij}$ in Eq. 43. Recall that

$$\tilde{\omega}_{ij}(\mathbf{x}) = \left(\frac{\omega_{ij}\int_{\mathbf{y}}p_{ij}(\mathbf{x},\mathbf{y})d\mathbf{y}}{\sum_{r,k}\omega_{rk}\int_{\mathbf{y}'}p_{rk}(\mathbf{x},\mathbf{y}')d\mathbf{y}'}\right)$$

Following the derivation in Section 4.2, the optimality conditions yield

$$\omega_{ij} = \exp\left(\frac{\lambda_i + \tau_j - \mathcal{L}_{p_{ij}}}{\varepsilon_2}\right)\alpha_{0_i}\alpha_{1_j} \qquad (59)$$

where

$$\mathcal{L}_{p_{ij}} = \mathbb{E}_{p_{ij}}\left[\varphi_{ij}(\mathbf{x})\right] + \mathbb{E}_{p_{ij}}\left[\psi_{ij}(\mathbf{y})\right] = \int\int\left(\varphi_{ij}(\mathbf{x}) + \psi_{ij}(\mathbf{y})\right)p_{ij}(\mathbf{x},\mathbf{y})d\mathbf{x}d\mathbf{y}$$

and the corresponding dual potentials are given by

$$\lambda_i = -\varepsilon_2\log\sum_j\exp\left(\frac{\tau_j - \mathcal{L}_{p_{ij}}}{\varepsilon_2}\right)\alpha_{1_j}, \qquad \tau_j = -\varepsilon_2\log\sum_i\exp\left(\frac{\lambda_i - \mathcal{L}_{p_{ij}}}{\varepsilon_2}\right)\alpha_{0_i}. \qquad (60)$$

Expanding the component distributions $p_{ij}$ in exponential form and substituting the optimality conditions from Eq. 59-60, we obtain

$$\tilde{\omega}_{ij}(\mathbf{x}) = \frac{\omega_{ij}\mu_{0_i}(\mathbf{x})\exp\left(\frac{\varphi_{ij}(\mathbf{x})}{\varepsilon_1}\right)\int_{\mathbf{y}}\exp\left(\frac{\psi_{ij}(\mathbf{y}) - \|\mathbf{x}-\mathbf{y}\|_2^2}{\varepsilon_1}\right)\mu_{1_j}(\mathbf{y})d\mathbf{y}}{\sum_{r,k}\exp\left(\frac{\lambda_r + \tau_k - \mathcal{L}_{p_{rk}}}{\varepsilon_2}\right)\alpha_{0_r}\alpha_{1_k}\mu_{0_r}(\mathbf{x})\exp\left(\frac{\varphi_{rk}(\mathbf{x})}{\varepsilon_1}\right)\int_{\mathbf{y}'}\exp\left(\frac{\psi_{rk}(\mathbf{y}') - \|\mathbf{x}-\mathbf{y}'\|_2^2}{\varepsilon_1}\right)\mu_{1_j}(\mathbf{y}')d\mathbf{y}'}.$$

Having exponential terms cancel out, yielding the simplified expression.

$$\tilde{\omega}_{ij}(\mathbf{x}) = \frac{\omega_{ij}\mu_{0_i}(\mathbf{x})\exp\left(\frac{\varphi_{ij}(\mathbf{x})}{\varepsilon_1}\right)\exp\left(\frac{-\varphi_{ij}(\mathbf{x})}{\varepsilon_1}\right)}{\sum_r\exp\left(\frac{\lambda_r}{\varepsilon_2}\right)\alpha_{0_r}\mu_{0_r}(\mathbf{x})\sum_k\exp\left(\frac{\tau_k - \mathcal{L}_{p_{rk}}(\mathbf{x},\mathbf{y})}{\varepsilon_2}\right)\exp\left(\frac{\varphi_{rk}(\mathbf{x})}{\varepsilon_1}\right)\exp\left(\frac{-\varphi_{rk}(\mathbf{x})}{\varepsilon_1}\right)\alpha_{1_k}}$$

$$= \frac{\omega_{ij}\mu_{0_i}(\mathbf{x})}{\sum_r\exp\left(\frac{\lambda_r}{\varepsilon_2}\right)\alpha_{0_r}\mu_{0_r}(\mathbf{x})\underbrace{\sum_k\exp\left(\frac{\tau_k - \mathcal{L}_{p_{rk}}(\mathbf{x},\mathbf{y})}{\varepsilon_2}\right)\alpha_{1_k}}_{\exp\left(\frac{-\lambda_r}{\varepsilon_2}\right)}}$$

$$= \frac{\omega_{ij}\mu_{0_i}(\mathbf{x})}{\sum_r\alpha_{0_r}\mu_{0_r}(\mathbf{x})} = \frac{\omega_{ij}\mu_{0_i}(\mathbf{x})}{\nu_0(\mathbf{x})}\ .$$

Differentiating $\tilde{\omega}_{ij}(\mathbf{x})$ with respect to $\mathbf{x}$ gives

$$\nabla\tilde{\omega}_{ij}(\mathbf{x}) = \frac{\omega_{ij}}{\nu_0(\mathbf{x})}\nabla\mu_{0_i}(\mathbf{x}) - \frac{\tilde{\omega}_{ij}(\mathbf{x})}{\nu_0(\mathbf{x})}\nabla\nu_0(\mathbf{x}).$$

For $\nu_0 \in G_{K_0}(\mathbb{R}^d)$, each component density can be written as

$$\nabla\mu_{0_i}(\mathbf{x}) = \mu_{0_i}(\mathbf{x})\nabla g_i(\mathbf{x}),$$

where

$$g_i(\mathbf{x}) = -\frac{1}{2}(\mathbf{x} - \mathbf{m}_{i_\mathbf{x}})^T\Sigma_{i_{\mathbf{x}\mathbf{x}}}^{-1}(\mathbf{x} - \mathbf{m}_{i_\mathbf{x}}). \qquad (61)$$

Taking the gradient gives

$$\nabla g_i(\mathbf{x}) = -\Sigma_{i_{\mathbf{x}\mathbf{x}}}^{-1}(\mathbf{x} - \mathbf{m}_{i_\mathbf{x}})$$

$$\nabla\mu_{0_i}(\mathbf{x}) = -\mu_{0_i}(\mathbf{x})\Sigma_{i_{\mathbf{x}\mathbf{x}}}^{-1}(\mathbf{x} - \mathbf{m}_{i_\mathbf{x}})$$

$$\nabla\nu_0(\mathbf{x}) = -\sum_r\alpha_{0_r}\mu_{0_r}\Sigma_{r_{\mathbf{x}\mathbf{x}}}^{-1}(\mathbf{x} - \mathbf{m}_{r_\mathbf{x}}). \qquad (62)$$

Substituting Eq. 62 into the derivative expression implies

$$\nabla \tilde{\omega}_{ij}(\mathbf{x}) = -\tilde{\omega}_{ij}(\mathbf{x})\Sigma_{i_{\mathbf{xx}}}^{-1}(\mathbf{x} - \mathbf{m}_{i_{\mathbf{x}}}) + \frac{\tilde{\omega}_{ij}(\mathbf{x})}{\nu_0(\mathbf{x})} \sum_r \alpha_{0_r}\mu_{0_r}(\mathbf{x})\Sigma_{r_{\mathbf{xx}}}^{-1}(\mathbf{x} - \mathbf{m}_{r_{\mathbf{x}}}),$$

$$= \tilde{\omega}_{ij}(\mathbf{x})\left( \frac{\displaystyle\sum_r \alpha_{0_r}\mu_{0_r}(\mathbf{x})\Sigma_{r_{\mathbf{xx}}}^{-1}(\mathbf{x} - \mathbf{m}_{r_{\mathbf{x}}})}{\nu_0(\mathbf{x})} - \Sigma_{i_{\mathbf{xx}}}^{-1}(\mathbf{x} - \mathbf{m}_{i_{\mathbf{x}}}) \right). \tag{63}$$

By substituting the expressions in Eqs. 56, 57, and 63 into Eq. 47, we can obtain

$$\nabla T_{\mathrm{OMT}}(\mathbf{x}) = \sum_{i,j} 2\varepsilon_1^{-1}\tilde{\omega}_{ij}(\mathbf{x})\mathrm{Cov}_{p_{ij}(\mathbf{y}|\mathbf{x})}(Y|X = \mathbf{x}) +$$

$$\tilde{\omega}_{ij}(\mathbf{x})\mathbb{E}_{p_{ij}(\mathbf{y}|\mathbf{x})}[\mathbf{y}]\left( \frac{\displaystyle\sum_r \alpha_{0_r}\mu_{0_r}(\mathbf{x})\Sigma_{r_{\mathbf{xx}}}^{-1}(\mathbf{x} - \mathbf{m}_{r_{\mathbf{x}}})}{\nu_0(\mathbf{x})} - \Sigma_{i_{\mathbf{xx}}}^{-1}(\mathbf{x} - \mathbf{m}_{i_{\mathbf{x}}}) \right)^T,$$

$$= \sum_{i,j} \tilde{\omega}_{ij}(\mathbf{x})2\varepsilon_1^{-1}\mathrm{Cov}_{p_{ij}(\mathbf{y}|\mathbf{x})}(Y|X = \mathbf{x}) +$$

$$\tilde{\omega}_{ij}(\mathbf{x})\mathbf{m}_{ij}(Y|X = \mathbf{x})\left( \frac{\displaystyle\sum_r \alpha_{0_r}\mu_{0_r}(\mathbf{x})\Sigma_{r_{\mathbf{xx}}}^{-1}(\mathbf{x} - \mathbf{m}_{r_{\mathbf{x}}})}{\nu_0(\mathbf{x})} - \Sigma_{i_{\mathbf{xx}}}^{-1}(\mathbf{x} - \mathbf{m}_{i_{\mathbf{x}}}) \right)^T.$$

For simplicity, let's define $\mathrm{Cov}_{\mathbf{y}|\mathbf{x}}^{ij} := \mathrm{Cov}_{p_{ij}(\mathbf{y}|\mathbf{x})}(Y|X = \mathbf{x})$ and $\mathbf{m}_{\mathbf{y}|\mathbf{x}}^{ij} := \mathbf{m}_{ij}(Y|X = \mathbf{x})$. Then, the gradient can be compactly expressed as

$$\nabla T_{\mathrm{OMT}}(\mathbf{x}) = \sum_{i,j} 2\tilde{\omega}_{ij}(\mathbf{x})\varepsilon_1^{-1}\mathrm{Cov}_{\mathbf{y}|\mathbf{x}}^{ij} + \sum_{i,j} \tilde{\omega}_{ij}(\mathbf{x})\mathbf{m}_{\mathbf{y}|\mathbf{x}}^{ij}\left( \sum_r \frac{\alpha_{0_r}\mu_{0_r}(\mathbf{x})}{\nu_0(\mathbf{x})}(\mathbf{x} - \mathbf{m}_{r_{\mathbf{x}}})^T\Sigma_{r_{\mathbf{xx}}}^{-1} - (\mathbf{x} - \mathbf{m}_{i_{\mathbf{x}}})^T\Sigma_{i_{\mathbf{xx}}}^{-1} \right)$$

Since $\alpha_{0_r} = \sum_k \omega_{rk}$, we can then have,

$$\sum_r \frac{\alpha_{0_r}\mu_{0_r}(\mathbf{x})}{\nu_0(\mathbf{x})}\Sigma_{r_{\mathbf{xx}}}^{-1}(\mathbf{x} - \mathbf{m}_{r_{\mathbf{x}}}) = \sum_{r,k} \frac{\omega_{rk}\mu_{0_r}(\mathbf{x})}{\nu_0(\mathbf{x})}\Sigma_{r_{\mathbf{xx}}}^{-1}(\mathbf{x} - \mathbf{m}_{r_{\mathbf{x}}})$$

$$= \sum_{r,k} \tilde{\omega}_{rk}(\mathbf{x})\Sigma_{r_{\mathbf{xx}}}^{-1}(\mathbf{x} - \mathbf{m}_{r_{\mathbf{x}}})$$

Using the definition of $g(\mathbf{x})$ and its derivative in Eq. 61 and 62, we define

$$\overline{\nabla \mathbf{g}(\mathbf{x})} := \sum_{r,k} \tilde{\omega}_{rk}(\mathbf{x})\nabla g_r(\mathbf{x}) = -\sum_{r,k} \tilde{\omega}_{rk}(\mathbf{x})\Sigma_{r_{\mathbf{xx}}}^{-1}(\mathbf{x} - \mathbf{m}_{r_{\mathbf{x}}})$$

We define the average conditional covariance across mixture components as

$$\overline{\mathrm{Cov}_{\mathbf{y}|\mathbf{x}}} := \sum_{i,j} \tilde{\omega}_{ij}(\mathbf{x})\mathrm{Cov}_{\mathbf{y}|\mathbf{x}}^{ij} = \sum_{i,j} \tilde{\omega}_{ij}(\mathbf{x})(\Sigma_{j_{\mathbf{yy}}} - {\Sigma_{ij}^{\varepsilon_1}}^T\Sigma_{i_{\mathbf{xx}}}^{-1}\Sigma_{ij}^{\varepsilon_1}). \tag{64}$$

Similarly, the average conditional mean is given by

$$\overline{\mathbf{m}}_{\mathbf{y}|\mathbf{x}} := \sum_{i,j} \tilde{\omega}_{ij}(\mathbf{x})\mathbf{m}_{\mathbf{y}|\mathbf{x}}^{ij} = \sum_{i,j} \tilde{\omega}_{ij}(\mathbf{x})\left( \mathbf{m}_{j_{\mathbf{y}}} + {\Sigma_{ij}^{\varepsilon_1}}^T\Sigma_{i_{\mathbf{xx}}}^{-1}(\mathbf{x} - \mathbf{m}_{i_{\mathbf{x}}}) \right)$$

$$= \sum_{i,j} \tilde{\omega}_{ij}(\mathbf{x})\mathbf{m}_{j_{\mathbf{y}}} - \sum_{i,j} \tilde{\omega}_{i,j}(\mathbf{x}){\Sigma_{ij}^{\varepsilon_1}}^T\nabla g_i(\mathbf{x})$$

Substituting these definitions, we obtain

$$
\begin{aligned}
\nabla T_{\text{OMT}}(\mathbf{x}) &= 2\varepsilon_1^{-1}\overline{\text{Cov}}_{\mathbf{y}|\mathbf{x}} + \sum_{i,j} \tilde{\omega}_{ij}(\mathbf{x})\mathbf{m}_{\mathbf{y}|\mathbf{x}}^{ij} \left( \sum_{r,k} \tilde{\omega}_{rk}\Sigma_{r_{\mathbf{xx}}}^{-1}(\mathbf{x} - \mathbf{m}_{r_{\mathbf{x}}}) - \Sigma_{i_{\mathbf{xx}}}^{-1}(\mathbf{x} - \mathbf{m}_{i_{\mathbf{x}}}) \right)^T \\
&= 2\varepsilon_1^{-1}\overline{\text{Cov}}_{\mathbf{y}|\mathbf{x}} + \sum_{i,j} \tilde{\omega}_{ij}(\mathbf{x})\mathbf{m}_{\mathbf{y}|\mathbf{x}}^{ij} \left( \nabla g_i(\mathbf{x}) - \overline{\nabla \mathbf{g}(\mathbf{x})} \right)^T \\
&= 2\varepsilon_1^{-1}\overline{\text{Cov}}_{\mathbf{y}|\mathbf{x}} + \sum_{i,j} \tilde{\omega}_{ij}(\mathbf{x}) \left( \mathbf{m}_{j_{\mathbf{y}}} - \Sigma_{ij}^{\varepsilon_1 T}\nabla g_i(\mathbf{x}) \right) \left( \nabla g_i(\mathbf{x}) - \overline{\nabla \mathbf{g}(\mathbf{x})} \right)^T \\
&= 2\varepsilon_1^{-1}\overline{\text{Cov}}_{\mathbf{y}|\mathbf{x}} + \sum_{i,j} \tilde{\omega}_{ij}(\mathbf{x})\mathbf{m}_{j_{\mathbf{y}}} \left( \nabla g_i(\mathbf{x}) - \overline{\nabla \mathbf{g}(\mathbf{x})} \right)^T + \sum_{i,j} \tilde{\omega}_{ij}(\mathbf{x})\Sigma_{ij}^{\varepsilon_1 T}\nabla g_i(\mathbf{x}) \left( \overline{\nabla \mathbf{g}(\mathbf{x})} - \nabla g_i(\mathbf{x}) \right)^T
\end{aligned}
\tag{65}
$$

Taking norms on both sides yields

$$
\begin{aligned}
\|\nabla T_{\text{OMT}}(\mathbf{x})\| &= \|2\varepsilon_1^{-1}\overline{\text{Cov}}_{\mathbf{y}|\mathbf{x}} + \sum_{i,j} \tilde{\omega}_{ij}(\mathbf{x})\mathbf{m}_{j_{\mathbf{y}}} \left( \nabla g_i(\mathbf{x}) - \overline{\nabla \mathbf{g}(\mathbf{x})} \right)^T + \sum_{i,j} \tilde{\omega}_{ij}(\mathbf{x})\Sigma_{ij}^{\varepsilon_1 T}\nabla g_i(\mathbf{x}) \left( \overline{\nabla \mathbf{g}(\mathbf{x})} - \nabla g_i(\mathbf{x}) \right)^T \| \\
&= \| \sum_{i,j} \tilde{\omega}_{ij}(\mathbf{x}) \left( 2\varepsilon_1^{-1}\text{Cov}_{\mathbf{y}|\mathbf{x}}^{ij} + \mathbf{m}_{j_{\mathbf{y}}} \left( \nabla g_i(\mathbf{x}) - \overline{\nabla \mathbf{g}(\mathbf{x})} \right)^T + \Sigma_{ij}^{\varepsilon_1 T}\nabla g_i(\mathbf{x}) \left( \overline{\nabla \mathbf{g}(\mathbf{x})} - \nabla g_i(\mathbf{x}) \right)^T \right) \|
\end{aligned}
$$

Applying the triangle inequality, we obtain

$$
\|\nabla T_{\text{OMT}}(\mathbf{x})\| \le \sum_{i,j} \tilde{\omega}_{ij}(\mathbf{x}) \left( 2\varepsilon_1^{-1}\|\text{Cov}_{\mathbf{y}|\mathbf{x}}^{ij}\| + \|\mathbf{m}_{j_{\mathbf{y}}} \left( \nabla g_i(\mathbf{x}) - \overline{\nabla \mathbf{g}(\mathbf{x})} \right)^T \| + \|\Sigma_{ij}^{\varepsilon_1 T}\nabla g_i(\mathbf{x}) \left( \nabla g_i(\mathbf{x}) - \overline{\nabla \mathbf{g}(\mathbf{x})} \right)^T \| \right)
\tag{66}
$$

Since the last two matrices inside the summation are $rank$ 1 matrices, their operator norms can be decomposed multiplicatively as follows.

$$
\begin{aligned}
\|\nabla T_{\text{OMT}}(\mathbf{x})\| &\le \sum_{i,j} \tilde{\omega}_{ij}(\mathbf{x}) \left( 2\varepsilon_1^{-1}\|\text{Cov}_{\mathbf{y}|\mathbf{x}}^{ij}\| + \|\mathbf{m}_{j_{\mathbf{y}}}\| \|\nabla g_i(\mathbf{x}) - \overline{\nabla \mathbf{g}(\mathbf{x})}\| + \|\Sigma_{ij}^{\varepsilon_1 T}\nabla g_i(\mathbf{x})\| \|\nabla g_i(\mathbf{x}) - \overline{\nabla \mathbf{g}(\mathbf{x})}\| \right) \\
&= \sum_{i,j} \tilde{\omega}_{ij}(\mathbf{x}) \left( 2\varepsilon_1^{-1}\|\text{Cov}_{\mathbf{y}|\mathbf{x}}^{ij}\| + \left( \|\mathbf{m}_{j_{\mathbf{y}}}\| + \|\Sigma_{ij}^{\varepsilon_1 T}\nabla g_i(\mathbf{x})\| \right) \left( \|\nabla g_i(\mathbf{x}) - \overline{\nabla \mathbf{g}(\mathbf{x})}\| \right) \right)
\end{aligned}
\tag{67}
$$

Imposing the truncation constraint on $\mathbf{x}$ by bounded set $S$, the norm values are bounded as

$$
\|\mathbf{x} - \mathbf{m}_{i_{\mathbf{x}}}\| \le c + \|\bar{\mathbf{m}}_0 - \mathbf{m}_{i_{\mathbf{x}}}\|
$$
$$
\|\nabla g_i(\mathbf{x})\| \le \|\Sigma_{i_{\mathbf{xx}}}^{-1}\|(c + \|\bar{\mathbf{m}}_0 - \mathbf{m}_{i_{\mathbf{x}}}\|)
\tag{68}
$$
$$
\zeta_i(c) := c + \|\bar{\mathbf{m}}_0 - \mathbf{m}_{i_{\mathbf{x}}}\|
\tag{69}
$$

$$
\|\Sigma_{ij}^{\varepsilon_1 T}\nabla g_i(\mathbf{x})\| \le \|\Sigma_{ij}^{\varepsilon_1 T}\Sigma_{i_{\mathbf{xx}}}^{-1}\| \zeta_i(c)
$$

$$
\|\overline{\nabla \mathbf{g}(\mathbf{x})}\| \le \sum_{r,k} \tilde{\omega}_{rk}(\mathbf{x})\zeta_r(c)\|\Sigma_{r_{\mathbf{xx}}}^{-1}\|
\tag{70}
$$

Given the bounds in Eqs. 68, 69, and 70, the norm of the Jacobian of the OMT map in Eq. 67 admits the following upper bound.

$$
\begin{aligned}
\|\nabla T_{\text{OMT}}(\mathbf{x})\| &\le \sum_{i,j} \tilde{\omega}_{ij}(\mathbf{x})2\varepsilon_1^{-1} \left( \|\text{Cov}_{\mathbf{y}|\mathbf{x}}^{ij}\| + \|\mathbf{m}_{j_{\mathbf{y}}}\| \|\Sigma_{i_{\mathbf{xx}}}^{-1}\|\zeta_i(c) + \|\mathbf{m}_{j_{\mathbf{y}}}\| \|\overline{\nabla \mathbf{g}(\mathbf{x})}\| \right) + \\
&\quad \sum_{i,j} \tilde{\omega}_{ij}(\mathbf{x}) \left( \zeta_i^2(c)\|\Sigma_{ij}^{\varepsilon_1 T}\Sigma_{i_{\mathbf{xx}}}^{-1}\| \|\Sigma_{i_{\mathbf{xx}}}^{-1}\| + \zeta_i(c)\|\Sigma_{ij}^{\varepsilon_1 T}\Sigma_{i_{\mathbf{xx}}}^{-1}\| \|\overline{\nabla \mathbf{g}(\mathbf{x})}\| \right) \\
&= \sum_{i,j} \tilde{\omega}_{ij}(\mathbf{x}) \left( 2\varepsilon_1^{-1}\|\text{Cov}_{\mathbf{y}|\mathbf{x}}^{ij}\| + \left( \|\mathbf{m}_{j_{\mathbf{y}}}\| + \zeta_i(c)\|\Sigma_{ij}^{\varepsilon_1 T}\Sigma_{i_{\mathbf{xx}}}^{-1}\| \right) \left( \|\overline{\nabla \mathbf{g}(\mathbf{x})}\| + \zeta_i(c)\|\Sigma_{i_{\mathbf{xx}}}^{-1}\| \right) \right)
\end{aligned}
\tag{71}
$$

The local Lipschitz constant can thus be defined as

$$L_{\nu_0} := \sup_{\mathbf{x} \in S_c} \|\nabla T_{\text{OMT}}(\mathbf{x})\| \tag{72}$$

For each supremum term, let

$$k, q := \arg\max_{i,j} \{2\varepsilon_1^{-1}\|\text{Cov}_{\mathbf{y}|\mathbf{x}}^{ij}\| + \left(\|\mathbf{m}_{j_\mathbf{y}}\| + \zeta_i(c)\|\Sigma_{ij}^{\varepsilon_1 T}\Sigma_{i_{\mathbf{xx}}}^{-1}\|\right)\left(\|\overline{\nabla \mathbf{g}(\mathbf{x})}\| + \zeta_i(c)\|\Sigma_{i_{\mathbf{xx}}}^{-1}\|\right)\}$$

$$u := \arg\max_i \{\zeta_i(c)\|\Sigma_{i_{\mathbf{xx}}}^{-1}\|\}$$

Accordingly, the Lipschitz constant admits the closed-form upper bound

$$L_{\nu_0} = 2\varepsilon_1^{-1}\|\Sigma_{q_{\mathbf{yy}}} - \Sigma_{kq}^{\varepsilon_1 T}\Sigma_{k_{\mathbf{xx}}}^{-1}\Sigma_{kq}^{\varepsilon_1}\| + \left(\|\mathbf{m}_{q_\mathbf{y}}\| + \zeta_k(c)\|\Sigma_{kq}^{\varepsilon_1 T}\Sigma_{k_{\mathbf{xx}}}^{-1}\|\right)\left(\zeta_u(c)\|\Sigma_{s_{\mathbf{xx}}}^{-1}\| + \zeta_k(c)\|\Sigma_{k_{\mathbf{xx}}}^{-1}\|\right).$$

$$\tag{73}$$

$\square$

### A.1.2 TILT STABILITY OF $T_{\text{OMT}}$

We now introduce a notion of stability for optimal transport mappings under small perturbations of the underlying measure. Intuitively, stability describes how sensitively the transport structure or barycenter of a measure responds when the measure is perturbed in a particular direction. One way to formalize it is through tilt stability (also known as entropic stability), which connects the change in the expectation of a measure to its relative entropy under exponential tilting.

**Definition 5** (Tilt stability (Entropic stability), Chen & Eldan (2022); Bauerschmidt et al. (2024)). *Let $\mu$ be a probability measure on $\mathcal{X} \subset \mathbb{R}^d$, with finite second moment and a function $f : \mathbb{R}^d \times \mathbb{R}^d \to \mathbb{R}_+$ and $\alpha > 0$, we say that $\mu$ is $\alpha$-entropically stable with respect to $f$ if*

$$f\left(\mathbb{E}_{\mathcal{T}_\mathbf{v}\mu}[\mathbf{x}], \mathbb{E}_\mu[\mathbf{x}]\right) \leq \alpha D_{KL}(T_\mathbf{v}\mu || \mu), \quad \forall \mathbf{v} \in \mathbb{R}^d \tag{74}$$

*where $\mathcal{T}_\mathbf{v}\mu$ denotes the exponential tilt of a the probability measure $\mu$ through vector $\mathbf{v} \in \mathbb{R}^d$ such that*

$$d\mathcal{T}_\mathbf{v}\mu(\mathbf{x}) := \frac{e^{\langle \mathbf{v}, \mathbf{x} \rangle}}{\int e^{\langle \mathbf{v}, \mathbf{z} \rangle} d\mu(\mathbf{z})} d\mu(\mathbf{x}).$$

**Lemma 3** (Lemma 40 Chen & Eldan (2022)). *Let $\nu$ be a probability measure on $\mathbb{R}^n$ with finite second moment and $A$ be positive-definite matrix. Suppose that for every $\mathbf{v} \in \mathbb{R}^d$ one has*

$$\|\text{Cov}(\mathcal{T}_\mathbf{v}\nu)\|_{\text{op}} \leq \alpha$$

*Then $\nu$ is tilt-stable with constant $\alpha \in \mathbb{R}^+$ with respect to the function $f(x, y) = \frac{1}{2}\|\mathbf{x} - \mathbf{y}\|_2^2$.*

**Corollary 2.** *Let $\pi_{\text{OMT}}(\cdot|\mathbf{x})$, be a conditional OMT coupling between two probability measures $\nu_0(\mathbf{x}, c_0) \in G_{K_0}(\mathbb{R}^d), \nu_1(\mathbf{y}, c_1) \in G_{K_1}(\mathbb{R}^d)$ for all $\mathbf{x} \in S_{c_0} = \{\mathbf{x} \sim \nu_0(\mathbf{x}) \mid \|\mathbf{x} - \bar{\mathbf{m}}\|_2^2 \leq c_0^2, \bar{\mathbf{m}} = 1/K_0 \sum_i \alpha_i \mathbf{m}_{i_\mathbf{x}}\}$. Then, for all $\mathbf{v} \in \mathbb{R}^d$ the conditional OMT coupling is tilt-stable with constant*

$$C_{\nu_0} := \sup_{\mathbf{x} \in S_{c_0}} \{\|\text{Cov}_{\pi_{\text{OMT}}}(Y|X = \mathbf{x})\|\}.$$

*Proof.* Our derivation follows the same proof for Lemma 2.2 and Corollary 2.3 in Divol et al. (2025). According to the definition of Eq. 43 and Eq. 51, we have

$$d\pi_{\text{OMT}}(\mathbf{y}|\mathbf{x}) = \sum_{i,j} \tilde{\omega}_{ij}(\mathbf{x}) dp_{ij}(\mathbf{y}|\mathbf{x})$$

$$= \sum_{i,j} \tilde{\omega}_{ij}(\mathbf{x}) \exp\left(\frac{\varphi_{ij}(\mathbf{x}) + \psi_{ij}(\mathbf{y}) - \|\mathbf{x} - \mathbf{y}\|_2^2}{\varepsilon_1}\right) d\mu_{1_j}(\mathbf{y})$$

$$= \sum_{i,j} \tilde{\omega}_{ij}(\mathbf{x}) \exp\left(\frac{\langle \mathbf{x}, \mathbf{y} \rangle - \tilde{\varphi}_{ij}(\mathbf{x}) - \tilde{\psi}_{ij}(\mathbf{y})}{1/2\varepsilon_1}\right) d\mu_{1_j}(\mathbf{y})$$

where $\tilde{\varphi}_{ij}(\mathbf{x}) := 1/2(\|x\|^2 - \varphi_{ij}(\mathbf{x}))$ and $\tilde{\psi}_{ij}(\mathbf{y}) := 1/2(\|y\|^2 - \psi_{ij}(\mathbf{y}))$, which are also known as Brenier potentials.

$$
\begin{aligned}
\mathbb{E}_{\pi_{\mathrm{OMT}}(\mathbf{y}|\mathbf{x})}[e^{\langle\mathbf{v},\mathbf{y}\rangle}] &= \sum_{i,j} \tilde{\omega}_{ij}(\mathbf{x}) \exp\frac{-2\tilde{\varphi}_{ij}(\mathbf{x})}{\varepsilon_1} \int \exp\left(\frac{\langle\mathbf{x},\mathbf{y}\rangle + 1/2\varepsilon_1\langle\mathbf{v},\mathbf{y}\rangle - \tilde{\psi}_{ij}(\mathbf{y})}{1/2\varepsilon_1}\right) d\mu_{1_j}(\mathbf{y}) \\
&= \sum_{i,j} \tilde{\omega}_{ij}(\mathbf{x}) \exp\frac{-2\tilde{\varphi}_{ij}(\mathbf{x})}{\varepsilon_1} \underbrace{\int \exp\left(\frac{\langle\mathbf{x}+1/2\varepsilon_1\mathbf{v},\mathbf{y}\rangle - \tilde{\psi}_{ij}(\mathbf{y})}{1/2\varepsilon_1}\right) d\mu_{1_j}(\mathbf{y})}_{\exp\left(\frac{\tilde{\varphi}_{ij}(\mathbf{x}+1/2\varepsilon_1\mathbf{v})}{1/2\varepsilon_1}\right)} \\
&= \sum_{i,j} \tilde{\omega}_{ij}(\mathbf{x}) \exp\left(\frac{\tilde{\varphi}_{ij}(\mathbf{x}+1/2\varepsilon_1\mathbf{v}) - \tilde{\varphi}_{ij}(\mathbf{x})}{1/2\varepsilon_1}\right) \\
&= \sum_{i,j} \tilde{\omega}_{ij}(\mathbf{x}) h_{\varepsilon_{1ij}}(\mathbf{x},\mathbf{v}) = \overline{h}_{\varepsilon_1}(\mathbf{x},\mathbf{v}), \quad \forall \mathbf{v} \in \mathbb{R}^d.
\end{aligned}
$$

Based on the Definition 5, the tilted coupling measure

$$
\begin{aligned}
d\mathcal{T}_{\mathbf{v}}\pi_{\mathrm{OMT}}(\mathbf{y}|\mathbf{x}) &= \frac{1}{\overline{h}_{\varepsilon_1}(\mathbf{x},\mathbf{v})} \sum_{i,j} \tilde{\omega}_{ij}(\mathbf{x}) \exp\left(\frac{\langle\mathbf{x}+1/2\varepsilon_1\mathbf{v},\mathbf{y}\rangle - \tilde{\varphi}_{ij}(\mathbf{x}) - \tilde{\psi}_{ij}(\mathbf{y})}{1/2\varepsilon_1}\right) d\mu_{1_j}(\mathbf{y}) \\
&= \frac{1}{\overline{h}_{\varepsilon_1}(\mathbf{x},\mathbf{v})} \sum_{i,j} \tilde{\omega}_{ij}(\mathbf{x}) \frac{h_{\varepsilon_{1ij}}(\mathbf{x},\mathbf{v})}{h_{\varepsilon_{1ij}}(\mathbf{x},\mathbf{v})} \exp\left(\frac{\langle\mathbf{x}+1/2\varepsilon_1\mathbf{v},\mathbf{y}\rangle - \tilde{\varphi}_{ij}(\mathbf{x}) - \tilde{\psi}_{ij}(\mathbf{y})}{1/2\varepsilon_1}\right) d\mu_{1_j}(\mathbf{y}) \\
&= \sum_{i,j} \tilde{\omega}_{ij}(\mathbf{x}) \frac{h_{\varepsilon_{1ij}}(\mathbf{x},\mathbf{v})}{\overline{h}_{\varepsilon_1}(\mathbf{x},\mathbf{v})} \exp\left(\frac{\langle\mathbf{x}+1/2\varepsilon_1\mathbf{v},\mathbf{y}\rangle - \tilde{\varphi}_{ij}(\mathbf{x}+1/2\varepsilon_1\mathbf{v}) - \tilde{\psi}_{ij}(\mathbf{y})}{1/2\varepsilon_1}\right) d\mu_{1_j}(\mathbf{y}) \\
&= \sum_{i,j} \tilde{\omega}_{ij}(\mathbf{x}) \frac{h_{\varepsilon_{1ij}}(\mathbf{x},\mathbf{v})}{\overline{h}_{\varepsilon_1}(\mathbf{x},\mathbf{v})} dp_{ij}(\mathbf{y}|\mathbf{x}+1/2\varepsilon_1\mathbf{v}) \\
&= \sum_{i,j} \tilde{\tilde{\omega}}_{ij}(\mathbf{x},\mathbf{v}) dp_{ij}(\mathbf{y}|\mathbf{x}+1/2\varepsilon_1\mathbf{v}).
\end{aligned}
$$
(75)
(76)

According to Definition 4, since the probability measure of any $\mathbf{x} \notin S_{c_0}$ is zero, the operator norm of the conditional covariance of tilted measure in Eq. 76 is also bounded as follows.

$$
\|\mathrm{Cov}_{\mathcal{T}_{\mathbf{v}}\pi_{\mathrm{OMT}}}(Y|X = \mathbf{x}+1/2\varepsilon_1\mathbf{v})\|_{\mathrm{op}} \leq \sup_{\mathbf{z}\in S_c}\{\|\mathrm{Cov}_{\pi_{\mathrm{OMT}}}(Y|X = \mathbf{z})\|\},
$$

where according to Lemma 3, supremum of the conditional covariance is the tilt stability constant, $C_{\nu_0}$. Using the law of total covariance, the conditional covariance of $\pi_{\mathrm{OMT}}(\mathbf{y}|\mathbf{x})$ can be written as:

$$
\mathrm{Cov}_{\pi_{\mathrm{OMT}}}(Y|X = \mathbf{x}) = \sum_{i,j} \tilde{\omega}_{ij}(\mathbf{x})\left(\mathrm{Cov}_{\mathbf{y}|\mathbf{x}}^{ij} + \mathbf{m}_{\mathbf{y}|\mathbf{x}}^{ij}\mathbf{m}_{\mathbf{y}|\mathbf{x}}^{ij^T}\right) - \bar{\mathbf{m}}_{\mathbf{y}|\mathbf{x}}\bar{\mathbf{m}}_{\mathbf{y}|\mathbf{x}}^T,
$$
(77)

where the overall conditional mean is

$$
\bar{\mathbf{m}}_{\mathbf{y}|\mathbf{x}} := \sum_{i,j} \tilde{\omega}_{ij}(\mathbf{x})\mathbf{m}_{\mathbf{y}|\mathbf{x}}^{ij}.
$$

To bound the total conditional covariance, we first take the norm on both sides and use the triangle inequality. This gives

$$\|\text{Cov}_{\pi_{\text{OMT}}}(Y|X=\mathbf{x})\| = \|\sum_{i,j} \tilde{\omega}_{ij}(\mathbf{x})\left(\text{Cov}_{\mathbf{y}|\mathbf{x}}^{ij} + \mathbf{m}_{\mathbf{y}|\mathbf{x}}^{ij}\mathbf{m}_{\mathbf{y}|\mathbf{x}}^{ij}{}^T\right) - \bar{\mathbf{m}}_{\mathbf{y}|\mathbf{x}}\bar{\mathbf{m}}_{\mathbf{y}|\mathbf{x}}^T\|$$

$$\leq \|\sum_{i,j} \tilde{\omega}_{ij}(\mathbf{x})\text{Cov}_{\mathbf{y}|\mathbf{x}}^{ij}\| + \|\sum_{i,j} \tilde{\omega}_{ij}(\mathbf{x})\mathbf{m}_{\mathbf{y}|\mathbf{x}}^{ij}\mathbf{m}_{\mathbf{y}|\mathbf{x}}^{ij}{}^T\| + \|\bar{\mathbf{m}}_{\mathbf{y}|\mathbf{x}}\|^2$$

$$\leq \sum_{i,j} \tilde{\omega}_{ij}(\mathbf{x})\|\text{Cov}_{\mathbf{y}|\mathbf{x}}^{ij}\| + \sum_{i,j} \tilde{\omega}_{ij}(\mathbf{x})\|\mathbf{m}_{\mathbf{y}|\mathbf{x}}^{ij}\|^2 + \|\bar{\mathbf{m}}_{\mathbf{y}|\mathbf{x}}\|^2$$

$$\leq \sum_{i,j} \tilde{\omega}_{ij}(\mathbf{x})\left(\|\text{Cov}_{\mathbf{y}|\mathbf{x}}^{ij}\| + 2\|\mathbf{m}_{\mathbf{y}|\mathbf{x}}^{ij}\|^2\right)$$

$$\leq \sum_{i,j} \tilde{\omega}_{ij}(\mathbf{x})\left(\|\text{Cov}_{\mathbf{y}|\mathbf{x}}^{ij}\| + 2\|\mathbf{m}_{j_{\mathbf{y}}}\|^2 + 2\|\Sigma_{ij}^{\varepsilon_1}{}^T\Sigma_{i_{\mathbf{xx}}}^{-1}(\mathbf{x}-\mathbf{m}_{i_{\mathbf{x}}})\|\right)$$

Taking the supremum over $\mathbf{x} \in S_{c_0}$, we define an upper bound on the total conditional covariance:

$$\sup_{\mathbf{x}\in S_{c_0}} \|\text{Cov}_{\pi_{\text{OMT}}}(Y|X=\mathbf{x})\| = \sup_{\mathbf{x}\in S_{S_{c_0}}} \{\sum_{i,j}\tilde{\omega}_{ij}(\mathbf{x})\left(\|\text{Cov}_{\mathbf{y}|\mathbf{x}}^{ij}\| + 2\|\mathbf{m}_{j_{\mathbf{y}}}\|^2 + 2\|\Sigma_{ij}^{\varepsilon_1}{}^T\Sigma_{i_{\mathbf{xx}}}^{-1}(\mathbf{x}-\mathbf{m}_{i_{\mathbf{x}}})\|\right)\}$$

$$= \|\Sigma_{r_{\mathbf{yy}}} - \Sigma_{lr}^{\varepsilon_1}{}^T\Sigma_{l_{\mathbf{xx}}}^{-1}\Sigma_{lr}^{\varepsilon_1}\| + 2\|\mathbf{m}_{r_{\mathbf{y}}}\|^2 + 2\zeta_l(c_0)\|\Sigma_{lr}^{\varepsilon_1}{}^T\Sigma_{l_{\mathbf{xx}}}^{-1}\| \quad (78)$$

where

$$l,r := \arg\max_{i,j}\{\|\text{Cov}_{\mathbf{y}|\mathbf{x}}^{ij}\| + 2\|\mathbf{m}_{j_{\mathbf{y}}}\|^2 + 2\zeta_i(c_0)\|\Sigma_{ij}^{\varepsilon_1}{}^T\Sigma_{i_{\mathbf{xx}}}^{-1}\|\}.$$

Then, based on Eq. 78, the tilt stability constant, which is defined as the supremum of the norm of the conditional covariance, can be expressed as

$$C_{\nu_0} = \|\Sigma_{r_{\mathbf{yy}}} - \Sigma_{lr}^{\varepsilon_1}{}^T\Sigma_{l_{\mathbf{xx}}}^{-1}\Sigma_{lr}^{\varepsilon_1}\| + 2\|\mathbf{m}_{r_{\mathbf{y}}}\|^2 + 2\zeta_l(c_0)\|\Sigma_{lr}^{\varepsilon_1}{}^T\Sigma_{l_{\mathbf{xx}}}^{-1}\|. \quad (79)$$

$\square$

**Lemma 4** (Lemma 3.21, Bauerschmidt et al. (2024)). *Let $\nu$ be a probability measure on $\mathbb{R}^d$ with finite second moment. Suppose for every $\mathbf{x} \in \mathbb{R}^d$, $\nu$ is tilt-stable with constant $\alpha$. Then for any probability measure $\rho$ on $\mathbb{R}^d$ with finite second moment,*

$$\|\mathbb{E}_\rho[\mathbf{x}] - \mathbb{E}_\nu[\mathbf{x}]\|^2 \leq 2\alpha D_{KL}(\rho\|\nu).$$

**Corollary 3.** *Let $T_{\text{OMT}}^{\nu_0\to\nu_1}(\mathbf{x})$ denote the OMT map between the probability measures $\nu_0(\mathbf{x}, c_0) \in G_{K_0}(\mathbb{R}^d)$ and $\nu_1(\mathbf{y}, c_1) \in G_{K_1}(\mathbb{R}^d)$, for all $\mathbf{x} \in S_{c_0}$. Then, for any probability measure $\rho(\mathbf{z}, c_0) \in G_{K_0}(\mathbb{R}^d)$, with support $\mathbf{z} \in S_{c_0}$, the following bound holds:*

$$\mathbb{E}_{\pi_{\text{OMT}}(\mathbf{x},\mathbf{z})}[\|T_{\text{OMT}}^{\rho\to\nu_1}(\mathbf{z}) - T_{\text{OMT}}^{\nu_0\to\nu_1}(\mathbf{x})\|] \leq 8\varepsilon_1^{-1}C_{\nu_0}\mathcal{W}_2(\nu_0, \rho) + 2\sqrt{C_{\nu_0}}M_{\nu_1}$$

*where*

$$M_{\nu_1} = \sup_{\mathbf{y}}\{\sum_{i,j,r,k} \tilde{\omega}_{ij}(\mathbf{x})\tilde{\omega}_{rk}(\mathbf{z})|D_{ij}^2(\mathbf{y}) - D_{rk}^2(\mathbf{y})|\},$$

*and $D_{ij}$ and $D_{rk}$ are Mahalanobis distances, defined as follows.*

$$D_{ij}^2 = (\mathbf{y} - \mathbf{m}_{j_{\mathbf{y}}})^T(\frac{1}{2}\Sigma_{j_{\mathbf{yy}}}^{-1} + \varepsilon^{-1}V_{ij})(\mathbf{y} - \mathbf{m}_{j_{\mathbf{y}}}), \quad \forall i \in [0, K_0], \forall j \in [0, K_1]$$

$$V_{ij} = \left(\Sigma_{ij}^{\varepsilon_1} + \varepsilon_1\mathbf{I}_d\right)^{-1}\Sigma_{i_{\mathbf{xx}}} - \mathbf{I}_d.$$

*Proof.* Given $\pi_{\text{OMT}}(\mathbf{y}|\mathbf{x}) = \sum_{i,j}\tilde{\omega}_{ij}(\mathbf{x})dp_{ij}(\mathbf{y}|\mathbf{x})$ and $\pi_{\text{OMT}}(\mathbf{y}|\mathbf{z}) = \sum_{r,k}\tilde{\omega}_{rk}(\mathbf{z})dp_{rk}(\mathbf{y}|\mathbf{z})$, we have

$$D_{KL}(\pi_{\text{OMT}}(\mathbf{y}|\mathbf{z})\|\pi_{\text{OMT}}(\mathbf{y}|\mathbf{x})) \leq \sum_{i,j}\sum_{r,k}\tilde{\omega}_{ij}(\mathbf{x})\tilde{\omega}_{rk}(\mathbf{z})D_{KL}(p_{rk}(\mathbf{y}|\mathbf{z})\|p_{ij}(\mathbf{y}|\mathbf{x})) \quad (80)$$

$$D_{KL}(p_{rk}(\mathbf{y}|\mathbf{z})\|p_{ij}(\mathbf{y}|\mathbf{x})) = \int \log \frac{\exp\left(\frac{\langle \mathbf{z}, \mathbf{y}\rangle - \tilde{\varphi}_{rk}(\mathbf{z}) - \tilde{\psi}_{rk}(\mathbf{y})}{1/2\varepsilon_1}\right)\mu_{1_k}(\mathbf{y})d\mathbf{y}}{\exp\left(\frac{\langle \mathbf{x}, \mathbf{y}\rangle - \tilde{\varphi}_{ij}(\mathbf{x}) - \tilde{\psi}_{ij}(\mathbf{y})}{1/2\varepsilon_1}\right)\mu_{1_j}(\mathbf{y})d\mathbf{y}} dp_{rk}(\mathbf{y}|\mathbf{z})$$

$$= \int 2\varepsilon_1^{-1}\left(\langle \mathbf{z}-\mathbf{x}, \mathbf{y}\rangle + \tilde{\varphi}_{ij}(\mathbf{x}) - \tilde{\varphi}_{rk}(\mathbf{z}) + \tilde{\psi}_{ij}(\mathbf{y}) - \tilde{\psi}_{rk}(\mathbf{y})\right)dp_{rk}(\mathbf{y}|\mathbf{z}) + \int \left(\log \mu_{1_k}(\mathbf{y}) - \log \mu_{1_j}(\mathbf{y})\right)dp_{rk}(\mathbf{y}|\mathbf{z})$$

$$= 2\varepsilon_1^{-1}\left(\langle \mathbf{z}-\mathbf{x}, T_{rk}(\mathbf{z})\rangle + \tilde{\varphi}_{ij}(\mathbf{x}) - \tilde{\varphi}_{rk}(\mathbf{z})\right) + \int \left(2\varepsilon^{-1}\tilde{\psi}_{ij}(\mathbf{y}) - \log \mu_{1_j}(\mathbf{y}) - 2\varepsilon^{-1}\tilde{\psi}_{rk}(\mathbf{y}) + \log \mu_{1_k}(\mathbf{y})\right)dp_{rk}(\mathbf{y}|\mathbf{z})$$

$$(81)$$

Similarly, the reverse direction of the KL divergence calculation can be expressed as

$$D_{KL}(p_{ij}(\mathbf{y}|\mathbf{x})\|p_{rk}(\mathbf{y}|\mathbf{z})) = 2\varepsilon_1^{-1}\left(\langle \mathbf{x}-\mathbf{z}, T_{ij}(\mathbf{x})\rangle + \tilde{\varphi}_{rk}(\mathbf{z}) - \tilde{\varphi}_{ij}(\mathbf{x})\right) +$$
$$\int \left(2\varepsilon^{-1}\tilde{\psi}_{rk}(\mathbf{y}) - \log \mu_{1_k}(\mathbf{y}) - 2\varepsilon^{-1}\tilde{\psi}_{ij}(\mathbf{y}) + \log \mu_{1_j}(\mathbf{y})\right)dp_{ij}(\mathbf{y}|\mathbf{x})$$

$$(82)$$

If $\forall j \in [1, K_1]$, $\mu_{1_j}(\mathbf{y}) = \mathcal{N}(\mathbf{y}|\mathbf{m}_{j_\mathbf{y}}, \Sigma_{j_{\mathbf{yy}}})$, it was shown in Janati et al. (2020) that $\varphi_{ij}$ and $\psi_{ij}$ admit closed-form solutions in the form of quadratic functions as discussed in Eq. 40. Using the closed-form for potential functions and the portability measures, we have

$$G_{ij}(\mathbf{y}) := \varepsilon_1^{-1}\psi_{ij}(\mathbf{y}) + \log \mu_{1_j}(\mathbf{y})$$

$$= \varepsilon_1^{-1}\left(-(\mathbf{y}-\mathbf{m}_{j_\mathbf{y}})^T V_{ij}(\mathbf{y}-\mathbf{m}_{j_\mathbf{y}})\right) - \frac{1}{2}\left((\mathbf{y}-\mathbf{m}_{j_\mathbf{y}})^T\Sigma_{j_{\mathbf{yy}}}^{-1}(\mathbf{y}-\mathbf{m}_{j_\mathbf{y}}) + \log (2\pi)^d|\Sigma_{j_{\mathbf{yy}}}|\right)$$

$$= -(\mathbf{y}-\mathbf{m}_{j_\mathbf{y}})^T(\frac{1}{2}\Sigma_{j_{\mathbf{yy}}}^{-1} + \varepsilon^{-1}V_{ij})(\mathbf{y}-\mathbf{m}_{j_\mathbf{y}}) - \frac{1}{2}\log (2\pi)^d|\Sigma_{j_{\mathbf{yy}}}| \qquad (83)$$

$$G_{rk}(\mathbf{y}) := \varepsilon_1^{-1}\psi_{rk}(\mathbf{y}) + \log \mu_{1_k}(\mathbf{y})$$

$$= -(\mathbf{y}-\mathbf{m}_{k_\mathbf{y}})^T(\frac{1}{2}\Sigma_{k_{\mathbf{yy}}}^{-1} + \varepsilon^{-1}V_{rk})(\mathbf{y}-\mathbf{m}_{k_\mathbf{y}}) - \frac{1}{2}\log (2\pi)^d|\Sigma_{k_{\mathbf{yy}}}| \qquad (84)$$

Using definitions of $G_{ij}, G_{rk}$ in Eqs. 83 and 84, we have

$$2\varepsilon_1^{-1}\tilde{\psi}_{rk}(\mathbf{y}) - 2\varepsilon_1^{-1}\tilde{\psi}_{ij}(\mathbf{y}) + \log \mu_{1_j}(\mathbf{y}) - \log \mu_{1_k}(\mathbf{y}) = \varepsilon_1^{-1}\psi_{ij}(\mathbf{y}) - \varepsilon_1^{-1}\psi_{rk}(\mathbf{y}) + \log \mu_{1_j}(\mathbf{y}) - \log \mu_{1_k}(\mathbf{y})$$
$$= G_{ij}(\mathbf{y}) - G_{rk}(\mathbf{y})$$

$$= (\mathbf{y}-\mathbf{m}_{k_\mathbf{y}})^T(\frac{1}{2}\Sigma_{k_{\mathbf{yy}}}^{-1} + \varepsilon^{-1}V_{rk})(\mathbf{y}-\mathbf{m}_{k_\mathbf{y}}) + \frac{1}{2}\log \frac{|\Sigma_{k_{\mathbf{yy}}}|}{|\Sigma_{j_{\mathbf{yy}}}|} - (\mathbf{y}-\mathbf{m}_{j_\mathbf{y}})^T(\frac{1}{2}\Sigma_{j_{\mathbf{yy}}}^{-1} + \varepsilon^{-1}V_{ij})(\mathbf{y}-\mathbf{m}_{j_\mathbf{y}})$$

$$= D_{rk}^2(\mathbf{y}) - D_{ij}^2(\mathbf{y}) + a_{jk}$$

where $D_{rk}^2(\mathbf{y})$ and $D_{ij}^2(\mathbf{y})$ denote the Mahalanobis distance. Adding the two KL divergence terms in Eq. 81 and 82, we obtain

$$D_{KL}(p_{rk}(\mathbf{y}|\mathbf{z})\|p_{ij}(\mathbf{y}|\mathbf{x})) + D_{KL}(p_{ij}(\mathbf{y}|\mathbf{x})\|p_{rk}(\mathbf{y}|\mathbf{z})) = 2\varepsilon_1^{-1}\langle \mathbf{x}-\mathbf{z}, T_{ij}(\mathbf{x}) - T_{rk}(\mathbf{z})\rangle +$$
$$\mathbb{E}_{p_{ij}(\mathbf{y}|\mathbf{x})}[D_{rk}^2(\mathbf{y}) - D_{ij}^2(\mathbf{y}) + a_{jk}] - \mathbb{E}_{p_{rk}(\mathbf{y}|\mathbf{z})}[D_{rk}^2(\mathbf{y}) - D_{ij}^2(\mathbf{y}) + a_{jk}]$$

$$= 2\varepsilon_1^{-1}\langle \mathbf{x}-\mathbf{z}, T_{ij}(\mathbf{x}) - T_{rk}(\mathbf{z})\rangle + \mathbb{E}_{p_{ij}(\mathbf{y}|\mathbf{x})}[D_{rk}^2(\mathbf{y}) - D_{ij}^2(\mathbf{y})] - \mathbb{E}_{p_{rk}(\mathbf{y}|\mathbf{z})}[D_{rk}^2(\mathbf{y}) - D_{ij}^2(\mathbf{y})]$$

$$\leq 2\varepsilon_1^{-1}\langle \mathbf{x}-\mathbf{z}, T_{ij}(\mathbf{x}) - T_{rk}(\mathbf{z})\rangle + |\mathbb{E}_{p_{ij}(\mathbf{y}|\mathbf{x})}[D_{rk}^2(\mathbf{y}) - D_{ij}^2(\mathbf{y})] - \mathbb{E}_{p_{rk}(\mathbf{y}|\mathbf{z})}[D_{rk}^2(\mathbf{y}) - D_{ij}^2(\mathbf{y})]|$$

$$\leq 2\varepsilon_1^{-1}\langle \mathbf{x}-\mathbf{z}, T_{ij}(\mathbf{x}) - T_{rk}(\mathbf{z})\rangle + \int |D_{rk}^2(\mathbf{y}) - D_{ij}^2(\mathbf{y})|\, |p_{ij}(\mathbf{y}|\mathbf{x}) - p_{rk}(\mathbf{y}|\mathbf{z})|d(\mathbf{y})$$

$$(85)$$

Applying Pinsker's inequality, we have

$$\int |D_{rk}^2(\mathbf{y}) - D_{ij}^2(\mathbf{y})|\, |p_{ij}(\mathbf{y}|\mathbf{x}) - p_{rk}(\mathbf{y}|\mathbf{z})|d(\mathbf{y}) \leq M_{ijrk}\sqrt{2D_{KL}(dp_{ij}(\mathbf{y}|\mathbf{x})\|dp_{rk}(\mathbf{y}|\mathbf{z}))} \quad (86)$$

where $M_{ijrk} = \sup_{\mathbf{y} \in S_{c_1}} \{|D_{rk}^2(\mathbf{y}) - D_{ij}^2(\mathbf{y})|\}$. Using the upper bound in Eq. 86 in Eq. 85, we have

$$D_{KL}(p_{ij}(\mathbf{y}|\mathbf{x})\|p_{rk}(\mathbf{y}|\mathbf{z})) \leq 2\varepsilon_1^{-1}\langle \mathbf{x}-\mathbf{z}, T_{ij}(\mathbf{x}) - T_{rk}(\mathbf{z})\rangle + M_{ijrk}\sqrt{2D_{KL}(dp_{ij}(\mathbf{y}|\mathbf{x})\|dp_{rk}(\mathbf{y}|\mathbf{z}))}$$

$$\mathbb{E}_{\pi_{\text{OMT}}(\mathbf{x},\mathbf{z})}[D_{KL}(p_{ij}(\mathbf{y}|\mathbf{x})\|p_{rk}(\mathbf{y}|\mathbf{z}))] \leq 2\varepsilon_1^{-1} \int\int \langle \mathbf{x} - \mathbf{z}, T_{ij}(\mathbf{x}) - T_{rk}(\mathbf{z})\rangle d\pi_{\text{OMT}}(\mathbf{x},\mathbf{z})+$$

$$M_{ijrk}\mathbb{E}_{d\pi_{\text{OMT}}(\mathbf{x},\mathbf{z})}[\sqrt{2D_{KL}(dp_{ij}(\mathbf{y}|\mathbf{x})\|dp_{rk}(\mathbf{y}|\mathbf{z}))}]$$

$$\leq 2\varepsilon_1^{-1}\mathbb{E}_{\pi_{\text{OMT}}(\mathbf{x},\mathbf{z})}[\|T_{ij}(\mathbf{x}) - T_{rk}(\mathbf{z})\|]\,\mathcal{W}_2(\nu_0,\rho)+$$

$$M_{ijrk}\mathbb{E}_{d\pi_{\text{OMT}}(\mathbf{x},\mathbf{z})}[\sqrt{2D_{KL}(dp_{ij}(\mathbf{y}|\mathbf{x})\|dp_{rk}(\mathbf{y}|\mathbf{z}))}] \quad (87)$$

which can be expressed as follows (see Section A.1.3 for the full derivation).

$$\mathbb{E}_{\pi_{\text{OMT}}(\mathbf{x},\mathbf{z})}[D_{KL}(p_{ij}(\mathbf{y}|\mathbf{x})\|p_{rk}(\mathbf{y}|\mathbf{z}))] \leq 2M_{ijrk}^2 + 4\varepsilon_1^{-1}\mathbb{E}_{\pi_{\text{OMT}}(\mathbf{x},\mathbf{z})}[\|T_{ij}(\mathbf{x}) - T_{rk}(\mathbf{z})\|]\,\mathcal{W}_2(\nu_0,\rho)$$
$$(88)$$

Moreover,

$$\int\int D_{KL}(\pi_{\text{OMT}}(\mathbf{y}|\mathbf{z})\|\pi_{\text{OMT}}(\mathbf{y}|\mathbf{x}))d\pi_{\text{OMT}}(\mathbf{x},\mathbf{z}) \leq \sum_{i,j,r,k}\tilde{\omega}_{ij}(\mathbf{x})\tilde{\omega}_{rk}(\mathbf{y})\mathbb{E}_{\pi_{\text{OMT}}(\mathbf{x},\mathbf{z})}[D_{KL}(p_{ij}(\mathbf{y}|\mathbf{x})\|p_{rk}(\mathbf{y}|\mathbf{z}))]$$
$$(89)$$

From the tilt-stability condition applied to each component pair, we further obtain

$$\frac{1}{2}C_{\mu_{ij}}^{-1}\|T_{pr}(\mathbf{z}) - T_{ij}(\mathbf{x})\|^2 \leq D_{KL}(p_{rk}(\mathbf{y}|\mathbf{z})\|p_{ij}(\mathbf{y}|\mathbf{x}))$$

$$\frac{1}{2}C_{\mu_{ij}}^{-1}\mathbb{E}_{\pi_{\text{OMT}}(\mathbf{x},\mathbf{z})}[\|T_{rk}(\mathbf{z}) - T_{ij}(\mathbf{x})\|^2] \leq \mathbb{E}_{\pi_{\text{OMT}}(\mathbf{x},\mathbf{z})}[D_{KL}(p_{ij}(\mathbf{y}|\mathbf{x})\|p_{rk}(\mathbf{y}|\mathbf{z}))]$$

$$\frac{1}{2}C_{\mu_{ij}}^{-1}\left(\mathbb{E}_{\pi_{\text{OMT}}(\mathbf{x},\mathbf{z})}[\|T_{rk}(\mathbf{z}) - T_{ij}(\mathbf{x})\|]\right)^2 \leq \mathbb{E}_{\pi_{\text{OMT}}(\mathbf{x},\mathbf{z})}[D_{KL}(p_{ij}(\mathbf{y}|\mathbf{x})\|p_{rk}(\mathbf{y}|\mathbf{z}))] \quad (90)$$

where $C_{\mu_{ij}} = \sup_{\mathbf{z}\in S_{c_0}}\{\|\text{Cov}_{\mathbf{y}|\mathbf{z}}^{ij}\|\}$, as defined in Eq. (64).

Using the upper bound in Eq. 88 in Eq. 90, we have

$$\frac{1}{2}C_{\mu_{ij}}^{-1}\left(\mathbb{E}_{\pi_{\text{OMT}}(\mathbf{x},\mathbf{z})}[\|T_{rk}(\mathbf{z}) - T_{ij}(\mathbf{x})\|]\right)^2 \leq 2M_{ijrk}^2 + 4\varepsilon_1^{-1}\mathbb{E}_{\pi_{\text{OMT}}(\mathbf{x},\mathbf{z})}[\|T_{ij}(\mathbf{x}) - T_{rk}(\mathbf{z})\|]\,\mathcal{W}_2(\nu_0,\rho)$$
$$(91)$$

This can be simplified as follow (see Section A.1.3 for the full derivation).

$$\mathbb{E}_{\pi_{\text{OMT}}(\mathbf{x},\mathbf{z})}[\|T_{rk}(\mathbf{z}) - T_{ij}(\mathbf{x})\|] \leq 8\varepsilon_1^{-1}C_{\mu_{ij}}\mathcal{W}_2(\nu_0,\rho) + 2\sqrt{C_{\mu_{ij}}}M_{ijrk} \quad (92)$$

Aggregating over all component combinations, we obtain

$$\sum_{i,j,r,k}\tilde{\omega}_{ij}(\mathbf{x})\tilde{\omega}_{rk}(\mathbf{z})\mathbb{E}_{\pi_{\text{OMT}}(\mathbf{x},\mathbf{z})}[\|T_{rk}(\mathbf{z}) - T_{ij}(\mathbf{x})\|^2] \leq \sum_{i,j,r,k}\tilde{\omega}_{ij}(\mathbf{x})\tilde{\omega}_{rk}(\mathbf{z})\left(8\varepsilon_1^{-1}C_{\mu_{ij}}\mathcal{W}_2(\nu_0,\rho) + 2\sqrt{C_{\mu_{ij}}}M_{ijrk}\right)$$

Since $C_{\mu_{ij}} \leq C_{\nu_0}$, it follows that

$$\sum_{i,j,r,k}\tilde{\omega}_{ij}(\mathbf{x})\tilde{\omega}_{rk}(\mathbf{z})\mathbb{E}_{\pi_{\text{OMT}}(\mathbf{x},\mathbf{z})}[\|T_{rk}(\mathbf{z}) - T_{ij}(\mathbf{x})\|^2] \leq 8\varepsilon_1^{-1}C_{\nu_0}\mathcal{W}_2(\nu_0,\rho) + 2\sqrt{C_{\nu_0}}\underbrace{\sum_{i,j,r,k}\tilde{\omega}_{ij}(\mathbf{x})\tilde{\omega}_{rk}(\mathbf{z})M_{ijrk}}_{M_{\nu_1}}$$
$$(93)$$

Using the triangle inequality, we can relate the expected difference between the overall transport maps:

$$\mathbb{E}_{\pi_{\text{OMT}}(\mathbf{x},\mathbf{z})}[\|T_{\text{OMT}}^{\rho\to\nu_1}(\mathbf{z}) - T_{\text{OMT}}^{\nu_0\to\nu_1}(\mathbf{x})\|] = \mathbb{E}_{\pi_{\text{OMT}}(\mathbf{x},\mathbf{z})}[\|\sum_{i,j,r,k}\tilde{\omega}_{ij}(\mathbf{x})\tilde{\omega}_{rk}(\mathbf{z})(T_{ij}(\mathbf{x}) - T_{rk}(\mathbf{z}))\|]$$

$$\leq \mathbb{E}_{\pi_{\text{OMT}}(\mathbf{x},\mathbf{z})}[\sum_{i,j,r,k}\tilde{\omega}_{ij}(\mathbf{x})\tilde{\omega}_{rk}(\mathbf{z})\|T_{ij}(\mathbf{x}) - T_{rk}(\mathbf{z})\|] \quad (94)$$

Combining the upper and lower bounds from Eq. 93 and Eq. 94, we obtain the desired inequality:

$$\mathbb{E}_{\pi_{\text{OMT}}(\mathbf{x},\mathbf{z})}[\|T_{\text{OMT}}^{\rho\to\nu_1}(\mathbf{z}) - T_{\text{OMT}}^{\nu_0\to\nu_1}(\mathbf{x})\|] \leq 8\varepsilon_1^{-1}C_{\nu_0}\mathcal{W}_2(\nu_0,\rho) + 2\sqrt{C_{\nu_0}}M_{\nu_1}\,.$$

$$\square$$

**Theorem 2** (Stability of OMT under perturbation). *Consider the restriction of $\nu_0(\mathbf{x}, c), \nu_0'(\mathbf{x}', c) \in G_{K_0}(\mathbb{R}^d)$ to $\mathbf{x}, \mathbf{x}' \in S_c$. The following stability bound holds:*

$$\mathbf{E}_{\nu_0}\left[\|T_{\text{OMT}}^{\nu_0 \to \nu_1}(\mathbf{x}) - T_{\text{OMT}}^{\nu_0' \to \nu_1}(\mathbf{x})\|\right] \leq (8\varepsilon^{-1}C_{\nu_0'} + L_{\nu_0'})\mathcal{W}_2(\nu_0, \nu_0') + 2\sqrt{C_{\nu_0'}}M_{\nu_1},$$

*where $T_{\text{OMT}}^{\nu_0 \to \nu_1}$ denotes the OMT map from $\nu_0$ to $\nu_1$.*

*Proof.* Let $(\Omega', P')$ denote the transport weights and the set of Gaussian couplings associated with the OMT between $\nu_0$ and $\nu_0'$, where $\nu_0(\mathbf{x}), \nu_0'(\mathbf{x}')$ are doubly truncated GMM, and $\nu_0'$ denotes the perturbed counterpart of $\nu_0$, for all $\mathbf{x}, \mathbf{x}' \in S_c$, where $S_c \in \mathbb{R}^+$ is the truncation set.

For brevity, we define $T_{\text{OMT}}^{\nu_0 \to \nu_1} := T_{\text{OMT}}^{\nu_0}$. We begin by quantifying the average norm of deviation between the two optimal transport maps:

$$\int \|T_{\text{OMT}}^{\nu_0}(\mathbf{x}) - T_{\text{OMT}}^{\nu_0'}(\mathbf{x})\| d\nu_0(\mathbf{x}) = \iint \|T_{\text{OMT}}^{\nu_0}(\mathbf{x}) - T_{\text{OMT}}^{\nu_0'}(\mathbf{x})\| d\pi'(\mathbf{x}, \mathbf{x}')$$

$$\leq \iint \|T_{\text{OMT}}^{\nu_0}(\mathbf{x}) - T_{\text{OMT}}^{\nu_0'}(\mathbf{x}')\| d\pi'(\mathbf{x}, \mathbf{x}')$$

$$+ \iint \|T_{\text{OMT}}^{\nu_0'}(\mathbf{x}) - T_{\text{OMT}}^{\nu_0'}(\mathbf{x}')\| d\pi'(\mathbf{x}, \mathbf{x}') .$$

According to Lemma 2, $T_{\text{OMT}}^{\nu_0'}$ is $L_{\nu_0'}$-Lipschitz for all $\mathbf{x} \in S$ (see Eq. 73), which follows that

$$\iint \|T_{\text{OMT}}^{\nu_0'}(\mathbf{x}) - T_{\text{OMT}}^{\nu_0'}(\mathbf{x}')\| d\pi'(\mathbf{x}, \mathbf{x}') \leq L_{\nu_0'} \iint \|\mathbf{x} - \mathbf{x}'\| d\pi'(\mathbf{x}, \mathbf{x}')$$

$$\leq L_{\nu_0'} \mathcal{W}_2(\nu_0, \nu_0') .$$

Using the results of corollary 3, we have

$$\mathbb{E}_{\pi'}\left[\|T_{\text{OMT}}^{\nu_0}(\mathbf{x}) - T_{\text{OMT}}^{\nu_0'}(\mathbf{x}')\|\right] \leq 8\varepsilon^{-1}C_{\nu_0'}\mathcal{W}_2(\nu_0, \nu_0') + 2\sqrt{C_{\nu_0'}}M_{\nu_1}$$

Finally, combining the bounds derived above, we obtain

$$\int \|T_{\text{OMT}}^{\nu_0}(\mathbf{x}) - T_{\text{OMT}}^{\nu_0'}(\mathbf{x})\| d\nu_0(\mathbf{x}) \leq (8\varepsilon^{-1}C_{\nu_0'} + L_{\nu_0'})\mathcal{W}_2(\nu_0, \nu_0') + 2\sqrt{C_{\nu_0'}}M_{\nu_1}$$

$\square$

### A.1.3 ADDITIONAL DERIVATIONS

$$u := \mathbb{E}_{\pi_{\text{OMT}}(\mathbf{x}, \mathbf{z})}[D_{KL}(p_{ij}(\mathbf{y}|\mathbf{x})\|p_{rk}(\mathbf{y}|\mathbf{z}))]$$
$$a := 2\varepsilon_1^{-1}\mathbb{E}_{\pi_{\text{OMT}}(\mathbf{x}, \mathbf{z})}[\|T_{ij}(\mathbf{x}) - T_{rk}(\mathbf{z})\|]\,\mathcal{W}_2(\nu_0, \rho)$$
$$b := M_{ijrk},$$

where $u, a, b \geq 0$.

The function $f(x) = \sqrt{u}$ is concave. By Jensen's inequality, we have $\mathbb{E}[\sqrt{(.)}] \leq \sqrt{E(.)}$ . Then, Eq. 87 can be written as

$$u \leq a + b\sqrt{2u} \tag{95}$$

Letting $v^2 = u$, we obtain the quadratic equation $v^2 - (\sqrt{2}b)v - a = 0$. Using the quadratic formula and noting that $v \geq 0$, the valid solution is

$$v = \frac{\sqrt{2}b + \sqrt{2b^2 + 4a}}{2}.$$

To preserve the inequity in Eq. 95, we have

$$0 \leq v \leq \frac{\sqrt{2}b + \sqrt{2b^2 + 4a}}{2} .$$

where

$$u \leq b^2 + a + b\sqrt{b^2 + 2a} \, .$$

Since

$$b\sqrt{b^2 + 2a} \leq b^2 + a,$$

To simplify the upper bound, we consider a looser as follows.

$$u \leq 2(b^2 + a)$$

which is equivalent to

$$\mathbb{E}_{\pi_{\text{OMT}}(\mathbf{x},\mathbf{z})}[D_{KL}(p_{ij}(\mathbf{y}|\mathbf{x})\|p_{rk}(\mathbf{y}|\mathbf{z}))] \leq 2M_{ijrk}^2 + 4\varepsilon_1^{-1}\mathbb{E}_{\pi_{\text{OMT}}(\mathbf{x},\mathbf{z})}[\|T_{ij}(\mathbf{x}) - T_{rk}(\mathbf{z})\|]\, \mathcal{W}_2(\nu_0, \rho) \, .$$

Now, let's define

$$u := \mathbb{E}_{\pi_{\text{OMT}}(\mathbf{x},\mathbf{z})}[\|T_{rk}(\mathbf{z}) - T_{ij}(\mathbf{x})\|]$$

$$b := 4\varepsilon_1^{-1}\mathcal{W}_2(\nu_0, \rho)$$

$$c := 2M_{ijrk}^2$$

$$a := \frac{1}{2}C_{\mu_{ij}}^{-1},$$

where $u, a, b, d \geq 0$. Then the Eq. 91 can be expressed as

$$au^2 \leq bu + c$$

Since $u$ is non-negative, the negative root is not a valid solution, so the upper bound is:

$$u \leq \frac{b + \sqrt{b^2 + 4ac}}{2a}.$$

Since $\sqrt{x + y} \leq \sqrt{x} + \sqrt{y}$, a looser upper bound for $u$ can be written as

$$u \leq \frac{2b + 2\sqrt{ac}}{2a} = a^{-1}b + \sqrt{a^{-1}c}.$$

Finally,

$$\mathbb{E}_{\pi_{\text{OMT}}(\mathbf{x},\mathbf{z})}[\|T_{rk}(\mathbf{z}) - T_{ij}(\mathbf{x})\|] \leq 8\varepsilon_1^{-1}C_{\mu_{ij}}\mathcal{W}_2(\nu_0, \rho) + 2\sqrt{C_{\mu_{ij}}}M_{ijrk} \, .$$

## B  W2-BENCHMARK TASK

### B.1  EXPERIMENT

For the continuous Wasserstein-2 benchmark task, we adapted the experimental setup from the publicly available repository of Korotin et al. (2021). We evaluated all models across a range of dimensions ($d$) with corresponding training sample sizes ($n$). For each configuration, performance was assessed on a separate test set of $10,000$ samples. To ensure statistical robustness, every experiment was repeated five times with different random initializations. These same settings were also used for the ablation study on the impact of noise.

The specific dimension and sample size pairs were as follows:

- for $d \in \{2, 4\}$ with $n = 10,000$,
- for $d \in \{8, 16, 32\}$ with $n = 20,000$,
- for $d \in \{8, 16, 32\}$ with $n = 20,000$,
- for $d \in \{8, 16, 32, 64\}$ with $n = 20,000$,
- for $d \in \{128, 256\}$ with $n = 50,000$.

## B.2 ABLATION STUDY

To evaluate the stability of OMT solvers, compared to top OT solvers in the benchmark task (Figure 2), we conducted an ablation study examining the sensitivity of transport maps to input perturbations. OT solvers were first optimized on noisy samples and then evaluated on the original, unperturbed test set. Noise was added to the source distribution samples during training, while the evaluation used the original test set ($N = 10,000$ samples).

Here, the source samples were subjected to two distinct forms of stochastic perturbation:

- Additive Gaussian (White) Noise: We added noise sampled from $\mathcal{N}(0, \sigma^2 I)$ to the source input, varying the standard deviation $\sigma \in \{0.1, 0.25, 0.5, 1.0\}$.
- Dropout Noise: To simulate missing data, a common issue in single-cell genomics data (e.g., gene dropout), we applied a random drop-out operation where input features were zeroed out with a probability $p \in \{0.05, 0.1, 0.25, 0.5\}$.

The performance variations shown in Figure 3 were quantified as the relative percentage difference in MSE and Sinkhorn Divergence ($D_\varepsilon$) compared to the noise-free baseline.

## B.3 TRAINING CONFIGURATIONS

We compared our OMT solver against several state-of-the-art baselines. We implemented EOT, ExNOT, and PROGOT using their official versions in the OTT-JAX toolbox (Cuturi et al., 2022), following the recommended settings from the tutorials.

- EOT: The entropy regularization was set to $\varepsilon = 0.1$, with a maximum of $10^6$ iterations.
- W2-OT: We used the recommended 3-layer MLP architectures for dual potentials and employed regression-based amortization with L-BFGS fine-tuning, training for a maximum of $10^5$ iterations (Table 4 in Amos (2022)).
- PROGOT: We used the recommended schedulers with $K = 4$ steps.
- ExNOT: We employed the recommended network architecture, using a 5-layer MLP for each potential function, with 128 nodes per hidden layer, and trained for a maximum of $10^5$ iterations.
- ENOT: We employed the original code released by the authors for this benchmark, using the same configuration as reported.
- OMT (proposed): We set the number of source components to $K = 5$ and target components to $K = 15$. Gaussian mixture models (GMMs) were fitted to the source and target data using Python's scikit-learn, employing a full covariance structure for $d \leq 64$. The model was trained for a maximum of $10^5$ iterations with $\varepsilon = 0.01$ for both entropy regularizers.
- GMM-OT: We used a training process similar to that of OMT, except that we set $\varepsilon = 0$ and employed Wasserstein minimization to optimize the mixing-component matrix ($\Omega$).

Table 5: Transportation costs for different OT solvers.

| Method | Transportation Cost |
|---|---|
| OMT (GMM-OT) | $\sum_{i,j}^{K} \omega_{ij}^* \int_{\mathcal{X} \times \mathcal{Y}} \|\mathbf{x} - \mathbf{y}\|_2^2 \, dp_{ij}^*(\mathbf{x}, \mathbf{y})$ |
| ExNOT (W2-OT) | $\int_{\mathcal{X}} f^*(\mathbf{x}) d\nu_0(\mathbf{x}) + \int_{\mathcal{Y}} g^*(\mathbf{y}) d\nu_1(\mathbf{y})$ |
| ENOT | $\int_{\mathcal{X}} \|\mathbf{x} - T^*(\mathbf{x})\|_2^2 \, d\nu_0(\mathbf{x})$ |
| PROGOT (EOT) | $\int_{\mathcal{X} \times \mathcal{Y}} \|\mathbf{x} - \mathbf{y}\|_2^2 \, d\pi^*(\mathbf{x}, \mathbf{y})$ |

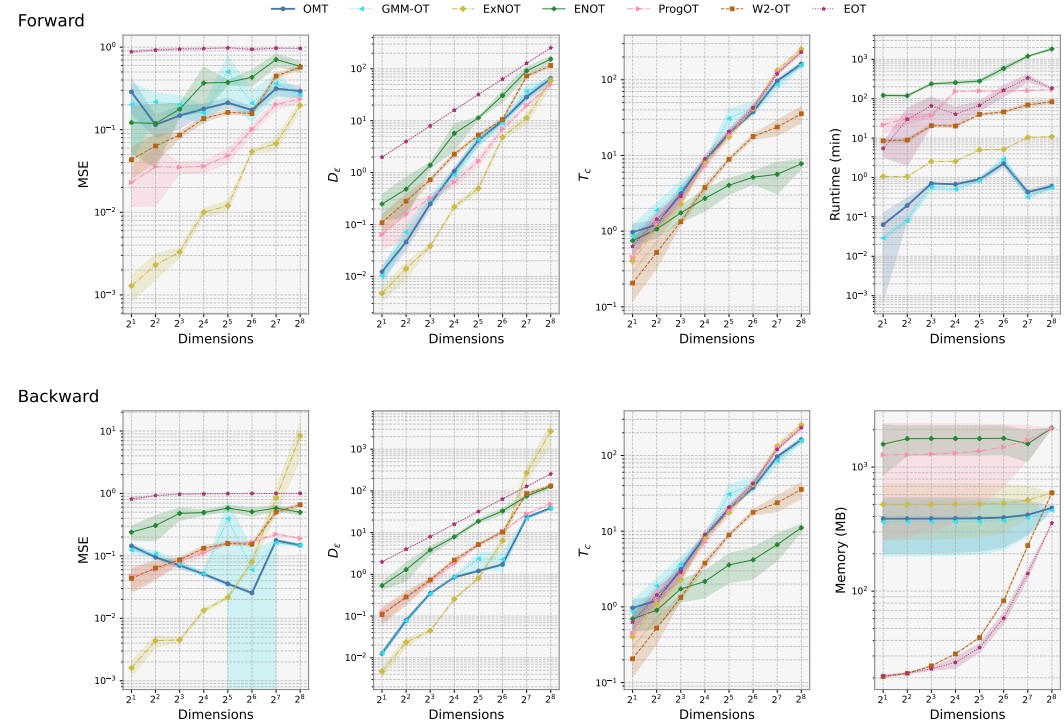

Figure 7: Comparison of OT solvers on forward and backward paths in Wasserstein-2 benchmark tasks Korotin et al. (2021). Results are computed on the test set with $10,000$ samples and averaged over five random initializations. Transport cost $T_c$ is defined in Table 5. Among the evaluated methods, PROGOT exhibits the highest computational cost among OT solvers.

## B.4 OMT VS. GMM-OT

OMT and GMM-OT ($GW_2$) (Delon & Desolneux, 2020) both operate on pairs of GMMs, but they fundamentally differ in their optimization landscapes and solution guarantees. GMM-OT fits a GMM to the coupling function to minimize the Wasserstein distance. This formulation is non-convex and may lack a unique solution, meaning multiple distinct transport plans can yield identical costs, thereby hampering interpretability. In contrast, OMT utilizes a dual-entropic regularization scheme, applying regularization to both the component transport $p_{ij}$ and the mixing weights matrix $\Omega$. This formulation offers significant computational and theoretical advantages:

- Closed-form and faster updates: The regularization on component transportation allows for analytical closed-form solutions for $p_{ij}$ (Eq. 21) and the transport potentials $\varphi_{ij}$ (Eq. 26) and $\psi_{ij}$ (Eq. 27). This simplifies the optimization, reducing the problem to optimizing only the mixing parameters, which can be solved efficiently using convex solvers including Sinkhorn algorithm.

- Uniqueness and Smoothness: The entropic term makes the objective strictly biconvex (Lemma 1). Consequently, the solution is unique (Theorem 1) and differentiable with respect to the inputs.

- Gradient Stability: Regularization discourages degenerate or overly *peaky* transport plans, preventing overfitting to empirical distributions with limited samples. This ensures stable gradients with respect to the cost and input distributions.

To empirically validate the stability offered by this regularization, we evaluate the relative changes to the transport plan under input perturbation using the $W_2$-benchmark task. We used the same source and target training samples ($\mathbf{x}_{train}, \mathbf{y}_{train}$) and identical GMMs (same number of components and parameters) for both methods. For each solver, we then optimize to obtain a baseline coupling map $\Omega$. Following the ablation study in B.2, we perturb the input data by adding Gaussian noise $\mathcal{N}(0, \sigma^2 I)$

to $\mathbf{x}_{train}$, with $\sigma \in \{0.1, 0.25, 0.5, 1.0\}$, and re-optimize the transport plans ($\tilde{\Omega}$). Here, we quantify the robustness of the methods using two metrics: (i) the percentage change in Transport Cost ($T_c$) (Table 5), and (ii) the relative Transport Map Deviation (TMD), defined as:

$$TMD := \frac{\|\tilde{\Omega} - \Omega\|_F}{\|\Omega\|_F}$$

Figure 8 shows that OMT exhibits smaller changes in the transport plan than GMM-OT, highlighting the benefit of regularization in the mixture transportation problem.

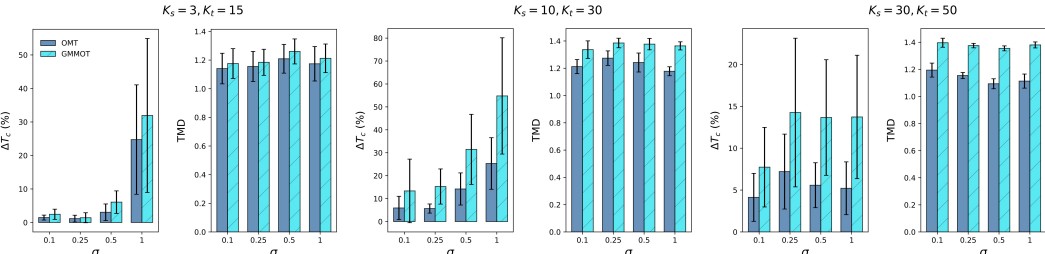

Figure 8: Comparing OMT and GMM-OT maps under input perturbation. We evaluate robustness on W2-benchmark tasks ($d = 2$) across varying noise levels $\sigma \in \{0.1, 0.25, 0.5, 1.0\}$ and increasing mixture complexities ($K_s, K_t$). To ensure a direct comparison of the optimization objectives, identical training samples and fitted GMMs were used for both methods in each setting. The bar charts report two stability metrics: changes of transport cost ($\Delta T_c$) and the relative transport map deviation (TMD). Error bars represent the standard deviation across 10 random initializations. OMT maps consistently demonstrate lower sensitivity to noise compared to GMM-OT, validating the stability benefits of the biconvex formulation.

## C  SINGLE-CELL DATA ANALYSIS

### C.1  DATA AVAILABILITY

- sci-Plex3 data can be downloaded from NCBI GEO (#GSE139944).
- MERFISH Mouse Brain Receptor data is available on the Vizgen website (https://info.vizgen.com/mouse-brain-map). Map data release (https://info.vizgen.com/mouse-brain-map)
- Mouse developmental data is available through Neuroscience Multi-omic Data Archive (NeMO), (RRID:SCR-016152). The 10x scRNA-seq dataset is available at https://assets.nemoarchive.org/dat-0oyried.
- Mouse ageing scRNA-seq is also available through NeMO, https://nemoarchive.org/, and can be accessed at https://assets.nemoarchive.org/dat-61kfys3.

### C.2  PREPROCESSING

For human scRNA-seq data, sci-Plex, we followed the same processing steps recommended in Cuturi et al. (2023). Genes which appear in less than 20cells, and cells with less that 20 gene expressed are excluded. Then we normalized gene expression, by first normalized to counts per million (CPM) and then transformed using the formula $\log(CPM + 1)$. Then we whiten the data and apply PCA.

For the mouse scRNA-seq datasets, we preprocessed the raw count matrix. First, we performed library size normalization by converting counts to counts per million (CPM), followed by log-transformation. For feature selection, we chose a subset of highly variable genes combined with a list of known marker genes from the mouse brain atlas.

### C.3  OMT TRAINING

For each dataset, we first performed dimensionality reduction and then trained the OMT model on the resulting low-dimensional embeddings. The specific hyperparameters were tailored to each dataset.

**sci-Plex Dataset.** As previously described, we used PCA for dimensionality reduction. On the resulting PCA embeddings, we trained the OMT model with the number of source components set to $K_s = 3$ and target components to $K_t = 5$. The entropy regularization parameter for both $(\Omega, P)$ was set to $0.01$.

**MERFISH Mouse Brain Data.** For the Vizgen MERFISH brain data, similar to Halmos et al. (2025), we only focused on transporting cell spatial coordinates from the source slice to the target one in 2-dimensional space $(x, y)$. We first fit GMMs to the source and target cells' location independently. Using the resulting components, we minimized the OMT loss to transport cells across slices. Figure 11 shows the impact of number of GMM components on transport accuracy. Due to the dense distribution of cells, using a small number of components results in a poor alignment between the transported cells and the ground truth. Conversely, increasing the number of components, i.e., $K_s = K_t = 1000$, provides significantly better alignment between the transported cells and the target.

**Mouse scRNA-seq Data.** For the mouse scRNA-seq data, we first trained a variational autoencoder (VAE) to learn a compressed cellular representation in a latent space of dimension $d_z = 10$. We then trained the OMT model on these VAE embeddings. The number of components was set within a range of $5$ to $25$, with the specific value chosen based on the biological context; we typically used approximately twice the number of known cell types present at the analyzed timepoints.

## C.4 Additional Results

Table 6: Average $D_\varepsilon \downarrow$ values for transport maps across different neural OT methods (W2-OT, ENOT, ExNOT) versus non-neural methods (EOT, PROGOT, GMM-OT) and the proposed OMT method on the sciPlex dataset for two treatments (Srivatsan et al., 2020).Results are reported as mean and standard deviation over 2 random seeds.

| Method | Belinostat | | | Dacinostat | | |
| --- | --- | --- | --- | --- | --- | --- |
| | d=16 | d=64 | d=256 | d=16 | d=64 | d=256 |
| W2-OT | $8.39 \pm 0.01$ | $166.9 \pm 10.6$ | $113.9 \pm 115.2$ | $8.26 \pm 0.01$ | $45.6 \pm 1.10$ | $833.4 \pm 60.2$ |
| ENOT | $3.6 \pm 1.6$ | $40.28 \pm 11.3$ | $53.5 \pm 53.6$ | $3.15 \pm 2.15$ | $10.3 \pm 9.28$ | $45.2 \pm 35.2$ |
| ExNOT | $3.52 \pm 1.50$ | $45.6 \pm 1.10$ | $101 \pm 22.5$ | $3.05 \pm 1.82$ | $21.5 \pm 20.4$ | $305 \pm 50.2$ |
| EOT | $17.4 \pm 0.01$ | $65.6 \pm 0.03$ | $255.9 \pm 0.07$ | $18.4 \pm 0.01$ | $66.9 \pm 0.03$ | $256.2 \pm 0.12$ |
| PROGOT | $8.42 \pm 0.01$ | $47.9 \pm 0.06$ | $245.7 \pm 0.65$ | $8.83 \pm 0.03$ | $46.6 \pm 0.13$ | $239.1 \pm 0.36$ |
| GMM-OT | $7.88 \pm 0.06$ | $48.1 \pm 0.09$ | $242.9 \pm 0.30$ | $8.15 \pm 0.03$ | $46.0 \pm 0.36$ | $232.9 \pm 1.13$ |
| OMT | $7.87 \pm 0.05$ | $47.5 \pm 0.23$ | $237.1 \pm 0.72$ | $8.10 \pm 0.02$ | $45.9 \pm 0.06$ | $229.7 \pm 1.33$ |

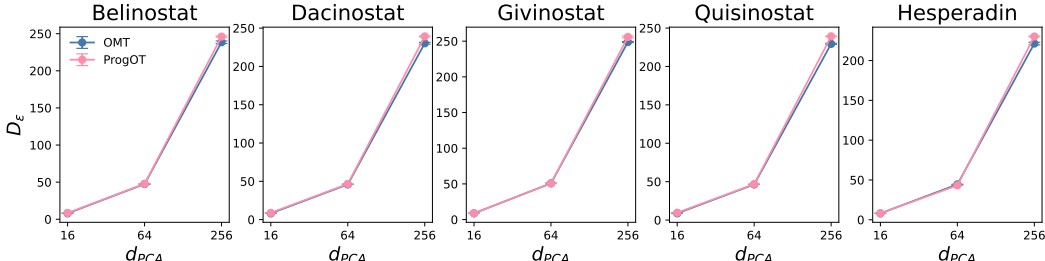

Figure 9: Average $D_\varepsilon \downarrow$ values for the forward and backward OMT mappings compared to PROGOT sci-Plex dataset (Srivatsan et al., 2020). The results reported as the mean over $5$ randomly initialized runs, with standard deviations.

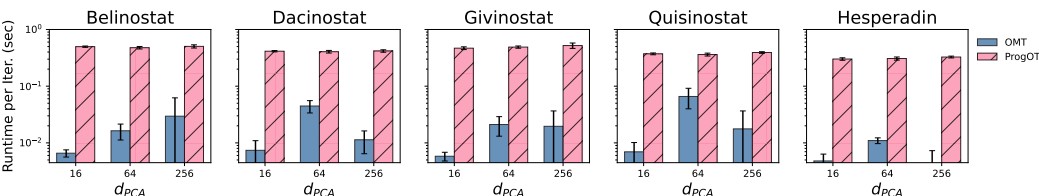

Figure 10: Comparison of the average runtime per iteration for OMT and ProgOT on the sci-Plex dataset as a function of latent space dimensionality $d_{PCA}$. The sharp decrease in OMT's runtime at $d = 256$ arises from switching from a full covariance approximation to a diagonal structure. Results are shown as the mean $\pm$ standard deviation over 5 randomly initialized runs.

# D   IMAGE DATASETS

## D.1   TRANSLATION TASK

Similar to our approach with scRNA-seq data, our method for image datasets in translation tasks involves a two-step process. We first train a deep neural network to learn a low-dimensional representation of the images, and then train the OMT model on these resulting embeddings. For the MNIST dataset, we employed a convolutional VAE featuring a dual-decoder design, where each decoder reconstructs images for the source and target domains, respectively. The latent dimension for this network was set to $d_z = 10$. The full architecture is detailed in Table 7.

For the CIFAR-10 dataset, we utilized the DoubleRessNet architecture, described in Table 8, with a latent space dimension of $d_z = 32$. For all experiments on these datasets, the subsequent OMT model was trained using 10 components for both the source and target measure and $\varepsilon = 0.01$. We found the number of components choice to be robust, as preliminary experiments with other values did not yield significant changes in the final results.

## D.2   ALIGNMENT TASK

For the sake of comparability, we follow the experimental setting provided in Halmos et al. (2025), without using any dimensionality reduction. We selected ImageNet data with $1,281,166$ images (resized to $224 \times 224$) from the ImageNet ILSVRC dataset Deng et al. (2009). The alignment was performed on high-dimensional embeddings obtained using a pretrained ResNet-50 (He et al., 2016) (accessible through PyTorch). The dataset was randomly split into two sets of $640,500$ images for the source and target. We then optimized OMT with $1,000$ components on each domain to learn the alignment function between source and target images. Since the mixture transport function is optimized in a high-dimensional embedding space, and learning full covariance is challenging, we used a diagonal covariance structure for all components.

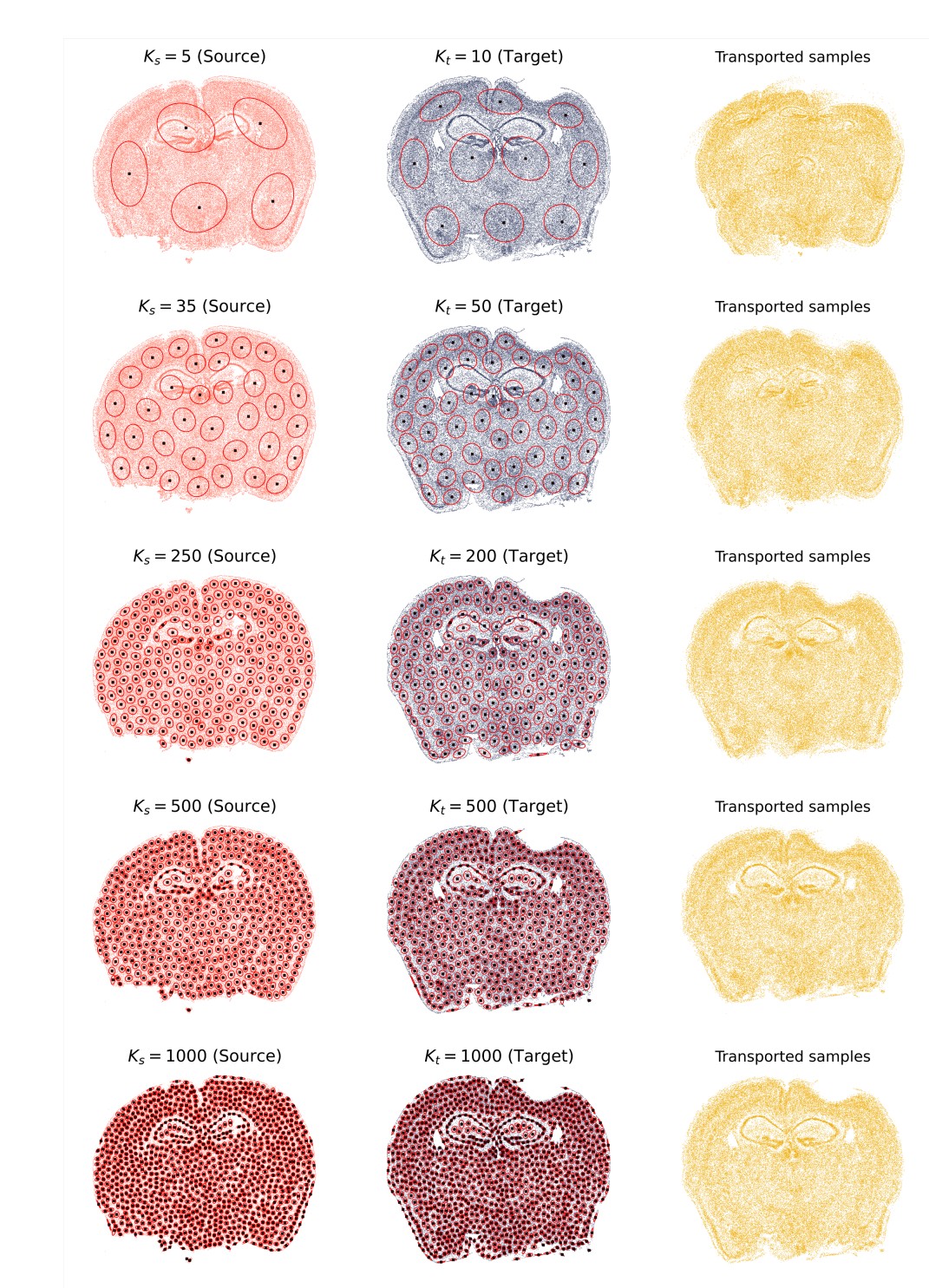

Figure 11: Impact of number of Gaussian components on OMT performance. The source (red) and target (blue) cell distributions are approximated using Gaussian mixtures with increasing numbers of source ($K_s$) and target ($K_t$) components. The rows show the progression of fitting accuracy as the number of components increases from $K_s = 5$ (top) to $K_s = 1000$ (bottom). (Left) Source cells (red dots) and (Center) target cells (blue dots) overlaid with their respective Gaussian components. (Right) The resulting transported samples (yellow dots).

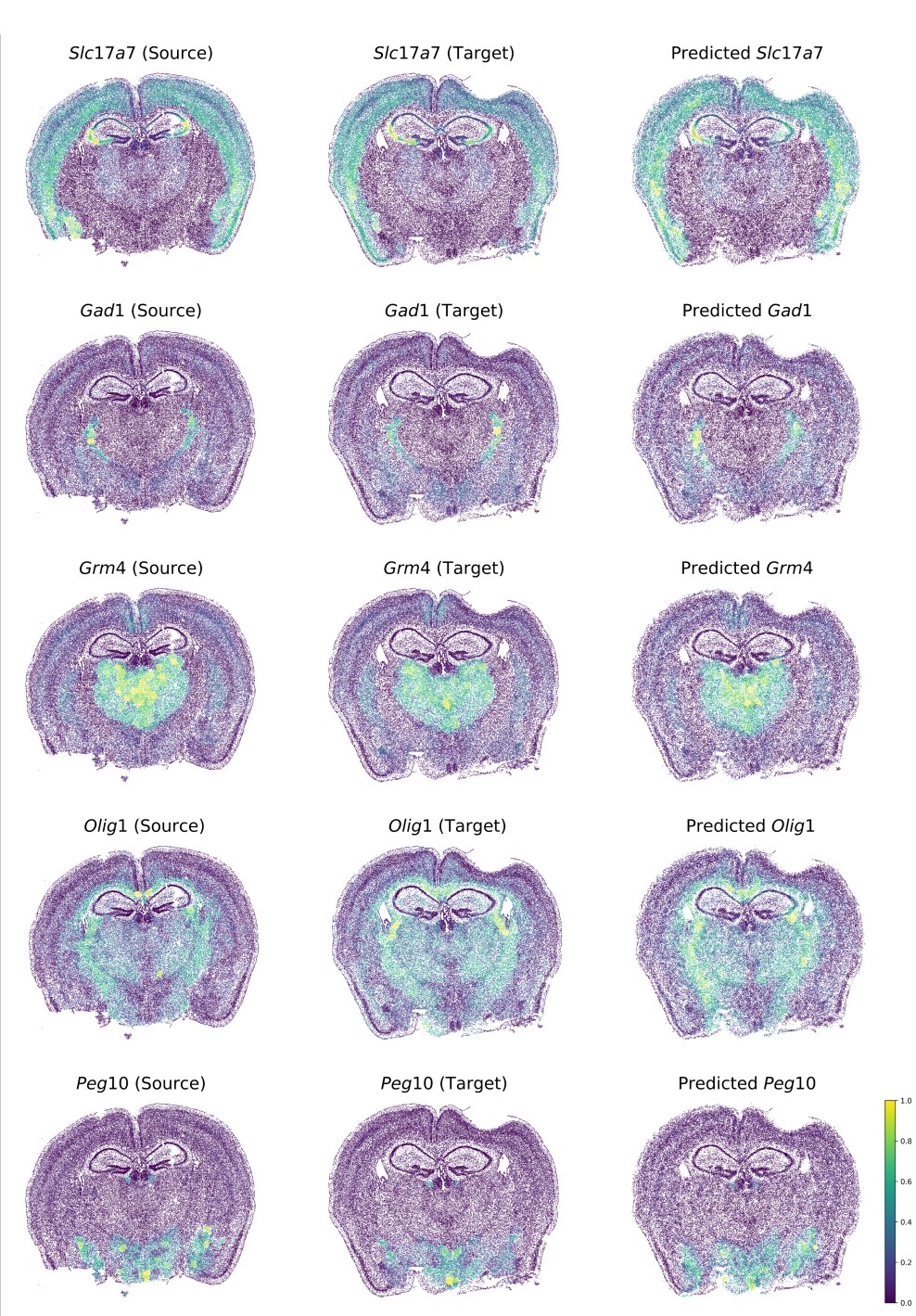

Figure 12: Spatial gene expression imputation using OMT with $K_s = 1000$, $K_t = 1000$. Rows show expression maps for five distinct genes: *Slc17a7*, *Gad1*, *Grm4*, *Olig1*, and *Peg10*. From left to right, the plots illustrate the source distribution, the target ground truth, and the predicted expression after transport, demonstrating how well the transported expression (right) aligns with the ground-truth target (middle). Here, the cosine similarity exceeds 0.75, with a total displacement cost of 107.69.

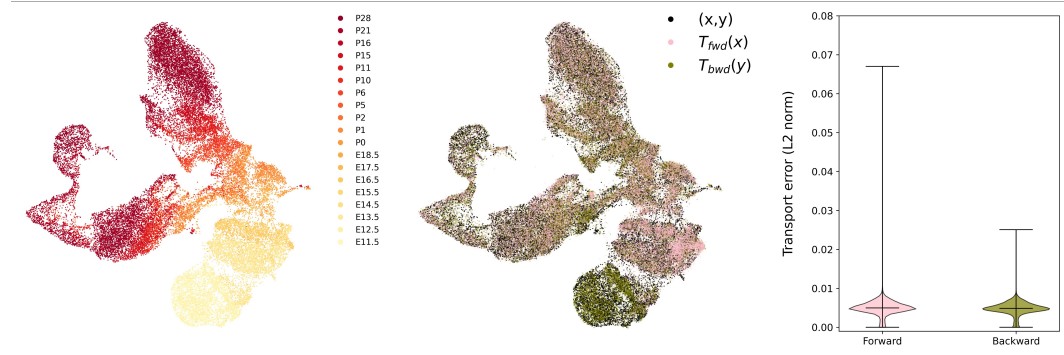

Figure 13: OMT performance on the mouse visual cortex developmental dataset. The data comprises $32,998$ cells and $9,900$ HVGs (Gao et al., 2025). (Left) UMAP visualization of cell distribution colored by developmental time point, from $E11.5$ to $P28$. (Middle) UMAP overlay showing the alignment between measurement, $(x, y)$ (black dots) and predicted cells from forward, $T_{fwd}(x)$ (pink dots) and backward, $T_{bwd}(y)$ (olive dots) transport. (Right) Violin plots of the transportation error (L2 norm) across all genes and cells for both directions.

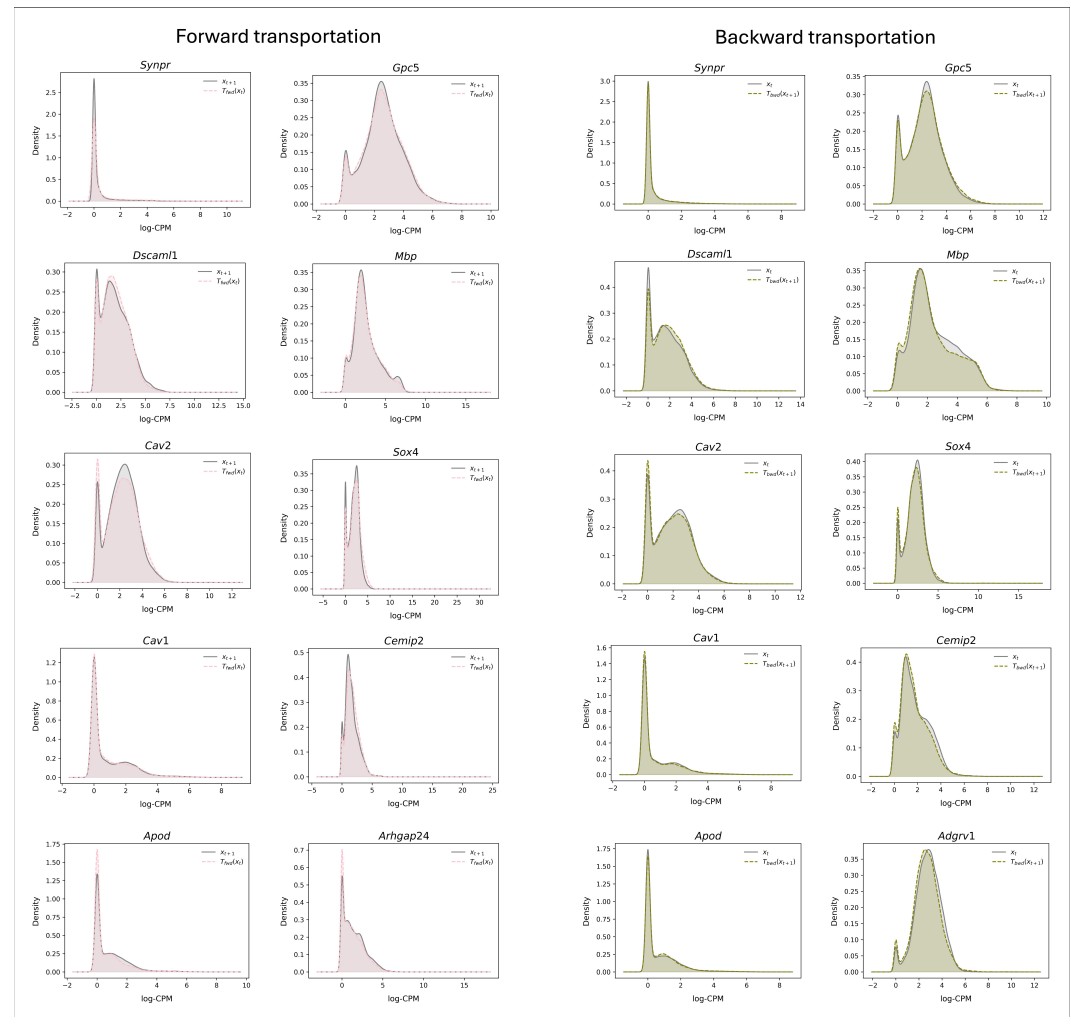

Figure 14: Distributional alignment of log-CPM expression values for a subset of marker genes in non-neuronal cells across developmental time. Each subfigure shows distributions of the ground-truth expression (solid line) and the OMT-transported values (dashed line), for both the forward (Left) and backward (Right) directions.

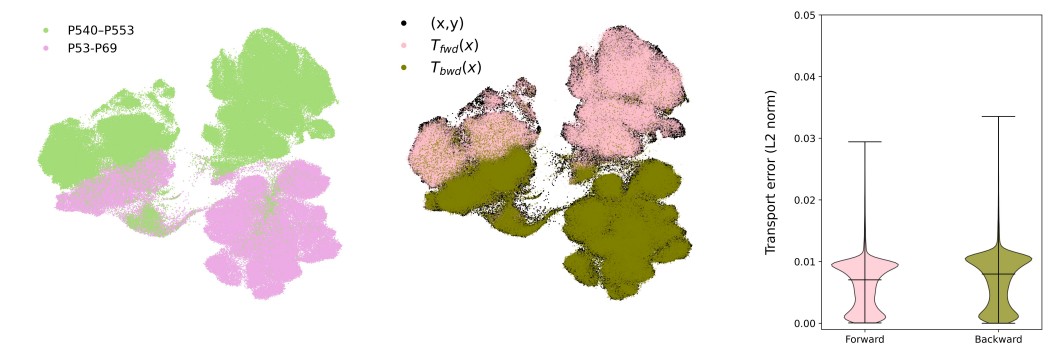

Figure 15: OMT performance on the mouse aging dataset. The dataset comprises $253,468$ cells and $9,359$ HVGs sampled from six brain regions (Jin et al., 2025). (Left) UMAP embedding of cell distributions at two time points, adult and aged. (Middle) UMAP overlay showing the alignment between collected cells, $(x, y)$ (black dots), and cells predicted by the forward, $T_{\mathrm{fwd}}$ (pink dots), and backward, $T_{\mathrm{bwd}}$ (olive dots), OMT maps. (Right) Violin plots of the transportation error (L2 norm) across all genes and cells for both directions.

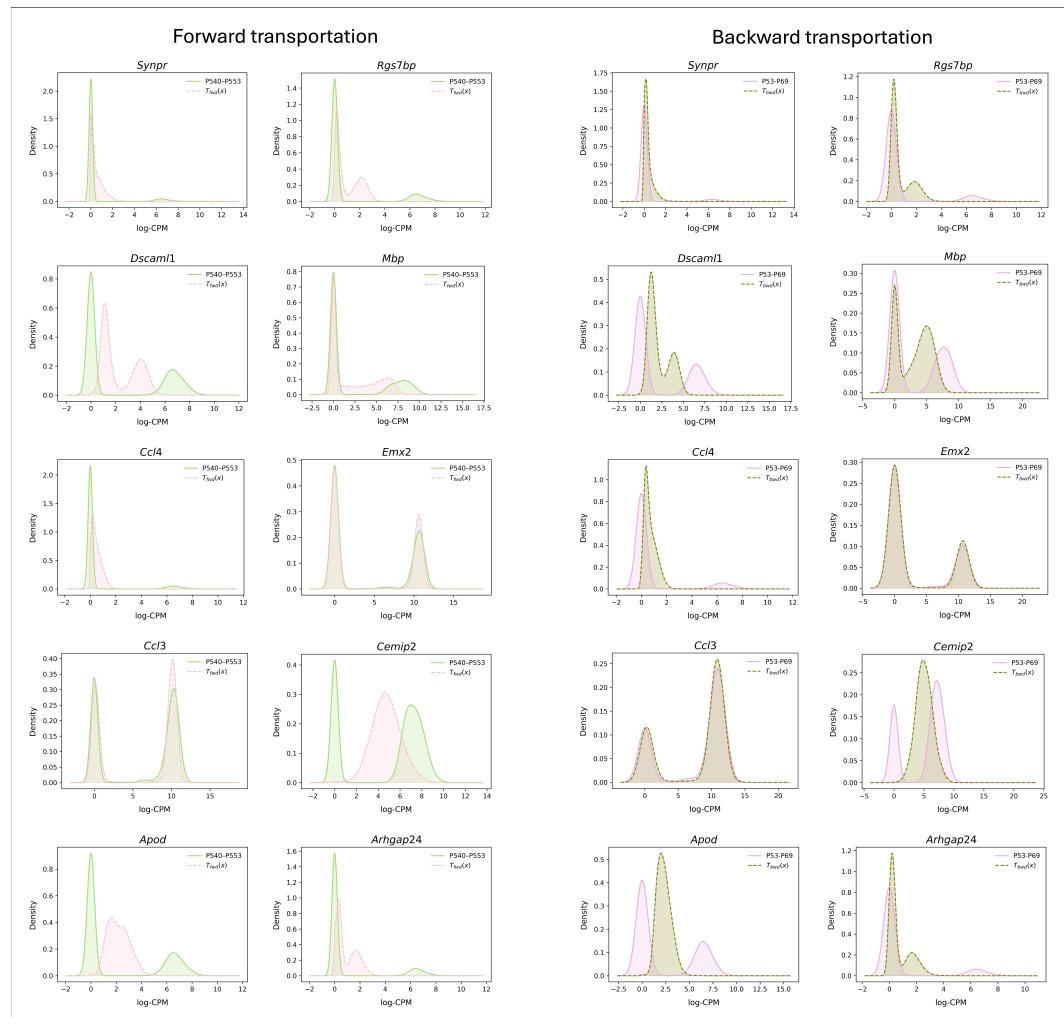

Figure 16: Distributional alignment of log-CPM expression values for a subset of marker genes in non-neuronal cells in the mouse ageing dataset. Each subpanel shows distributions of the ground-truth expression (solid line) and the OMT-transported values (dashed line), for both the forward (Left) and backward (Right) directions.

Table 7: Architecture of the VAE for MNIST.

| Layer | Configuration Details | Activation |
|---|---|---|
| **Encoder** | | |
| Input Layer | 2 x (Batch, 1, N, N) Image | - |
| Dropout | | - |
| Conv2d (x2) | $1 \rightarrow 16$ channels, kernel=(5,5), stride=2 | ReLU |
| Norm2d | 16 features | - |
| Conv2d | $16 \rightarrow 32$ channels, kernel=(3,3), stride=2 | ReLU |
| Norm2d | 32 features | - |
| Conv2d | $32 \rightarrow 32$ channels, kernel=(3,3), stride=2 | ReLU |
| Norm2d | 32 features | - |
| Flatten | Reshapes feature map to (Batch, 128) | - |
| Linear | $128 \rightarrow 100$ units | ReLU |
| **Decoders** | | |
| Linear | $d_z \rightarrow 100$ units | ReLU |
| Linear | $100 \rightarrow 128$ units | ReLU |
| Unflatten | Reshapes to (Batch, 32, 2, 2) | - |
| ConvTranspose2d | $32 \rightarrow 32$ channels, kernel=(3,3), stride=2 | ReLU |
| Norm2d | 32 features | - |
| ConvTranspose2d | $32 \rightarrow 16$ channels, kernel=(5,5), stride=2 | ReLU |
| Norm2d | 16 features | - |
| ConvTranspose2d | $16 \rightarrow 1$ channel, kernel=(5,5), stride=2 | ReLU |
| Norm2d | 1 feature | - |
| ConvTranspose2d | $1 \rightarrow 1$ channel, kernel=(4,4) | ReLU |

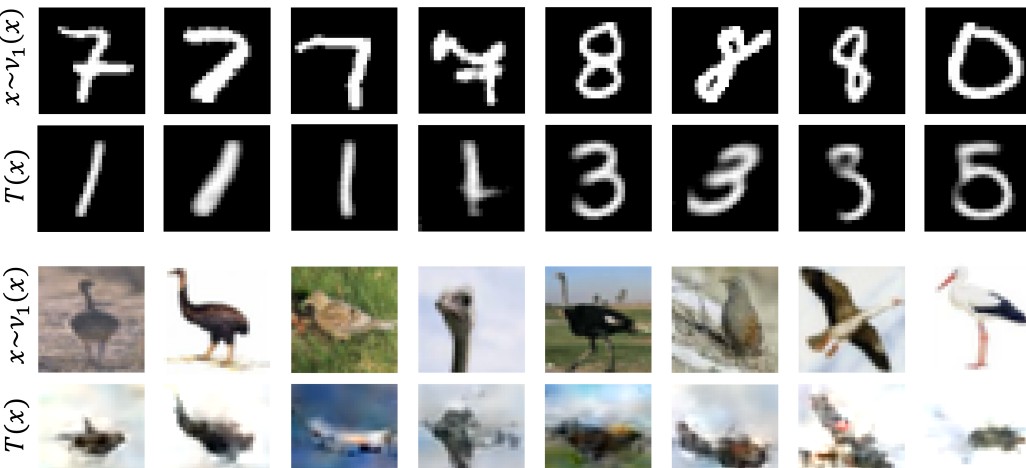

Figure 17: Additional results for unpaired image-to-image translation on the MNIST and CIFAR-10 datasets. For each dataset, the top row shows samples from the target distribution $x \sim \nu_1$, while the bottom row shows the corresponding transported images $T_{\text{OMT}}^{\nu_1 \rightarrow \nu_0}$, generated by OMT in the backward direction.

Table 8: Architecture of the DoubleRessNet for CIFAR-10.

| Layer | Configuration Details | Activation |
|---|---|---|
| **Shared Encoder** | | |
| Input Layer | 2 x (Batch, 3, H, W) | - |
| Conv2d | $3 \rightarrow 32$ channels, kernel=9, stride=1 | ReLU |
| Norm2d | 32 features | - |
| Conv2d | $32 \rightarrow 64$ channels, kernel=3, stride=2 | ReLU |
| Norm2d | 64 features | - |
| Conv2d | $64 \rightarrow 128$ channels, kernel=3, stride=2 | ReLU |
| Norm2d | 128 features | - |
| Residual Block | 4 stacked blocks (128 channels) | ReLU |
| Flatten | Reshapes to (Batch, $H \times 64$) | - |
| Linear | $H \times 64 \rightarrow H \times 16$ units | ReLU |
| Norm1d | $H \times 16$ features | - |
| **Latent Space** | | |
| Linear | $H \times 16 \rightarrow H \times 4$ units | Tanh |
| **Decoders** | | |
| Linear | $H \times 4 \rightarrow H \times 16$ units | - |
| Norm1d | $H \times 16$ features | - |
| Linear | $H \times 16 \rightarrow H \times 64$ units | ReLU |
| Norm1d | $H \times 64$ features | - |
| Unflatten | Reshapes to (Batch, $H$, 8, 8) | - |
| Upsample, Conv2d | $H \rightarrow 64$ channels, kernel=3, upsample=2 | ReLU |
| Norm2d | 64 features | - |
| Upsample, Conv2d | $64 \rightarrow 32$ channels, kernel=3, upsample=2 | ReLU |
| Norm2d | 32 features | - |
| Upsample, Conv2d | $32 \rightarrow 3$ channels, kernel=9, stride=1 | Linear |

