# OpenReview forum: "A Biconvex Formulation for Transport of Mixture Models"
_ICLR.cc/2026/Conference — Submitted to ICLR 2026_

### Official Review · Reviewer_JQ23 · 2025-10-29

**Soundness:** 3
**Presentation:** 3
**Contribution:** 3
**Rating:** 4
**Confidence:** 4

**Summary:**

Summary:
This paper introduces Optimal Mixture Transport (OMT), an entropic OT framework that transports mixture models (instantiated for GMMs) via a strictly biconvex objective with entropy on both component couplings and mixing weights; the authors prove existence of a unique minimizer and show the GOP procedure reaches it in one iteration. They evaluate on the Korotin W2 benchmark against EOT/ENOT/ExNOT/PROGOT and present single-cell RNA-seq and image translation case studies; for biology, OMT is trained on PCA embeddings rather than raw gene space.

**Strengths:**

Strength:
1. The objective is strictly biconvex for any ε₁,ε₂>0, and the GOP optimization is proven to converge to a unique solution in a single iteration. The method also has theoretical elegance, for GMMs, the optimal mixture transport is itself a GMM, preserving structure and enabling parametric efficiency.
2. OMT matches or surpasses strong baselines on Dε/MSE while reporting favorable runtime/memory on the Korotin setup.
3. The method yields component-to-component coupling matrices that expose biologically meaningful correspondences (e.g., OPC→oligodendrocyte maturation trajectories).
4. Noise ablations show comparatively stable performance across methods.

**Weaknesses:**

Weakness:
1. This paper lacks the comparison with an amortized optimized W2OT (https://arxiv.org/abs/2210.12153) on the W2OT benchmark. Though these methods are general neural OT, it is still insightful to be included.
2. The biological evaluation part focuses on evaluation in PCA/latent space, which is generally less informative in biology. Though it is sound in machine learning, you should add evaluation in gene space. Please refer to CellOT (https://www.nature.com/articles/s41592-023-01969-x) and W1OT (https://academic.oup.com/bioinformatics/article/41/Supplement_1/i513/8199349) to add evaluation on the top differentially expressed genes (e.g., DE-gene preservation) and you may also include these two methods as biologist do not really care methodology but care more about biological performance.
3. Though the whole method looks fine, the writing is not well structured, it takes me a little bit more time to understand the method is not either discrete OT nor common neural OT while the paper introduces a lot about them. The writing may be improved but it is fine if the authors keep current version.

**Questions:**

Please see questions in weakness.

**Details Of Ethics Concerns:**

No concerns.

---

> ### Author Response · Authors · 2025-11-25
> **Response to Reviewer JQ23**
>
> We thank the reviewer for the feedback and for evaluating our paper as “good” in all aspects (Contribution, Soundness, Presentation). Below, we address all of the reviewer’s suggestions.
>
> **Comparison with an amortized optimized W2OT on the W2-benchmark task**
>
> We have included the amortized W2-OT in the benchmark task, as recommended by the reviewer (Figures 2, 7). Our new results show that OMT outperforms this new baseline as well as other newly added baselines.
>
> **Additional evaluation in gene space**
>
> We thank the reviewer for this suggestion. To address it, we have conducted an additional analysis on a group of non-neuronal marker genes (as reported in Gao et al., 2025 and Jin et al., 2025), which is presented in Appendix, Figures 14 and 16. Furthermore, we have reported the transport errors at the gene level in Appendix, Figures 13 and 15. We also included additional MERFISH mouse brain data. Please see the results in Figures 4, 11, 12 and Tables 2, 4.
>
> **Manuscript structure**
>
> We aim to work on the organization of the paper once the experiments solidify during this rebuttal period. We welcome suggestions from the reviewers regarding the presentation.
>
>
> We would be happy to provide any further clarification or improvements during the discussion phase.

---

> ### Comment · Reviewer_JQ23 · 2025-11-27
>
> Hi Authors,
>
> Thank you for your rebuttal. However, I do not see the correct W2OT baseline. You said "W2-OT: We used the recommended ICNN architectures and employed regression-based amortization with L-BFGS fine-tuning, training for a maximum of 10^5
> iterations." However, the paper clearly shows that W2OT model works best when they use MLP as the potential model (see the original paper Table 1). As far as I know, using ICNN would lead to worse performance for sure becuase the optimization is not easy. And the difference between ICNN and MLP's results are very large in the origianl paper. Please explain why you choose a low performance one.

---

> > ### Author Response · Authors · 2025-11-29
> > **Response to comments by Reviewer JQ23**
> >
> > Our understanding was that the reviewer’s main request was to include an amortized W2OT baseline in the table for more comprehensive coverage of neural OT methods. Since there are many possible configurations for these solvers, we initially chose one of the recommended ICNN-based setups with convex implementation. We did not intend to select a “low-performance” configuration. To the best of our knowledge, ExNOT (Expectile Regularized Neural OT,  Buzun et al., 2024) is reported as the best performer on the W2 benchmark, and thus we did not expect W2OT to surpass it in any setting.
> >
> > In light of your comment, we have rerun the W2OT experiments using the PotentialMLP architecture for the dual potentials and updated Figures 2 and 7.

---

### Official Review · Reviewer_A9fm · 2025-10-30

**Soundness:** 2
**Presentation:** 2
**Contribution:** 1
**Rating:** 2
**Confidence:** 2

**Summary:**

The paper introduces Optimal Mixture Transport (OMT), a computational framework for efficient and interpretable optimal transport between probability distributions. The core of OMT is a biconvex optimization problem that leverages mixture models, specifically Gaussian Mixture Models (GMMs), which are universal approximators. The model is trained using a doubly regularized objective: one entropy regularizer acts on the transport couplings between individual mixture components, while another acts on the coupling of the mixture weights themselves. The authors demonstrate the utility of their method on synthetic benchmarks and large-scale real-world applications, including single-cell RNA sequencing data, where it performs competitively with existing methods while requiring substantially less computation and memory, and on image translation tasks, where it achieves competitive performance with generative adversarial networks.

**Strengths:**

I think that the core idea behind the paper is quite compelling, namely, studying OT-based mappings on populations. In fields like cellular biology, cells exist and evolve as a coordinated system rather than individually. I believe that a distribution-based approach for learning cellular dynamics is intriguing and holds promise for potential applications to the field of spatial transcriptomics or patient modeling. Moreover, I find it positive that the authors considered multiple biological settings and shared some developmental insight on single-cell datasets. This corroborates the usability of the approach in applied scenarios, going beyond a simple method implementation.

**Weaknesses:**

Unfortunately, in the current state, I am leaning towards a rejection score. Although the model has some positive aspects, I think the contribution and experimental design (especially on the image translation task) fall below the acceptance line for a conference like ICLR. I am happy to discuss more with the authors during the rebuttal phase.

- **General opinion.** If I understand correctly, the main novelty of the paper is to formulate the OMT problem (already existing [1]) in terms of entropy regularization and devise an approach to solve it. The lack of novelty is usually not a sufficient reason for rejection. But I also believe it should be supported by convincing experimental evidence, which I did not find here. If OMT already existed, as I understood, what's the advantage of entropic regularization versus the standard OMT? Is it shown somewhere in the paper? I assume that using the GOP algorithm for optimization is also a central part of the contribution, but in this case, the method already existed.

- Related to one of my questions: I think comparing performance against a continuous Flow Matching or Diffusio Schrödinger Bridge model would reinforce the relevance of the methods. While discrete OT is interesting when analyzing the coupling in the Kantorovivch sense, I feel it may be a bit limited for prediction.

-

- A relevant paper with citation here is [3], where the authors also leverage the closed-form formulation of OT between Gaussians. Can you briefly elaborate on the similarities between [3] and the current paper?

- **Minor.** I recommend defining the entropy $H$ for a broader audience. Of course, not in a formula, it's enough to have one sentence in natural language. Similarly, I would define somewhere $\phi$ in Eq. 7.

- I think you can enrich the citations for Schrödinger Bridges on page 4 (e.g. [4]). As of now, I see almost no citation thereof.

- I personally would move section 4.2 somewhere before where it is now. I feel it is more coherent with the general description of OT for Gaussian mixtures. Then, I suggest you could make a new section about optimization only.

- **Error bars Fig. 3.** I recommend the addition of error bars to Fig. 3. Since noise addition is inspected, error bars would allow a better understanding of the method's robustness.

- **Fig. 2:** When I look at Fig. 2, the performance of ExNOT stands out compared to the other models. As $>2^7$ is an unlikely feature range, I would probably pick ExNOT in a realistic scenario.

- **Data description and setting.** Although it is a machine learning paper, I feel it would be useful to have a more thorough description of the perturbation prediction task. At the moment, it is not clear what problem is solved (transporting control to perturbed states, I assume).

- **Line 430: typo on *developmetnal*.

- **Image experiments.** For me, the most critical part is the experiments on images, especially on ImageNet. Currently, hundreds of models have been tested on ImageNet and MNIST for translation. Reporting two of them feels a bit selective to me. Moreover, I would recommend presenting Tab. 2 differently, i.e., including error bars everywhere. Based on the results, I don't feel that OMT is coming out very strongly.

- **General remark.** In general, I feel that I miss some solid evidence that the approach is providing something additional than standard OT or image-to-image translation models. The method is not doing much better than others, and the benchmarks are not extremely comprehensive. Maybe you can find some advantage in performing population-based OT worth reporting?

[1] Chen, Yongxin, Tryphon T. Georgiou, and Allen Tannenbaum. "Optimal transport for Gaussian mixture models." IEEE Access 7 (2018): 6269-6278.

[2] Tong, Alexander, et al. "Improving and generalizing flow-based generative models with minibatch optimal transport." arXiv preprint arXiv:2302.00482 (2023).

[3] Bunne, Charlotte, et al. "The schrödinger bridge between gaussian measures has a closed form." International Conference on Artificial Intelligence and Statistics. PMLR, 2023.

[4] De Bortoli, Valentin, et al. "Diffusion schrödinger bridge with applications to score-based generative modeling." Advances in neural information processing systems 34 (2021): 17695-17709.

**Questions:**

- **Add some Flow Matching references.** Many people are exploring flow-matching-based OT. ** Have you thought of benchmarking against them? Flow Matching works are cited in the paper; have you considered using them for benchmarking, as they are well-established models, for example, in cellular trajectory prediction?

- For the considered problems, the model may be advantageous. But for high-dimensional cases, I feel that using Gaussian mixture models may become too approximate, and the use of another paradigm (like generative OT) could be preferred. Can the authors elaborate on this?

- Why aren't all methods included in Tab. 1 of those listed in Fig. 3? \

- What is the latent dimensionality for the VAE applied to single-cell data?

---

> ### Author Response · Authors · 2025-11-25
> **Response to Reviewer A9fm**
>
> We thank the reviewer for recognizing the value of our biological experiments and developmental insights. We would like to note that applying OMT to single-cell genomics is our central focus. Our goal was to provide a practical, robust tool capable of uncovering biological insights, such as tracing cellular lifespans and developmental pathways.
>
> **Addressing major comments/concerns**
> - *Novelty:* We would like to clarify that the OMT method, specifically the entropic solver proposed in Eq. 13, is a novel contribution of this paper and did not exist previously.  OMT is an entropic OT solver, but effectively operates by transporting parametric sub-populations (clusters) rather than individual samples. This avoids solving the full entropic optimization for every single data point, offering significant computational advantages (Please see our explanation of OMT vs. GMM-OT and LOTs in our general response above).
>
> - *" ... the GOP algorithm for optimization is also a central part of the contribution, but in this case, the method already existed."*
>
>      The GOP algorithm is not the central contribution; it is a general algorithm for biconvex optimization. Our main contribution lies in the novel formulation of the OMT objective function (Eq. 13). We demonstrate that this specific formulation possesses unique theoretical properties (Theorem 1) that do not hold for general biconvex problems solved by GOP or for other existing mixture OT solvers. Additionally, we prove the stability of OMT maps, in which we show that the expected deviation of transported samples is bounded with respect to perturbations in the source measure (Theorem 2).
>
> - *"I think comparing performance against a continuous Flow Matching or Diffusio Schrödinger Bridge model would reinforce the relevance of the methods. While discrete OT is interesting when analyzing the coupling in the Kantorovivch sense, I feel it may be a bit limited for prediction."*
>
>     OMT does not solve the standard discrete Kantorovich problem; it solves the specific objective defined in Eq. 13, which utilizes two distinct entropic regularizers. The resulting transport plan by OMT is a mixture map composed of both continuous (Gaussian) and discrete (mixing matrix) components. Regarding comparisons to continuous flow matching or Schrödinger Bridge (SB) methods: These approaches learn a dynamical model (a time-dependent trajectory), whereas OMT learns a static transport map between the source and target measures. While our focus is not on dynamical transport, we explicitly included ENOT in our W2-benchmark tasks, which is a diffusion-based (SB) OT solver. OMT outperforms ENOT in these benchmark tasks. We have updated the text to better clarify this aspect.
>
> - *ImageNet:* We have added more experiments with two additional datasets, including ImageNet, in the updated manuscript. Briefly, we we added experiments using two new datasets (including ImageNet) to the updated manuscript (Tables 2, 4 and Figures 6, 11, 12). We compared OMT against additional low-rank baselines, such as the recent HiRef solver, by adopting the image alignment task. Our analysis confirms that OMT outperforms LOTs across all these new settings.
>
> - *General remark:*  What we are aiming here is to provide a (i) computationally efficient, (ii) theoretically-sound, non-neural OT solver (iii) that scales efficiently without relying on expensive sample-to-sample transport. This design is particularly advantageous for single-cell applications. Despite being a non-neural method, OMT delivers competitive performance (often outperforming existing neural solvers), while guaranteeing theoretical stability and robustness. Furthermore, it consistently outperforms SOTA non-neural baselines (ProgOT and HiRef) across diverse tasks. We hope that the new experiments and the new baselines models will also help to provide a stronger, coherent perspective on OMT. In total, the revised manuscript benchmarks OMT using 8 datasets, 6 tasks, 10 baseline OT solvers.

---

> ### Author Response · Authors · 2025-11-25
> **Addressing Minor concerns**
>
> **Addressing minor comments**
> - *"I recommend defining the entropy  for a broader audience. Of course, not in a formula, it's enough to have one sentence in natural language. Similarly, I would define somewhere  in Eq. 7."*
>
>    Thanks for pointing this out. We have added the rationale behind entropic regularization in the revised text in the “Entropic optimal transport” section.
>
> - *"I think you can enrich the citations for Schrödinger Bridges on page 4 (e.g. [4]). As of now, I see almost no citation thereof."*
>
>    We note that our manuscript already cited the recent SB-based OT work by Gushchin et al., 2024. To ensure a comprehensive literature review, we have now added citation [4] along with two additional recent studies (*) and (**) in the Related Work section.
>
> - *"moving section 4.2 somewhere before where it is now"*
>
>     Thanks for the suggestion. We added a new stability result to the main text. Together with it, we found it hard to move this section earlier because these novel/key components depend on the previously introduced preliminaries. For now, we chose to keep it at the end of the Methods section. We remain open to changes on the organization and would welcome further suggestions from the referee.
>
> - *Error bars Fig. 3:* We have updated Figure 3 and included std error bars.
>
> - *Data description and setting in the ablation study*: Thank you for pointing this out. We have added an additional section to Appendix B.2, which explains the ablation study conducted with noise to evaluate stability.
>
> - *typo:* Thanks. We fixed the typo.
>
> - *"I would recommend presenting Tab. 2 differently, i.e., including error bars everywhere. Based on the results, I don't feel that OMT is coming out very strongly."*
>
>     As noted in the table caption, the reported values for WGAN and WGAN-GP are sourced from previous studies (Choi et al., 2023; Rout et al., 2022; Qian et al., 2021), which did not report std values for all datasets. However, assuming std<< 1, our claims remain valid. Here, our objective is to provide baselines showing that OMT achieves translation quality comparable to powerful generative models (GANs),  while being computationally much simpler.
>
>
>
>
>
>
> (*) Shi et al., Diffusion Schrödinger bridge matchin, NeurIPS, 2023.
>
> (**) Gushchin et al., Light and optimal Schrödinger bridge matching. ICML, 2024.

---

> ### Author Response · Authors · 2025-11-25
> **Addressing Reviewer's Questions**
>
> **Q1:** A relevant paper with citation here is [3], where the authors also leverage the closed-form formulation of OT between Gaussians. Can you briefly elaborate on the similarities between [3] and the current paper?
>
> **Answer:**  In Bunne et al. (2023) paper, authors extend the findings of static *Gaussian* (not a mixture) transport (Janati et al., 2020) to the dynamic OT setting, using the SB framework. In this dynamic context, the objective shifts from learning a static map to modeling the full transport dynamics and learning a continuous trajectory of the probability measure. There is currently no trivial way to generalize the findings of Bunne et al. to the Gaussian mixture setting. While their work handles dynamic transport for Gaussians, extending the OMT (which operates on mixtures) to support dynamic OT remains an open challenge. We view this as a promising direction for future research and have mentioned this potential extension in the Conclusion of the revised manuscript.
>
> **Q2:** Add some Flow Matching references. Have you thought of benchmarking against them? Flow Matching works are cited in the paper; have you considered using them for benchmarking, as they are well-established models, for example, in cellular trajectory prediction?
>
> **Answer:** We acknowledge the relevance of dynamic OT, such as flow matching and SB-based methods, for inferring cellular trajectories. However, one major unaddressed issue with standard dynamic OT implementations is the lack of constraints at intermediate time points. While these methods guarantee a continuous flow between the source and target samples, they enforce mass preservation constraints only at the start and end points (marginals). They do not guarantee that the transported distribution matches the true biological population at intermediate time steps. For instance, if a dynamic SB model is trained to map cells at E11 to P28, the learned global trajectory may not accurately recover the biological structure of intermediate stages, as these intermediate distributions are not part of the optimization objective. While some recent studies attempt to solve this via multi-marginal or iterative constraints, the purpose differs significantly from the work proposed here. Flow matching will become more relevant in extensions studying multi-marginal dynamic transport based on datasets with multiple time points. In the problems studied here, there are no natural constraints on the “flow”.
>
>
> **Q3:** For the considered problems, the model may be advantageous. But for high-dimensional cases, I feel that using Gaussian mixture models may become too approximate, and the use of another paradigm (like generative OT) could be preferred. Can the authors elaborate on this?
>
> **Answer:** This is indeed a relevant question. Please see our response to the last question from Reviewer Dmn where we aim to disambiguate GMM fitting vs GMM inference.
>
>
> **Q4:** Why aren't all methods included in Tab. 1 of those listed in Fig. 3?
>
> **Answer:**  In the ablation study (Figure 3), our goal was to assess whether high-performing methods maintain their accuracy under perturbations. We therefore focused our perturbation analysis on SOTA OT methods with the best performance in the W2-benchmark. Other EOT or W2-OT methods do not meet the baseline performance threshold even in the absence of noise. For the single-cell and image datasets, we primarily focused on non-neural methods, which are the most comparable to OMT in terms of performance. We have now added a sentence to clarify experimental methodology and we thank the reviewer for this comment.
>
>
> **Q5:** What is the latent dimensionality for the VAE applied to single-cell data?
>
> **Answer:** It is a 10-dimensional space. Please refer to lines 440–441 in the main text and Appendix C.3 for details.

---

> > ### Comment · Reviewer_A9fm · 2025-11-26
> >
> > Dear authors,
> >
> > Thank you very much for your answer, and apologies for leaving so many typos in my initial review. After a better contextualization of the method, I have provisionally revised my initial scoring and moved from 2 to 4.
> >
> > **Novelty and contribution.** I apologize for downplaying the methodological contribution of the paper. I must admit I had not contextualized the model properly and was not fully aware of how it differs from existing solvers. All the points made above about novelty and the GOP algorithm make sense to me. I was misunderstood when referring to the Kantorovich formulation aspect. Sorry for not being clear. My general thought was that I found the comparison with discrete OT methods only quite insufficient when dynamical formulations are convenient in an application setting. In a way, the comparison with WGAN and WGAN-GP also entails comparing a discrete OT model (OMT) with parameterized generative models for translation.
> >
> > **New experiments.** Thank you for the new experiments. I find them convincing.
> >
> > **Performance concerns.** My general performance concerns remain are still present (hence, not a higher grade despite I now appreciate the nuances of the contribution much more), especially after adding error bars to Fig. 3.
> >
> > **Remark about Flow Matching.** I am not sure I understand the answer provided by the authors. I do agree that dynamic OT models do not approximate intermediate time points when these are not observed or used in training. But the same applies to discrete methods. Indeed, I was not suggesting that the integration time for the neural ODE should correspond to physical time, but rather that these methods offer well-performing alternatives to approximate OT coupling between source and target distributions. In other words, I was pointing at such methods as alternatives to static OT rather than complementary approaches.
> >
> > **Q4:** Thanks for clarifying this aspect. I am not sure I agree with the choice, though. As the paper proposes OMT as an OT-model alternative, it should compare well with neural solvers. Omitting some of such models from Tab.1 sounds a bit unexpected to me. Am I missing something?
> >
> > Thank you again for your time!

---

> ### Author Response · Authors · 2025-11-29
> **Response to comments by Reviewer A9fm**
>
> We are glad that the reviewer found our explanations and additional experiments useful, and we appreciate their constructive contribution to improving this work. We also thank the reviewer for the positive feedback and increasing the score. Although, due to the technical issue with the OpenReview system, the score reverted to its initial state.
>
> Response to the remaining comments:
>
> - **Remark about Flow Matching:** Thanks for the clarification. While dynamic OT methods can serve as alternative solvers for the coupling problem, they are not efficient for this specific setting: Dynamic OT methods extend static OT solvers by modeling the underlying transport process as a continuous-time dynamical system. The main advantage of such models is that they can be trained across all time points to learn dynamics that explains the entire process, thereby avoiding discontinuities at temporal boundaries. If this advantage is not exploited, a dynamic OT method effectively reduces to a static OT solver with additional inner iterative loop (similar to ProgOT) but without offering the benefit of a globally consistent ODE.
> We also highlight that we have already included a recent dynamic OT solver in our benchmarking, ENOT (Gushchin et al., 2023), which is a SB–based dynamic solver. ENOT did not yield performance improvements over static OT baselines in W2-benchmark tasks.
> To address the reviewer’s concern, we will add the above reasoning to the revised Related Work section.
>
>
> - **Adding neural solvers in Table 1:** As we discussed OMT is a non-neural, static solver. We used the W2 benchmark, which is specifically designed for neural OT solvers, primarily to demonstrate that OMT is competitive with SOTA neural solvers. Once this baseline was established, we narrowed our focus in Table 1 to comparable non-neural, static solvers.
> Since the reviewer remained unconvinced, we additionally ran the reported neural OT solvers on the human scRNA-seq task in Table 1 as well. The new results are reported in Table 6, Appendix C.4. Given the discussion timeframe and the computational demands of neural OTs, we only could run this evaluation on two treatments across two random seeds. We will report the results for all treatments in Table 1 with more random seeds in the camera-ready version.

---

### Official Review · Reviewer_Dmnz · 2025-10-31

**Soundness:** 3
**Presentation:** 3
**Contribution:** 2
**Rating:** 4
**Confidence:** 4

**Summary:**

This paper proposes Optimal Mixture Transport (OMT), a method to speed up computation of optimal transport maps. The core idea is to approximate source and target distributions with Mixture Models, and compute OT maps between components of these mixtures, rather than expensive sample-to-sample transport. The main theoretical contribution is the entropic formulation of this problem as a strictly biconvex optimization problem that jointly optimizes the mixing weights ($\Omega$) that define how much mass flows between pairs of components and the entropic transport plans ($P$) between each of those component pairs. The authors prove that this biconvex problem has a unique minimizer and, critically, that it can be solved efficiently in a single alternating iteration. This makes the algorithm efficient, as it essentially reduces to computing a cost matrix where each entry is the closed-form entropic OT cost between two Gaussian components, and then solving a single, small, discrete entropic OT problem on this cost matrix. The authors demonstrate empirically that OMT is faster, less memory-intensive, and achieves competitive or superior performance compared to several sample-based OT solvers

**Strengths:**

The paper's primary strength is its  novel theoretical formulation. While GMM-based OT is not new, the authors frame the problem as a doubly-regularized objective (with $\epsilon_1$ for component plans and $\epsilon_2$ for mixing weights). Proving that this biconvex problem, which could be optimize as is, can in fact be solved in a single step.  This theoretical contribution translates directly into an efficient and practical algorithm, allowing the solver to effectively compute an "entropic-OT-on-an-entropic-cost-matrix" in a single pass.

The authors obtain compelling empirical results against chosen baselines, showing that OMT provides as significant speed while retaining high-quality transport plans on multiple dataset.

**Weaknesses:**

The paper's primary weakness is its evaluation, which omits comparisons to the most relevant and obvious baselines. The current experiments compare a mixture-based method (OMT) almost exclusively against sample-based methods (EOT, PROGOT, etc.). This makes it impossible to determine if the performance gains come from the paper's novel doubly-regularized biconvex formulation or simply from the GMM approximation itself, which is a common strategy. Key missing comparisons include the standard, well-known GMM-OT solver, which uses the unregularized Bures-Wasserstein distance for its inner cost matrix; without this, the central claim that double regularization is superior is unsubstantiated. Furthermore, comparisons to other clustering-based methods (e.g., k-means centroids), Low-Rank Sinkhorn [1] (which can be thought of a different form of clustering), and modern scalable solvers like Hierarchical Refinement [2] are all necessary to properly contextualize this work.

[1] Low-rank Sinkhorn factorization: https://arxiv.org/abs/2103.04737
[2] Hierarchical Refinement: Optimal Transport to Infinity and Beyond: https://arxiv.org/abs/2503.03025

**Questions:**

My main concern is the accuracy and scalability of the GMM approximation itself. The entire algorithm's success hinges on the ability to fit an accurate GMM to the source and target data with a small number of components ($K$). GMMs are known to suffer from the curse of dimensionality. In simple, low-dimensional cases (like 2D synthetic data), a good fit is easy, but in high-dimensional spaces (e.g., scRNA-seq or image embeddings), fitting a GMM is notoriously difficult and may require a large $K$ to capture the data's structure. The OMT algorithm's complexity scales with $O(K_0 K_1)$, which is only fast if $K_0$ and $K_1$ are small.

How does the performance of OMT (both in accuracy and runtime) degrade as the number of components $K$ must be increased to faithfully represent complex, high-dimensional distributions? How much does the final map's accuracy rely on the quality of the initial GMM fit, and what happens when this fit is poor?

---

> ### Author Response · Authors · 2025-11-25
> **Response to Reviewer Dmnz**
>
> Thanks for the valuable feedback. Below, we address the two main concerns regarding the evaluation and the comparison of OMT with the other baseline methods.
>
> **Comparison with GMM-OT**
>
> We thank the reviewer for their suggestion. Regarding the fundamental differences between GMM-OT and OMT, please see our General Response and the newly added Appendix section (B.4, Figure 8). Following the recommendation, we have included GMM-OT as a mixture-based baseline in our revised experiments (new Figures 1, 7 and Tables 1, 2).
>
> **Comparisons to other clustering-based methods (e.g., k-means centroids), Low-Rank Sinkhorn [1] (which can be thought of a different form of clustering), and modern scalable solvers like Hierarchical Refinement**
>
> We have added low-rank solvers, including the HiRef method, for comparison. Across multiple, newly added experiments, OMT outperforms these baselines. We clarify that our choice of GMMs over general (non-parametric) clustering methods such as K-means is motivated by the need for a parametric representation of the distribution, rather than merely a partition of the data. The parametric form of GMMs allows us to compute the transportation cost between components analytically, which is not available in the general clustering case. In contrast, computing transport costs between K-means clusters requires reverting to costly sample-level calculations, as in standard discrete OT.

---

> ### Author Response · Authors · 2025-11-25
> **Addressing Reviewer's Questions**
>
> **Q1:** Difficulty of fitting a GMM
>
> **Answer:** We would like to highlight a potential source of confusion. As the reviewer mentioned, inferring the true underlying GMM is indeed hard in high-dimensional cases. On the other hand, GMMs are universal function approximators. When the aim is not to interpret the individual mixture components, arbitrarily good fits can be obtained by overparametrizing. Thus, OMT can obtain transport maps with low cost even when the underlying GMM is misidentified. Although this decreases interpretability, a new theoretical result we added during the intervening period (Theorem 2) shows that OMT maps are stable under perturbation: changes to the OMT transport maps are bounded when the source is perturbed (e.g., the GMM fit is suboptimal)  in a bounded way in a compact set.
>
> **Q2:** How does the performance of OMT (both in accuracy and runtime) degrade as the number of components
>  must be increased to faithfully represent complex, high-dimensional distributions?
>
> **Answer:** To address the concern on high-dimensional data, we would like to clarify two aspects:
> - We have now explicitly demonstrated the applicability of OMT on complex, high-dimensional datasets, e.g., ImageNet (performed in a 2048-dimensional feature space, fitting 1000 Gaussian components) demonstrating that our method scales effectively to these regimes.
> - In the presence of high-dimensional data (images or single-cells), it is standard practice to use an embedding or latent space for distance calculations. Computing Euclidean distances directly in pixel or gene space is often undesirable due to the curse of dimensionality, where distance metrics lose meaningfulness. Therefore, dimensionality reduction (e.g., PCA, VAE, etc.) is routinely employed to obtain a feature space where costs are well-defined. This is true for all compared methods; even end-to-end Neural OT methods perform implicit dimensionality reduction and transportation jointly within their network architectures. A consensus range for embedding dimensionality for single-cell datasets seems to be $5\leq K \leq 50$.
>
>
> **Q3:** How much does the final map's accuracy rely on the quality of the initial GMM fit, and what happens when this fit is poor?
>
> **Answer:**
> As explained above, the accuracy (as measured by the transport cost of a test set) depends on the quality of the fit, but not on whether the fit represents an underlying true model. To address the reviewer’s concern, we explicitly investigate this behavior in our analysis of the newly added MERFISH dataset (Figure 11). We demonstrate that when the number of components is too low to capture the structural complexity of the data, the transport map returns incorrect cell state transitions. However, as the number of components increases to an appropriate range, transport accuracy improves.

---

### Official Review · Reviewer_8kbG · 2025-11-03

**Soundness:** 2
**Presentation:** 2
**Contribution:** 3
**Rating:** 4
**Confidence:** 4

**Summary:**

The approach offers an alternative method to solving the OT plan through mixture modelling and entropic regularization, which reduces the computational complexity to obtain the transport map. The approach recasts the problem as a biconvex optimization problem with a unique minimizer, and demonstrates speed ups across various synthetic and real-world benchmarks.

**Strengths:**

The approach is clearly presented and mathematically rigorous, offering practical value given the importance of efficiently computing OT plans across diverse real-world datasets.

**Weaknesses:**

Although the approach seems to be significantly faster to compute than all of the other examined baselines, it remains unclear whether the computed OT plan is invertible. Additional studies around this with improved benchmarks should be included (e.g., hierarchical refinement, MOP, etc.).

**Questions:**

- In Fig. 2, the MSEs suggest that OMT isn't particularly invertible on the Korotin et al. benchmarks even for low-dimensional data? Although the MMDs appears low on the simpler target distributions in Fig. 1, this doesn't translate to these more challenging benchmarks?
- How does the approach compare with hierarchical refinement, another approach to inexpensively computing the OT plan (arXiv preprint arXiv:2503.03025)---both in terms of runtime and invertibility.
- Including an ML-based baseline like minibatch-OT would also be helpful for contrast.
- Testing on additional benchmarks would be beneficial, e.g., the MERFISH atlas, and/or ImageNet (to demonstrate scalability of the approach to larger systems, especially given the fast run-time achieved on the smaller systems).
- The OT cost across baselines would also be helpful to include for comparison.

---

> ### Author Response · Authors · 2025-11-25
> **Response to Reviewer 8kbG**
>
> We thank the reviewer for the insightful feedback. We begin by addressing two main concerns raised in the review.
>
> **It remains unclear whether the computed OT plan is invertible**
>
> Unlike HiRef, OMT is not optimized to learn a bijection (permutation) between the source and target domains. The probabilistic coupling produced by OMT explicitly allows mass splitting and merging, so the resulting map is generally non-injective and not invertible. This is a key feature and a deliberate design choice. Constraining the model to invertible maps (permutations) would restrict it to relatively simple one-to-one matching problems. By relaxing this constraint, OMT can be applied to complex biological processes such as cell differentiation (splitting) and state convergence (merging), and the common scenario of having unequal samples sizes in the source and the target. Importantly, OMT still handles standard matching and alignment tasks (as demonstrated in the ImageNet experiment). Moreover, even though the map is not bijective, we can perform both forward and backward transport using the learned joint coupling.
>
> **Additional benchmarking**
>
> We have added additional comparison studies between low-rank solvers, including HiRef, on both ImageNet and MERFISH datasets. We also incorporated amortized W2OT into the W2-benchmark tasks, as requested by Reviewer JQ23. Our results are presented in the updated manuscript. Across all newly added experiments, we showed that OMT outperforms LOT solvers.

---

> ### Author Response · Authors · 2025-11-25
> **Addressing Reviewer's Questions**
>
> **Q1:** In Fig. 2, the MSEs suggest that OMT isn't particularly invertible on the Korotin et al. benchmarks even for low-dimensional data? Although the MMDs appears low on the simpler target distributions in Fig. 1, this doesn't translate to these more challenging benchmarks?
>
> **Answer:** We may have misunderstood the specific concern regarding "invertibility". If this refers to a bijective function, we note that none of the compared entropic solvers produce invertible maps, as they are designed to model probabilistic couplings rather than permutations. If the concern relates to the performance trade-offs between MSE (sample-to-sample metric) and MMD/$D_\epsilon$ (population-based metrics) in Figures 1 and 2, we emphasize the following: it is important to distinguish between sample-to-sample metrics (MSE) and population-based metrics (MMD, $D_\epsilon$). While OMT is a probabilistic framework and thus not strictly optimized for deterministic MSE, it nevertheless outperforms comparable stochastic solvers (ENOT) on this metric. The primary objective of the W2-benchmark task was to demonstrate that OMT achieves performance comparable to SOTA neural OT solvers, which require extensive computational resources.
>
> **Q2:** How does the approach compare with hierarchical refinement, another approach to inexpensively computing the OT plan (arXiv preprint arXiv:2503.03025)---both in terms of runtime and invertibility.
>
> **Answer:** Please see our explanation in the general response. To address this concern, we have now compared and reported the performance of LOTs and HiRef relative to OMT (MERFISH and ImageNet experiments, Tables 2 and 4, Figure 12). In both cases, OMT outperforms HiRef.
>
> **Q3:** Including an ML-based baseline like minibatch-OT would also be helpful for contrast.
>
> **Answer:** We have now reported the performance of mini-batch OT for batches of size 1024 (MERFISH and ImageNet experiments, Tables 2 and 4).
>
> **Q4:** Testing on additional benchmarks would be beneficial, e.g., the MERFISH atlas, and/or ImageNet (to demonstrate scalability of the approach to larger systems, especially given the fast run-time achieved on the smaller systems).
>
> **Answer:** We added both datasets recommended by the reviewer and compared the OMT performance with the results reported in the HiRef papers (Tables 2,4 and Figures 4, 11, 12).
>
> **Q5:** The OT cost across baselines would also be helpful to include for comparison.
>
> **Answer:** The transportation costs (Tc) of all methods on the W2-benchmark tasks are reported in Figure 7. For the MERFISH and ImageNet datasets, we also report the total cost of OMT compared to low-rank OT methods (Tables 2 and 4).

---

### Author Response · Authors · 2025-11-25
**General Response**

We thank the reviewers for their constructive comments and suggestions. To address the concerns raised, we have incorporated all requested clarifications and additional experiments in this revision, including:

- Compared OMT with low-rank OT solvers (HiRef) (requested by Reviewers 8kbG and Dmnz) and GMM-OT (requested by Reviewer Dmnz).
- Analyzed two additional datasets: MERFISH (mouse brain spatial transcriptomics) and ImageNet, as requested by Reviewers 8kbG and A9fm.
- Included amortized W2OT performance in the W2-benchmark task, which is requested by Reviewer JQ23.

We have updated both the main text and the appendix. To facilitate the review process, all changes and added content are highlighted in blue.

In the current version of the manuscript, we evaluate the applicability and performance of OMT under three scenarios:

- Synthetic data: W2-benchmarking task.
- Four single-cell datasets: SciPlex (human scRNA-seq), MERFISH (mouse spatial transcriptomics), brain development (mouse scRNA-seq), and ageing (mouse scRNA-seq).
- Three image datasets: MNIST, CIFAR, and ImageNet (in image translation and alignment tasks).

In the intervening period, we also expanded our method’s theoretical foundations by establishing bounds on OMT maps under perturbations of the probability measures (Theorem 2, stability bound, Appendix A.1). While such guarantees are generally unavailable for neural OT methods, they are a key feature of many non-neural OT solvers (ProgOT & HiRef).

We hope that by addressing these concerns, providing a broad range of experiments, and theoretically justifying the stability of OMT under perturbation, we have resolved the concerns raised. We believe these revisions showcase the capabilities and advantages of OMT compared to existing OT solvers.

---

> ### Author Response · Authors · 2025-11-25
> **OMT vs. GMM-OT**
>
> While OMT and GMM-OT both act on pairs of GMMs, they fundamentally differ in the problems they solve, their optimization strategies and solution guarantees. GMM-OT fits a GMM to the coupling function to solve the OT problem, which is non-convex and may lack a unique solution. Hence, multiple transport plans may achieve identical transport costs, hampering interpretability of transport. OMT, however, solves the entropic mixture transport problem, through a biconvex formulation. It utilizes a dual-entropic regularization scheme, one on component transport $p_{ij}$ and one on mixing weights $\omega_{ij}$. This formulation provides a significant computational advantage: it allows for an analytical closed-form solution (Eq. 13) for $p_{ij}$ (Eq. 21) and transport potentials, $\varphi_{ij}$ (Eq. 26), $\psi_{ij}$  (Eq. 27), simplifying the problem to optimizing only the mixing parameters.
> - As a concrete example for interpretability, in single-cell analysis, GMM-OT transports are not guaranteed to be unique, which leads to ambiguity and lack of repeatability in inferring biological trajectories. By contrast, OMT provably (Theorem 1) yields a unique and robust solution, making it well suited for single-cell analysis.
> - The entropic penalty term can be considered as a relaxation of the Maximum Entropy Principle [Jaynes, 1957]. This regularization towards more parsimonious solutions within the feasible set can lead to lower transport costs, in addition to improved interpretability.
>
> Please see Appendix B.4 of the updated manuscript, for further details.

---

> > ### Author Response · Authors · 2025-11-25
> > **OMT vs. HiRef**
> >
> > We want to clarify a fundamental difference between our proposed OMT and low-rank solvers like HiRef (Halmos et al., 2025). While both approaches aim to reduce computational complexity by operating in a reduced space, their underlying mathematical formulations are distinct. HiRef is rooted in the Monge problem, which seeks a deterministic transport map (permutation matrix) and requires an equal number of samples in the source and target domains. This formulation cannot model mass splitting or merging and is not guaranteed to always have a solution.
> >
> > In contrast, OMT learns a probabilistic coupling transport function. This formulation does not require equal sample sizes between the source and target domains and allows for one-to-many and many-to-one mappings. Additionally, the biconvex formulation in OMT is guaranteed to admit a unique solution (Theorem 1).

---

### Author Response · Authors · 2025-11-29
**Summary of contribution and rebuttal improvements**

To assist with the final decision process, here, we provide a summary of our work and the enhancements implemented during the rebuttal period.

We proposed Optimal Mixture Transport (OMT), an entropic, static, non-neural solver designed for computational efficiency and theoretical robustness (admitting unique solution and stability bound, Theorems 1 & 2). We demonstrated that OMT can be positioned as a highly effective alternative: while offering significant advantages in speed and interpretability, it yields competitive results compared to neural OT solvers, and consistently outperforms SOTA non-neural solvers (e.g., ProgOT, HiRef) in high-dimensional and large-scale settings.

We thank the reviewers for their valuable feedback, which focused on requests for clarification and additional experiments. During the rebuttal, we incorporated all of them to strengthen the manuscript. We clarified key contributions, theoretical points, and expanded our analytical and experimental scope as follows:

- We provided a stability bound for the proposed OMT map (new Theorem 2).

- We added two new datasets (ImageNet, MERFISH) and showcased OMT’s scalability and superior performance (Figures 4, 11, 12, Tables 2, 4).

- We included more baselines (HiRef, LOTs, Amortized W2-OT, GMM-OT) where OMT outperforms across multiple experiments (Tables 1,2, 4, 6 and Figures 2, 7).

- We benchmarked OMT against non-entropic GMM-OT formulations to evaluate transport map robustness. Our experiments demonstrated that OMT’s biconvex formulation yields superior stability against noise compared to standard GMM approaches (Figure 8, Appendix B.4).

- The revised manuscript now provides a comprehensive benchmark across 8 datasets, 6 tasks, and 10 baseline OT solvers.

We thus position the proposed method as follows:

- OMT consistently outperforms SOTA non-neural solvers.

- OMT offers highly competitive performance against neural OT methods. While it does not beat all neural OT solvers in every metric, it offers superior analytical features (theoretical guarantees and interpretable maps) and computational efficiency (substantially faster).

All updates and newly added content are highlighted in blue in the latest submission.

---

### Meta-Review · Area_Chair_tqbK · 2026-01-03

**Summary:**

The paper introduces Optimal Mixture Transport (OMT) as a framework for efficient and interpretable optimal transport between complex distributions by operating at the level of mixture models, with the main contribution being a doubly entropic, biconvex objective that admits a unique minimizer and allows most computations to reduce to mixture-level couplings with closed-form inner terms. While the problem setting is well motivated and the formulation is technically appealing, several major weaknesses remain despite a substantial rebuttal. First, although concerns about novelty, positioning, and attribution were partly addressed by clarifying that the proposed entropic formulation and solver are new and by adding theoretical results on uniqueness and stability, it remains difficult to disentangle how much of the observed empirical benefit stems from the mixture approximation itself versus the specific double-regularized OMT formulation, as tighter ablations are still lacking. Second, the empirical evaluation, while significantly expanded in the rebuttal with many new baselines (GMM-OT, hierarchical and low-rank methods, amortized W2OT), datasets (MERFISH, ImageNet), and additional metrics, remains only partially convincing: some comparisons feel selective, alternative OT solvers such as dynamic or neural approaches are not exhaustively covered, and evidence for broad state-of-the-art performance (especially in image translation) is still not definitive. Third, scalability and modeling trade-offs persist as a concern: although the authors provide arguments, bounds, and experiments suggesting that large numbers of mixture components can work in practice, the quadratic scaling in the number of components and the tension between interpretability and overparameterization are not fully resolved. Finally, while several presentation and evaluation gaps (invertibility interpretation, missing baselines, biological metrics, and baseline configurations) were addressed in a generally convincing manner during rebuttal, the volume and depth of changes highlight that the paper would benefit from a more thorough restructuring. In conclusion, the authors clearly invested a major rebuttal effort and improved the submission along many dimensions, but the remaining questions about attribution of gains, comparative positioning, and empirical conclusiveness indicate that an in-depth rewrite is still needed, and the paper cannot be accepted in its current form.

**Reviewer Concerns:**

Several key reviewer concerns were meaningfully addressed in the rebuttal: (i) invertibility concerns were clarified by positioning OMT as a probabilistic coupling allowing mass splitting/merging rather than a bijective map, (ii) missing baselines were partially remedied via added comparisons (notably GMM-OT, low-rank/HiRef-style methods, and amortized W2OT), (iii) scalability was strengthened with added large-scale experiments (e.g., MERFISH and ImageNet) and additional discussion of embeddings/latent spaces, and (iv) presentation issues (definitions, added error bars, clarified experimental methodology, and some biological-space evaluation) improved. However, important issues remain outstanding: it is still hard to attribute gains specifically to the doubly-regularized formulation rather than the mixture approximation pipeline, the empirical picture (especially for images and broad “SOTA” claims) remains mixed and somewhat selective, and the scalability/interpretability tension (quadratic component scaling, overparameterization reducing interpretability) is not fully resolved. A deeper structural rewrite also still seems warranted given how much was added during rebuttal.

**Reviewer Scores:**

- 8kbG (initial score: 4): likely no change (concerns about invertibility were clarified and extra benchmarks were added, but the core “acceptability” threshold was already near and the remaining conclusiveness issues likely keep them at a cautious 4).
- Dmnz (initial score: 4): likely no change (major baseline omissions were addressed (GMM-OT, low-rank, etc.), but their central concern about separating “GMM approximation gains” from “new formulation gains” is only partially answered and would probably keep them at 4).
- A9fm (initial score: 2): would likely have increased to 4 if able to participate fully (they explicitly stated a provisional move from 2 to 4 after discussion, and the rebuttal directly engaged their novelty and experimental concerns).
- JQ23 (initial score: 4): likely no change (the rebuttal added biological/gene-level analyses and included amortized W2OT, with a follow-up correction to the W2OT architecture choice; this likely maintains a borderline-but-open stance around 4 rather than shifting materially).

---

### Decision · Program_Chairs · 2026-01-26

Reject